# Seasonal variability of nitrous oxide concentrations and emissions in a temperate estuary

Gesa Schulz[1,2], Tina Sanders[2], Yoana G. Voynova[2], Hermann W. Bange[3], and Kirstin Dähnke[2]

[1]Institute of Geology, Center for Earth System Research and Sustainability (CEN), University Hamburg, Hamburg, 20146, Germany
[2]Institute of Carbon Cycles, Helmholtz Centre Hereon, Geesthacht, 21502, Germany
[3]Marine Biogeochemistry Research Division, GEOMAR Helmholtz Centre for Ocean Research Kiel, Kiel, 24105, Germany

*Correspondence to*: Gesa Schulz (Gesa.Schulz@hereon.de)

## Abstract

Nitrous oxide ($N_2O$) is a greenhouse gas, with a global warming potential 298 times that of carbon dioxide. Estuaries can be sources of $N_2O$, but their emission estimates have significant uncertainties due to limited data availability and high spatiotemporal variability. We investigated the spatial and seasonal variability of dissolved $N_2O$ and its emissions along the Elbe Estuary (Germany), a well-mixed temperate estuary with high nutrient loading from agriculture. During nine research cruises performed between 2017 and 2022, we measured dissolved $N_2O$ concentrations, as well as dissolved nutrients and oxygen concentrations along the estuary and calculated $N_2O$ saturations, flux densities and emissions. We found that the estuary was a year-round source of $N_2O$, with highest emissions in winter when dissolved inorganic nitrogen (DIN) loads and wind speeds are high. However, in spring and summer, $N_2O$ saturations and emissions did not decrease alongside lower riverine nitrogen loads, suggesting that estuarine in-situ $N_2O$ production is an important source of $N_2O$. We identified two hot-spots areas of $N_2O$ production: the Port of Hamburg, a major port region, and the mesohaline estuary near the maximum turbidity zone (MTZ). $N_2O$ production was fueled by decomposition of riverine organic matter in the Hamburg Port and by marine organic matter in the MTZ. A comparison with previous measurements in the Elbe Estuary revealed that $N_2O$ saturation did not decrease alongside the decrease in DIN concentrations after a significant improvement of water quality in the 1990s that allowed for phytoplankton growth to reestablish in the river and estuary. The overarching control of phytoplankton growth on organic matter and, subsequently, on $N_2O$ production, highlights the fact that eutrophication and elevated agricultural nutrient input can increase $N_2O$ emissions in estuaries.

## 1 Introduction

Nitrous oxide ($N_2O$) is an important atmospheric trace gas that contributes to global warming and stratospheric ozone depletion (WMO, 2018; IPCC, 2021). Estuaries are important regions of nitrogen turnover (Middelburg and Nieuwenhuize, 2000; Crossland et al., 2005; Bouwman et al., 2013), and a potential source of $N_2O$ (Bange, 2006; Barnes and Upstill-Goddard, 2011; Murray et al., 2015). Together with coastal wetlands, estuaries contribute between 0.17 and 0.95 Tg $N_2O$-N of the annual global budget of 16.9 Tg $N_2O$-N (Murray et al., 2015; Tian et al., 2020). $N_2O$ emission estimates from estuaries are associated with significant uncertainties due to limited data availability and high spatiotemporal variability (e.g. Bange, 2006; Barnes and Upstill-Goddard, 2011; Maavara et al., 2019), presenting a big challenge for the global $N_2O$ emission estimates.

Nitrification and denitrification are the most important $N_2O$ production pathways in estuaries. Under oxic conditions, $N_2O$ is produced as a side product during the first step of nitrification, the oxidation of ammonia to nitrite (e.g. Wrage et al., 2001; Barnes and Upstill-Goddard, 2011). At low oxygen (but not anoxic) conditions, nitrifier-denitrification may occur, during which nitrifiers reduce nitrite to $N_2O$ (e.g. Wrage et al., 2001; Bange, 2008). Denitrification takes place under anoxic conditions and mostly acts as a source of $N_2O$, but can also reduce $N_2O$ to $N_2$ (e.g. Knowles, 1982; Bange, 2008). In estuaries, denitrification can occur in anoxic sediments, the anoxic water column or anoxic microsites of particles, whereas nitrification and nitrifier-denitrification take place in the oxygenated water column (Beaulieu et al., 2010; Murray et al., 2015; Ji et al., 2018; Tang et al., 2022).

In estuaries, the most important factors controlling $N_2O$ emissions are considered to be oxygen availability and dissolved inorganic nitrogen loads (Murray et al., 2015). Since $N_2O$ measurements in estuaries are scarce, global $N_2O$ emissions can be estimated by using emission factors and considering dissolved inorganic nitrogen (DIN) or total nitrogen (TN) loads, where it is assumed that higher nitrogen loads lead to higher $N_2O$ emissions (Kroeze et al., 2005, 2010; Ivens et al., 2011; Hu et al., 2016). However, several studies instead reported no obvious relationship between nitrogen concentrations and $N_2O$ emissions (Borges et al., 2015; Marzadri et al., 2017; Wells et al., 2018), highlighting the need to understand the causes for variability in the relationship between nitrogen loads and $N_2O$ emissions (Wells et al., 2018).

The Elbe Estuary is a heavily managed estuary with high agricultural nitrogen inputs that hosts the third largest port in Europe (e.g. Radach and Pätsch, 2007; Bergemann and Gaumert, 2008; Pätsch et al., 2010; Quiel et al., 2011). It has been identified as a $N_2O$ source, with a hotspot of $N_2O$ production in the Port of Hamburg (Hanke and Knauth, 1990; Brase et al., 2017). We aimed to investigate drivers for $N_2O$ emissions along the estuary, specifically the $N_2O$ and DIN ratio ($N_2O$:DIN). To do so, we (1) looked for potential long-term changes in $N_2O$ saturations, (2) investigated potential production hotspots, as well as the spatial and temporal distribution of $N_2O$ saturations, and (3) used the $N_2O$:DIN ratio for a comparison with other estuaries.

## 2    Methods

### 2.1    Study site

The Elbe River stretches over 1094 km from the Giant Mountains (Czech Republic) to the North Sea (Cuxhaven, Germany). The catchment of the Elbe River is 140 268 km² (Boehlich and Strotmann, 2019), with 74 % urban and agricultural land-use (Johannsen et al., 2008).The Elbe is the second largest German river discharging into the North Sea, as well as the largest source of dissolved nitrogen for the German Bright, which is heavily affected by eutrophication (van Beusekom et al., 2019).

The Elbe Estuary is a well-mixed temperate estuary, which begins at stream kilometer 586 at a weir in Geesthacht and stretches through the Port of Hamburg, entering the North Sea near Cuxhaven at stream kilometer 727 (Fig. 1). Estuaries are commonly structured along their salinity gradient into an oligohaline (salinity: 0.5 – 5.0), a mesohaline (salinity: 5.0 – 18.0) and a polyhaline (salinity > 18.0) region (US EPA, 2006). The Elbe Estuary has a length of 142 km (Boehlich and Strotmann, 2019) and a mean annual discharge of 712 m³ s$^{-1}$ (measured at gauge Neu Darchau at stream kilometer 536; HPA and Freie und Hansestadt Hamburg, 2017). The average water residence time is ~32 days, ranging from ~72 days during times of low discharge (300 m³ s$^{-1}$) to ~10 days during times of high discharge (2000 m³ s$^{-1}$; Boehlich and Strotmann, 2008). The Elbe Estuary has an annual nitrogen load of 84 Gg-N (FGG Elbe, 2018), and point sources along the estuary provide only a small part of the total

nitrogen input (Hofmann et al., 2005; IKSE, 2018). Oxygen concentrations in the Elbe Estuary vary seasonally,
with oxygen depletion during the summer months and oxygen minimum zones regularly experiencing
concentrations below 94 µmol $O_2$ $L^{-1}$ (Schroeder, 1997; Gaumert and Bergemann, 2007; Schöl et al., 2014).
The Elbe Estuary is dredged year-round to maintain a water depth of 15 – 20 m and to grant access for large
container ships to the Port of Hamburg (Boehlich and Strotmann, 2019; Hein et al., 2021). Construction work for
further deepening of the fairway was carried out during our study period, from 2019 to early 2022. Upstream of
the Port of Hamburg water depth is less than 10 m (Hein et al., 2021).

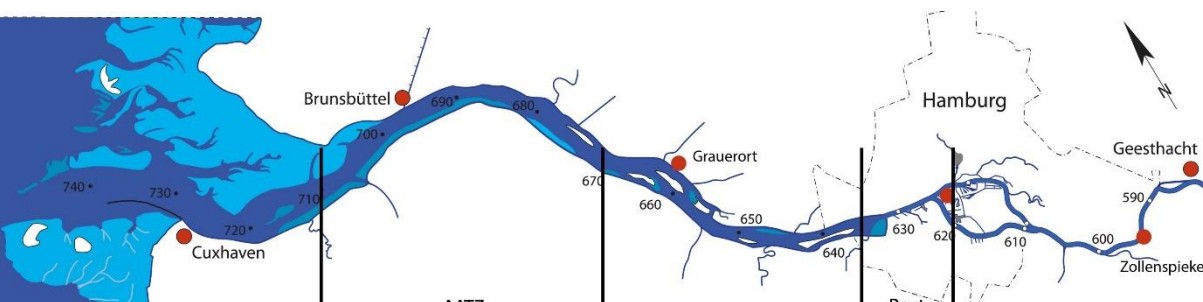

**Figure 1: Map of the Elbe Estuary sampled during our research cruises with stream kilometers (graphic courtesy of**
**FGG Elbe, modified after Amann et al. 2012)). The light blue color indicates Wadden Sea areas that are exposed at low**
**tide. The vertical black lines indicate the Hamburg Port region and a typical position for the maximum turbidity zone**
**(MTZ, Bergemann, 2004).**
**2.2    Transect sampling and measurements**
We performed nine sampling campaigns along the estuary with the research vessel *Ludwig Prandtl* (Table 1). Most
cruises took place during spring and summer, with water temperatures > 10 °C (May to September). Two cruises
were conducted during winter (early March, water temperature < 6 °C; Table 1). Transects started in the German
Bight, and continued along the salinity gradient, through the Port of Hamburg to Oortkaten (stream kilometer 609).
To ensure comparable current and mixing conditions, transect sampling was always done after high-tide, with the
ship travelling upstream against the tide. For comparison to previous measurements, we included summer data
from a previous study in 2015 (Brase et al., 2017).
**Table 1: Campaign dates with the sampled Elbe Estuary sections shown via stream kilometers, average discharge**
**during each cruise measured at the Neu Darchau gauging station, averages and standard deviations for water**
**temperature, wind speed at 10 m height, dissolved inorganic nitrogen (DIN) concentrations for each campaign.**

| Campaign Dates | Stream kilometers (km) | Water temperature (°C) | Wind speed 10 m ($m\ s^{-1}$) | Average discharge ($m^3\ s^{-1}$) | Average DIN concentrations ($\mu mol\ L^{-1}$) |
|---|---|---|---|---|---|
| 28.-29.04.2015 | 627 – 741 | $12.3 \pm 1.0$ | $7.4 \pm 2.3$ | 595 | $191.0 \pm 45.0$ |
| 02.-04.06.2015 | 609 – 739 | $17.4 \pm 1.7$ | $5.0 \pm 1.3$ | 276 | $105.9 \pm 36.2$ |
| 01.-02.08.2017 | 621 – 749 | $20.9 \pm 0.7$ | $3.6 \pm 1.5$ | 607 | $79.2 + 30.2$ |
| 04.-05.06.2019 | 610 – 750 | $18.7 \pm 2.2$ | $4.0 \pm 1.7$ | 423 | $108.3 \pm 35.9$ |
| 30.07.-01.08.2019 | 609 – 752 | $22.6 \pm 1.0$ | $4.2 \pm 1.4$ | 171 | $60.8 \pm 38.6$ |
| 19.-20.06.2020 | 609 – 747 | $19.8 \pm 1.4$ | $5.8 \pm 1.2$ | 331 | $74.6 \pm 33.8$ |
| 09.-11.09.2020 | 607 – 745 | $18.9 \pm 0.6$ | $5.9 \pm 2.8$ | 305 | $93.1 \pm 32.7$ |
| 10.-12.03.2021 | 609 – 748 | $5.4 \pm 0.5$ | $9.3 \pm 2.6$ | 862 | $324.4 \pm 83.8$ |
| 04.-05.05.2021 | 610 – 751 | $10.5 \pm 0.8$ | $11.0 \pm 3.1$ | 411 | $85.7 \pm 36.6$ |
| 27.-28.07.2021 | 621 – 751 | $22.2 \pm 0.7$ | $5.2 \pm 1.3$ | 721 | $139.8 \pm 58.4$ |
| 01.-02.03.2022 | 610 – 752 | $5.6 \pm 0.2$ | $2.9 \pm 1.0$ | 1282 | $238.0 \pm 74.7$ |

An onboard membrane pump continuously provided water at 1.2 m depth to an on-line in-situ FerryBox system
and to an equilibrator used for the measurements of $N_2O$ dry mole fraction (Section 2.4). The FerryBox system
continuously measured water temperature, salinity, oxygen concentrations, pH and turbidity. We corrected the
salinity corrected optode measurements using comparisons to Winkler titrations of discrete samples. See Table S1
for further details.
Discrete water samples (30-40 samples for each cruise) were collected every 20 min from a bypass of the FerryBox
system. For nutrient analysis, water samples were filtered immediately through combusted, pre-weighted GF/F
Filters (4 h, 450 °C), and were frozen in acid washed PE-bottles until analysis. The filters were also stored frozen
(-20 °C) and subsequently analyzed for suspended particulate matter (SPM), particulate nitrogen (PN), total
particulate carbon (PC) and C/N ratios (Fig. S1).
**2.3 Nutrient measurements**
Filtered water samples were measured in triplicates with a continuous flow auto analyzer (AA3, SEAL Analytics)
using standard colorimetric and fluorometric methods (Hansen and Koroleff, 1999) for dissolved nitrate ($NO_3^-$),
nitrite ($NO_2^-$) and ammonium ($NH_4^+$) concentrations. Detection limits were $0.05\ \mu mol\ L^{-1}$, $0.05\ \mu mol\ L^{-1}$, and
$0.07\ \mu mol\ L^{-1}$ for nitrate, nitrite and ammonium, respectively.
**2.4 Equilibrator based $N_2O$ measurements and calculations**
Equilibrated dry mole fractions of $N_2O$ were measured by an $N_2O$ analyzer based on off-axis integrated cavity
output (OA-ICOS) absorption spectroscopy (Model 914-0022, Los Gatos Res. Inc., San Jose, CA, USA), which
was coupled with a seawater/gas equilibrator using off-axis cavity output spectroscopy. Brase et al. (2017)
described the set-up and instrument precision in detail. Twice a day, two standard gas mixtures of $N_2O$ in synthetic
air (500.5 ppb $\pm$ 5 % and 321.2 ppb $\pm$ 3 %) were analyzed to validate our measurements. No drift was detected
during our cruises.

We calculated the dissolved $N_2O$ concentrations in water with the Bunsen solubility function of Weiss and Price (1980), using 1 min averages of the measured $N_2O$ dry mole fraction (ppb). Temperature differences between the sample inlet and the equilibrator were taken into account for the calculation of the final $N_2O$ concentrations Rhee et al. (2009). $N_2O$ saturation was calculated based on $N_2O$ concentrations in water ($N_2O_{cw}$) and the atmospheric equilibration concentrations ($N_2O_{eq}$; Eq. 1). Atmospheric $N_2O$ dry mole fractions were measured before and after each transect cruises using an air duct from the deck of the research vessel.

$$s = 100 \times \frac{N_2O_{cw}}{N_2O_{eq}} \tag{1}$$

The gas transfer coefficients ($k$) were determined based on Borges et al. (2004, Eq. 3), Nightingale et al. (2000), Wanninkhof (1992) and Clark et al. (1995), using the Schmidt number ($Sc$) and wind speeds ($u_{10}$) measured at 10 m height (Eq. 2). The Schmidt number was calculated as ratio of the kinematic viscosity in water (Siedler and Peters, 1986) to the $N_2O$ diffusivity in water (Rhee, 2000). Cruise wind speeds (Table 1) varied significantly from average annual wind speeds of the two federal states, in which the Elbe Estuary is located (4.7 m s$^{-1}$, Schleswig-Holstein u. Hamburg: Mittlere Windgeschwindigkeit (1986-2015)* | Norddeutscher Klimamonitor, 2023), and also compared to seasonal average wind speeds determined for the stations Cuxhaven and Hamburg (Rosenhagen et al., 2011). Thus, to estimate uncertainties due to varying wind conditions during our cruises, we used 1) the *in-situ* wind speeds measured on board the *R/V Ludwig Prandtl* at 10 m height by a MaxiMet GMX600 (Gill Instruments Limited, Hampshire, UK), 2) the average annual wind speed (Schleswig-Holstein u. Hamburg: Mittlere Windgeschwindigkeit (1986-2015)* | Norddeutscher Klimamonitor, 2023), and 3) the seasonally averaged wind speeds (Rosenhagen et al., 2011). The flux densities in the main text were calculated using Eq. 3 and the wind speeds measured on board the vessel. Results of the other calculations are listed in the supplementary material (Table S2).

$$k = 0.24 \times (4.045 + 2.58u_{10}) \times \left(\frac{Sc}{600}\right)^{-0.5} \tag{2}$$

$$f = k \times (N_2O_{cw} - N_2O_{air}) \tag{3}$$

To estimate $N_2O$ emissions, we separated the Elbe Estuary into five regions: limnic (stream kilometer 585 to 615), Port of Hamburg (stream kilometer 615 to 632), oligohaline (stream kilometer 632 to 704), mesohaline (stream kilometer 704 – 727) and polyhaline (stream kilometer 727 to 750), see Table S3. Respective areas were provided by the German Federal Waterways Engineering and Research Institute (BAW, pers. Comm., Oritz, 2023) and Geerts et al. (2012). In order to account for seasonality, cruises were defined as: winter (March), spring (April and May), summer (June and July) and late summer/autumn (August and September). We then calculated daily $N_2O$ emissions per section and season. For upscaling, we used the calculated monthly emissions to estimate annual emissions (winter: November to March, spring: April to May, summer: June to July and late summer/autumn: August to October). To address uncertainties, we calculated $N_2O$ emissions based on different parametrizations and wind speeds as described above.

## 2.5 Excess $N_2O$ and apparent oxygen utilization

The correlation between excess $N_2O$ ($N_2O_{xs}$) and apparent oxygen utilization ($AOU$) can provide insights into $N_2O$ production (Nevison et al., 2003; Walter et al., 2004). We calculated $N_2O_{xs}$ as the difference between the $N_2O$ concentration in water and the theoretical equilibrium concentration ($N_2O_{eq}$) (Eq. 4). AOU was determined using

Eq. 5, where $O_2$ is the measured dissolved oxygen concentration, and $O_2'$ is the theoretical equilibrium
concentration between water and atmosphere calculated according to Weiss (1970).

$$N_2O_{xs} = N_2O_{cw} - N_2O_{eq} \qquad (4)$$

$$AOU = O_2' - O_2 \qquad (5)$$

A linear relationship between AOU and $N_2O_{xs}$ is usually an indicator for $N_2O$ production from nitrification
(Nevison et al., 2003; Walter et al., 2004).

### 2.6 Statistical analysis

All statistical analyses were done using R packages. The packages ggpubr v.0.6.0 (Kassambara, 2023) and stats
v.4.0.2 (The R Stats Package, Version 4.0.2, 2021) were used to calculate Pearson correlations ($R$) and $p$-values.

## 3 Results

### 3.1 Hydrographic properties and DIN distribution

Discharge ranged between 171 m³ s⁻¹ and 1282 m³ s⁻¹ during our cruises (ZDM, 2022), with higher discharge in
winter and lower discharge in summer (Table 1). Average water temperature over the entire estuary ranged from
5.4 ± 0.5 °C in March 2021 to 22.6 ± 1.0 °C in August 2017 (Table 1). For further evaluation, March 2021 and
2022 cruises were regarded as winter cruises (water temperature < 6 °C), whereas all cruises with higher water
temperature were jointly regarded as spring and summer conditions.
Nitrate was the major form of dissolved inorganic nitrogen (DIN) during all cruises. In winter, high nitrogen
concentrations entered the estuary from the river. Towards summer, the riverine input of nitrate (stream kilometer
< 620) decreased, but along the estuary nitrate concentrations increased up to ~stream kilometer 700, then
decreased towards the North Sea. Nitrate concentrations were highest during both March cruises with averages of
319.0 ± 85.7 µmol L⁻¹ and 230.9 ± 76.2 µmol L⁻¹ in 2021 and 2022, respectively. During summer, nitrate
concentrations were lower, with averages between 151.0 ± 58.1 µmol L⁻¹ in May 2021 and 63.3 ± 38.8 µmol L⁻¹
in July 2019 (Fig. 2a and b).
Nitrite and ammonium concentrations were usually low (< 1 µmol L⁻¹) throughout the Elbe Estuary, but peaked
in the Hamburg Port region and around stream kilometer 720 (Fig. 2c and 2e). We measured pronounced variations
in nitrite concentrations during most of our cruises, ranging from > 6.0 µmol L⁻¹ (July 2019) to concentrations
below the detection limit (Fig. 2c and d). The highest ammonium concentration was measured in March 2021 at
23.5 µmol L⁻¹ (Fig. 2e and f).

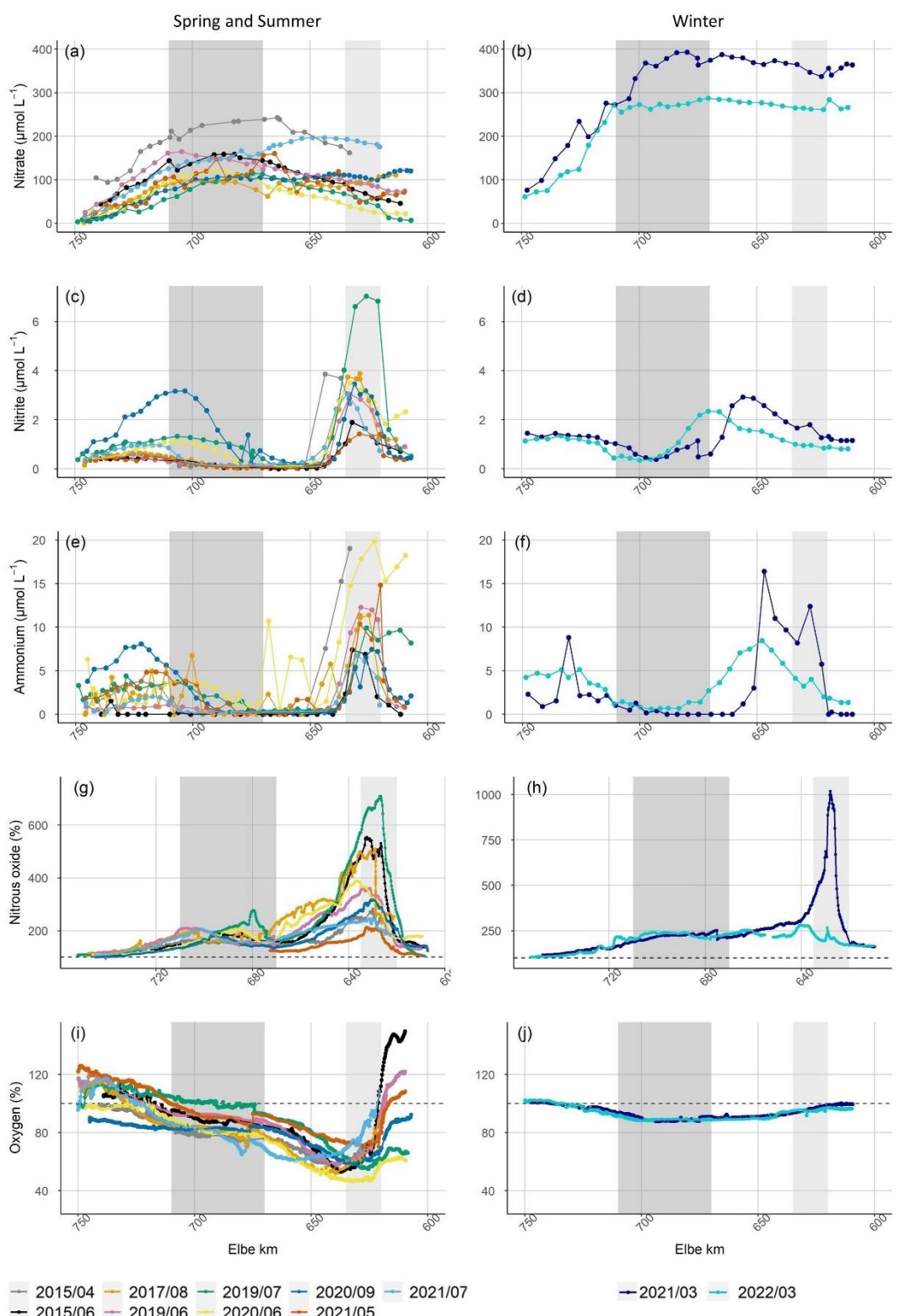


**Figure 2: Nitrate concentration along the Elbe Estuary (a) in spring/summer, (b) in winter. Nitrite concentration along the Elbe Estuary (c) in spring/summer and (d) in winter. Ammonium concentration along the Elbe Estuary (e) in spring/summer and (f) in winter. N$_2$O in % saturation along the Elbe Estuary (g) in spring/summer, (h) in winter. Dissolved oxygen in % saturation along the Elbe Estuary (i) in spring/summer and (j) in winter. All variables are plotted against Elbe stream kilometers (Elbe km). Light grey shading denotes the Hamburg Port region, dark grey shading the typical position of the maximum turbidity zone (MTZ, Bergemann, 2004). Note the difference in Y-axis scales for the plots of (g) and (h). The dashed black lines in (g) and (h), as well (i) and (j) indicate saturation of 100 % for nitrous oxide and dissolved oxygen, respectively.**

## 3.2 Atmospheric $N_2O$ and $N_2O$ saturation

The average atmospheric $N_2O$ dry mole fractions ranged from 325 ppb in June 2015 to 336 ppb in July 2022 (Table 2). The differences between our measurements and the mean monthly $N_2O$ mole fraction measured at the Mace Head atmospheric monitoring station (Ireland; Dlugokencky et al., 2022) were always less than 1.5 %, indicating a good agreement with the monitoring data.

During all cruises, the Elbe Estuary was supersaturated in $N_2O$ in the freshwater region (Fig. 2g, h). The average $N_2O$ saturation over the entire transect ranged between 146 % and 243 % with an overall average of 197 % for all cruises. Highest $N_2O$ occurred in the Hamburg Port region in spring and summer with an average $N_2O$ peak of 402 % saturation and a maximum supersaturation of 710 % in July 2019. The distributions of $N_2O$ during winter cruises were significantly different: In March 2022, highest $N_2O$ (280 % saturation) occurred at stream kilometer 640. In contrast, in March 2021, we found an extraordinarily high peak with a saturation of 1018 % at stream kilometer 627. Between stream kilometer 680 and 720, a supersaturation of up to 277 % occurred in spring and summer. Further towards the North Sea, $N_2O$ decreased, approaching equilibrium with the atmosphere.

## 3.3 $N_2O$ flux densities and $N_2O$ emissions

For $N_2O$ flux densities, we present calculated values after Borges et al. (2004, Table 2), but also include results using other parametrizations in Table S2 and Fig. S2. The $N_2O$ flux densities were usually highest in the Hamburg Port area, with an average of $95.0 \pm 97.9$ µmol m$^{-2}$ d$^{-1}$ and lowest towards the North Sea, with an average of $3.9 \pm 3.0$ µmol m$^{-1}$ d$^{-1}$ (Elbe stream kilometers > 735). The average $N_2O$ flux density of all cruises was $39.9 \pm 46.9$ µmol m$^{-2}$ d$^{-1}$ (calculated with *in-situ* wind speeds measured during the cruises).

**Table 2: Calculated average $N_2O$ saturation, sea-to-air fluxes calculated following Borges et al. (2004) and atmospheric $N_2O$ dry mole fractions during our cruises in the Elbe Estuary**

| Campaign Dates | Average saturation (%) | $N_2O$ Flux densities (µmol m$^{-2}$ d$^{-1}$) | | | Average atmospheric dry mole fraction (ppb) |
|---|---|---|---|---|---|
| | | In-situ wind | Annual wind | Seasonal wind | |
| 28.-29.04.15 | $160.8 \pm 37.9$ | $33.1 \pm 21.0$ | $23.1 \pm 14.7$ | $25.4 \pm 16.1$ | $331 \pm 0.5$ |
| 02.-04.06.15 | $203.8 \pm 112.7$ | $39.0 \pm 42.7$ | $37.2 \pm 40.7$ | $37.8 \pm 41.4$ | $325 \pm 0.8$ |
| 01.-02.08.17 | $221.0 \pm 106.5$ | $35.6 \pm 31.8$ | $43.2 \pm 38.5$ | $44.1 \pm 39.3$ | $331 \pm 1.2$ |
| 04.-05.06.19 | $192.6 \pm 66.0$ | $29.7 \pm 21.5$ | $33.5 \pm 24.2$ | $34.0 \pm 24.6$ | $332 \pm 0.2$ |
| 30.07.-01.08.19 | $232.5 \pm 155.3$ | $42.0 \pm 50.1$ | $45.7 \pm 54.5$ | $47.4 \pm 56.4$ | $327 \pm 1.0$ |
| 19.-20.06.20 | $193.9 \pm 74.1$ | $39.2 \pm 31.6$ | $33.3 \pm 26.9$ | $33.9 \pm 27.3$ | $330 \pm 0.6$ |
| 09.-11.09.20 | $160.5 \pm 53.6$ | $26.0 \pm 23.5$ | $21.8 \pm 19.7$ | $24.5 \pm 22.1$ | $331 \pm 0.7$ |
| 10.-12.03.21 | $242.5 \pm 141.6$ | $100.7 \pm 101.2$ | $58.1 \pm 58.4$ | $71.0 \pm 71.4$ | $331 \pm 1.3$ |
| 04.-05.05.21 | $145.6 \pm 28.8$ | $35.6 \pm 22.5$ | $17.8 \pm 11.2$ | $18.5 \pm 11.7$ | $331 \pm 0.8$ |
| 27.-28.07.21 | $172.6 \pm 37.2$ | $28.0 \pm 14.6$ | $25.9 \pm 13.6$ | $26.9 \pm 14.1$ | $334 \pm 3.8$ |
| 01.-02.03.22 | $196.5 \pm 47.0$ | $27.8 \pm 13.9$ | $39.0 \pm 19.5$ | $47.7 \pm 23.8$ | $333 \pm 0.7$ |

$N_2O$ emission estimates varied significantly depending on the used parametrization and wind speeds. Note that we calculated emissions twice: 1) including (w 03/2021) and 2) deliberately excluding (w/o 03/2021) the $N_2O$ peak saturation measured in the Port of Hamburg in March 2021, using linearly interpolated concentrations, respectively. Highest emissions were calculated following methods by Borges et al. (2004) and using *in-situ* wind

speeds, resulting in emissions of  $0.25 \pm 0.16$ Gg-N$_2$O yr$^{-1}$ and $0.23 \pm 0.12$ Gg-N$_2$O yr$^{-1}$ with and without the N$_2$O
peak in March 2021, respectively. Lowest emissions of 0.08 Gg-N$_2$O yr$^{-1}$ arose with parametrization of Nightingale
et al. (2000) and Wanninkhof (1992), and using annual wind speeds (Table 3).
**Table 3: Annual N$_2$O emission estimates in Gg-N$_2$O yr$^{-1}$ calculated with different parametrizations and wind speeds**

| | | Emissions in Gg-N$_2$O yr$^{-1}$ | | | |
| | | Borges et al. (2004) | Nightingale et al. (2000) | Wanninkhof (1992) | Clark et al. (1995) |
|---|---|---|---|---|---|
| w 03/2021 | In-situ wind | $0.25 \pm 0.16$ | $0.14 \pm 0.12$ | $0.17 \pm 0.15$ | $0.16 \pm 0.12$ |
| | Annual wind | $0.21 \pm 0.11$ | $0.08 \pm 0.04$ | $0.09 \pm 0.05$ | $0.09 \pm 0.05$ |
| | Seasonal wind | $0.24 \pm 0.12$ | $0.11 \pm 0.06$ | $0.13 \pm 0.06$ | $0.12 \pm 0.06$ |
| w/o 03/2021 | In-situ wind | $0.23 \pm 0.12$ | $0.13 \pm 0.09$ | $0.15 \pm 0.11$ | $0.14 \pm 0.09$ |
| | Annual wind | $0.20 \pm 0.08$ | $0.08 \pm 0.03$ | $0.08 \pm 0.03$ | $0.09 \pm 0.04$ |
| | Seasonal wind | $0.22 \pm 0.09$ | $0.11 \pm 0.04$ | $0.12 \pm 0.04$ | $0.12 \pm 0.04$ |

**3.4  Dissolved oxygen saturation**
Average oxygen varied between 76 and 95 in % saturation with an oxygen minimum in the Hamburg Port area.
Winter cruises varied little, with oxygen remaining relatively constant along the estuary (> 88 % saturation).
During most spring and summer cruises, water from the river coming into the estuary was supersaturated in oxygen
(> 100 % saturation). In the Hamburg Port region, oxygen saturation generally decreased. Lowest values occurred
in June 2020 with 47 % saturation. The along-estuary oxygen minimum in summer months (June to August) was
always below 61 % saturation. In spring and summer, oxygen increased towards the North Sea and reached
100 % saturation (Fig. 2i and j).
Plots of excess N$_2$O (N$_2$O$_{xs}$) and apparent oxygen utilization (AOU) revealed excess N$_2$O along the entire estuary
(Fig. 3). During all cruises, elevated riverine N$_2$O$_{xs}$ entered the estuary (stream kilometer < 620). A linear positive
relationship between N$_2$O$_{xs}$ and AOU suggested nitrification as main production pathway in large sections of the
estuary (Nevison et al., 2003; Walter et al., 2004). However, in summer, a change of slope in the Port of Hamburg
as well as in the mesohaline section of the estuary suggested either increased in-situ N$_2$O production or external
N$_2$O input. In winter, we found an increasing slope in the Hamburg Port region and in the oligohaline part of the
Elbe Estuary (Fig. 3h, k).

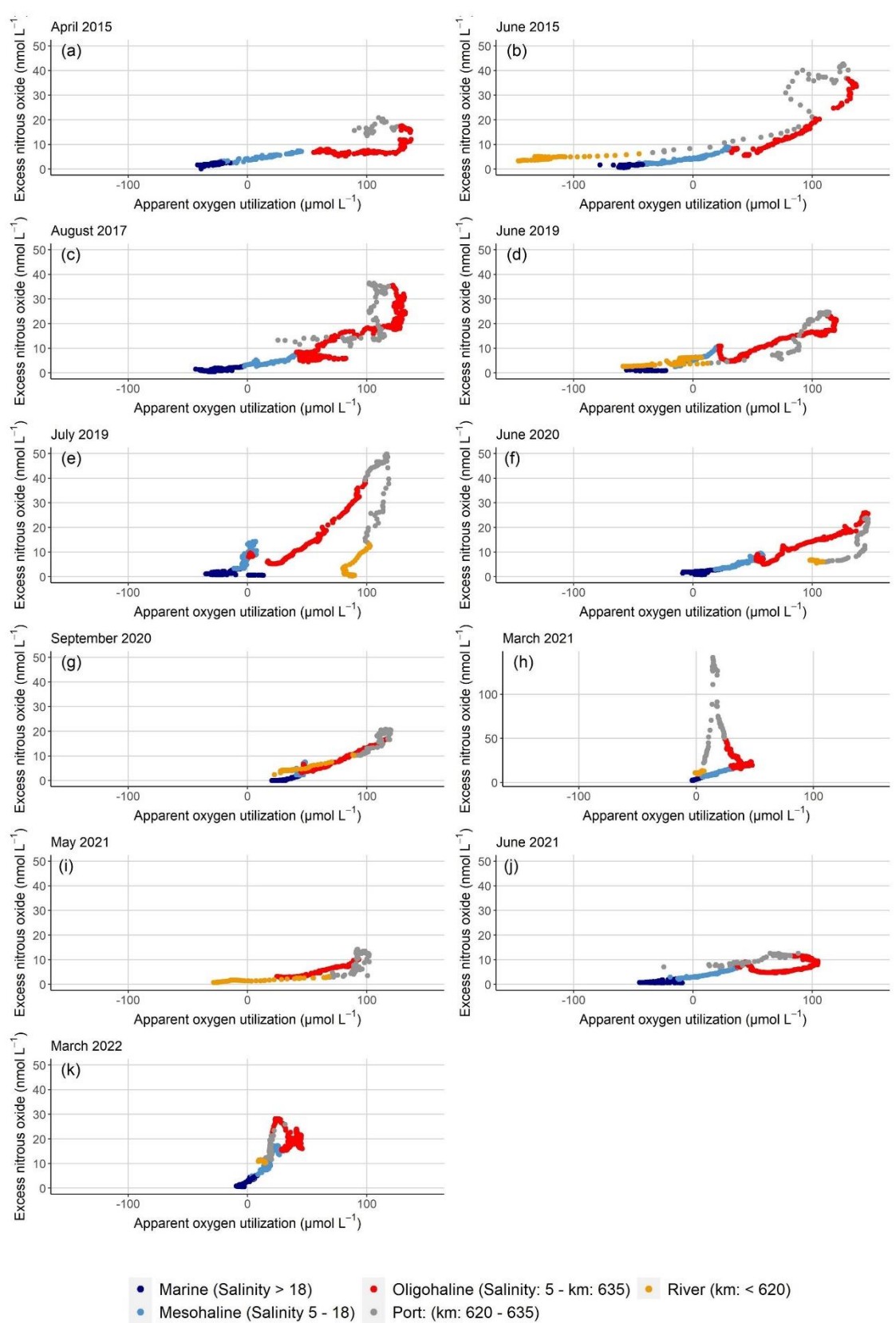


**Figure 3: Plots of $N_2O_{xs}$ vs AOU for (a) April 2015, (b) June 2015, (c) August 2017, (d) June 2019, (e) July 2019, (f) June 2020, (g) September 2020, (h) March 2021, (i) May 2021, (j) June 2021 and (k) March 2022. The values are colored to distinguish between different regions of the estuary. Y-axis scale differ for Fig. 3h.**

## 3.5 Statistical analysis

We performed statistical analyses to identify potential $N_2O$ production pathways and controlling factors. Table 4 summarizes the results for the entire data set with further separation into spring and summer cruises (sp/su), as well as separation according to the presence of a salinity gradient (salinity > 1) or of freshwater regions (salinity < 1). Furthermore, we performed corresponding analysis to assess the significance of correlations between for average values of different parameters for each cruise (Table 5). $N_2O$ saturation showed significant negative correlation with oxygen (Table 4) as well as a consistent negative correlation with pH (Table 4 and 5). Furthermore, nitrite concentrations positively correlated with $N_2O$ saturation in the freshwater section of the estuary (Table 4 and 5).

**Table 4: Pearson correlation coefficients (R) for $N_2O$ saturation (%) with temperature (T in °C), pH value, oxygen ($O_2$ in % saturation), ammonium concentrations ($NH_4^+$ in µmol $L^{-1}$), nitrite concentrations ($NO_2^-$ in µmol $L^{-1}$), nitrate concentrations ($NO_3^-$ in µmol $L^{-1}$), SPM concentrations (SPM in mg $L^{-1}$), C/N values, particulate carbon fraction (PC in %) and particulate nitrogen fraction (PN in %) for the entire data set, spring and summer cruises (sp/su), data with salinity > 1, spring and summer cruises with salinity > 1, data with salinity < 1 and spring and summer cruises with salinity < 1. The significance is shown as ** for p-value < 0.001, * for p-values < 0.01 and + for p-values < 0.05.**

| $N_2O$ saturation % | T °C | pH | $O_2$ % | $NH_4^+$ µM | $NO_2^-$ µM | $NO_3^-$ µM | SPM mg | C/N | PC % | PN % |
|---|---|---|---|---|---|---|---|---|---|---|
| Entire data | 0.06 | -0.47** | -0.56** | 0.27** | 0.48** | 0.23 | 0.10 | 0.60 | -0.05 | -0.13+ |
| sp/su | 0.33* | -0.59** | -0.65** | 0.23** | 0.53** | 0.09 | 0.02 | 0.24** | -0.09 | -0.13+ |
| Sal>1 | 0.03 | -0.40** | -0.53** | -0.32** | -0.05 | 0.71** | 0.32** | 0.11* | -0.24 | -0.39** |
| Sal<1, | 0.01 | -0.41** | -0.42** | 0.28** | 0.51** | -0.00 | -0.08 | 0.15 | -0.25* | -0.24* |
| Sal>1, sp/su | -0.10 | -0.21+ | -0.52** | -0.28** | 0.01 | 0.62** | 0.02 | 0.39** | -0.31** | -0.41** |
| Sal<1, sp/su | 0.30** | -0.60** | -0.57** | 0.21+ | 0.58** | -0.23* | -0.16. | 0.11 | -0.30* | -0.27* |

**Table 5: Pearson correlation coefficients (R) for average $N_2O$ saturation (%) with average discharge (Q in $m^3 s^{-1}$) temperature (T in °C), pH value, oxygen ($O_2$ in % saturation), ammonium concentrations ($NH_4^+$ in µmol $L^{-1}$), nitrite concentrations ($NO_2^-$ in µmol $L^{-1}$), nitrate concentrations ($NO_3^-$ in µmol $L^{-1}$), SPM concentrations (SPM in mg $L^{-1}$), C/N values, particulate carbon fraction (PC in %) and particulate nitrogen fraction (PN in %) for the entire data set, spring and summer cruises (sp/su), data with salinity > 1, spring and summer cruises with salinity > 1, data with salinity < 1 and spring and summer cruises with salinity < 1. The significance is shown as ** for p-value < 0.001, * for p-values < 0.01 and + for p-values < 0.05.**

| $N_2O$ saturation % | Q $m^3s^{-1}$ | T °C | pH | $O_2$ % | $NH_4^+$ µM | $NO_2^-$ µM | $NO_3^-$ µM | SPM mg | C/N | PC % | PN % |
|---|---|---|---|---|---|---|---|---|---|---|---|
| Entire data | 0.13 | 0.06 | -0.65 | -0.39 | 0.02 | 0.48 | 0.27 | -0.31 | 0.53 | 0.12 | -0.16 |
| sp/su | -0.26 | 0.76+ | -0.82+ | -0.32 | 0.01 | 0.35 | -0.40 | -0.92* | 0.15 | 0.18 | 0.31 |
| Sal>1 | -0.07 | -0.14 | -0.38 | -0.43 | -0.18 | 0.23 | 0.52 | -0.19 | 0.46 | -0.18 | -0.38 |
| Sal<1, | -0.21 | 0.29 | -0.59 | -0.39 | 0.26 | 0.76* | -0.11 | -0.57 | 0.12 | 0.61 | 0.47 |
| Sal>1, sp/su | -0.07 | -0.70+ | -0.41 | -0.26 | -0.42 | 0.03 | 0.05 | -0.81+ | -0.04 | -0.10 | 0.14 |
| Sal<1, sp/su | -0.48 | 0.72+ | -0.80 | -0.46 | 0.29 | 0.77+ | -0.58 | -0.87+ | -0.17 | 0.69 | 0.67 |

## 4    Discussion

### 4.1    $N_2O$ saturation and flux densities of the Elbe Estuary

The average $N_2O$ saturation and flux density were 197 % and $39.9 \pm 46.9$ µmol m$^{-2}$ d$^{-1}$, respectively. The $N_2O$ flux densities from the Elbe Estuary were in the mid-range of flux densities of other European estuaries ranging from 2.9 µmol m$^{-2}$ d$^{-1}$ to 96.5 µmol m$^{-2}$ d$^{-1}$ (Garnier et al., 2006; Gonçalves et al., 2010; Murray et al., 2015) and average $N_2O$ saturations fitted to values determined by Reading et al. (2020) for highly modified urban systems. The relationship of $N_2O_{xs}$ and AOU (Fig. 3), with changing slopes in the Port of Hamburg and mesohaline estuary, was determined by either initial riverine $N_2O$ production, or in-situ production along the estuary. During spring and summer, we found increasing $N_2O$ concentrations in the Hamburg Port region (see also Brase et al. (2017)), and in the salinity gradient (stream kilometer 680 – 700, salinity ~5). Both $N_2O$ peaks varied in magnitude and spatial extension, suggesting in-situ biological production (Fig. 2g). This matches earlier research linking estuarine $N_2O$ fluxes to in-situ generation (e.g. Bange, 2006; Barnes and Upstill-Goddard, 2011; Murray et al., 2015).

Previous measurements of $N_2O$ saturation and flux densities in the Elbe Estuary between the 1980s and 2015 (Hanke and Knauth, 1990; Barnes and Upstill-Goddard, 2011; Brase et al., 2017) showed a significant reduction of $N_2O$ saturation due to the reduced riverine nutrient load and higher dissolved oxygen concentrations (Brase et al., 2017). However, since the BIOGEST study in 1997 (Barnes and Upstill-Goddard, 2011), $N_2O$ remained relatively stable at ~ 200 % saturation despite a concurrent decrease in TN concentration from ~400 µmol L$^{-1}$ to around 200 µmol L$^{-1}$ (Fig. S3; Hanke and Knauth, 1990; Barnes and Upstill-Goddard, 2011; Brase et al., 2017; Das Fachinfomrationssystem (FIS) der FGG Elbe, 2022). Since $N_2O$ saturation did not decrease in scale with riverine nitrogen input, this suggests that the yield of $N_2O$ production increased along the estuary. Dähnke et al. (2008) showed a shift from dominating denitrification towards significant nitrification in the Elbe Estuary due to the significant improvement of water quality after the reunification of Germany in 1990, and this could influence $N_2O$ distributions in the estuary. In the following sections, we investigate the biogeochemical controls of this in-situ $N_2O$ production. For this purpose, we discuss the two zones of intense $N_2O$ production separately and also distinguish between cruises in spring and summer (water temperature > 10 °C) and in winter (water temperature < 6 °C).

### 4.2    $N_2O$ production in spring and summer in the mesohaline estuary

The $N_2O$ peak in the transition between oligohaline and mesohaline estuary was accompanied by a sudden change in the slope of the AOU vs $N_2O_{xs}$ plots, (Fig. 3), pointing towards $N_2O$ production in the oxic water column. Peaks of nitrite and ammonium concentrations coincided with the elevated nitrous oxide saturations between Elbe km 680-700, with an ammonium peak around stream kilometer ~720, and a nitrite peak at ~700 (Fig. 4a). Highest $N_2O$ concentrations were usually measured between the nitrite peak and the region with highest turbidity (Fig. 4a, September 2020, and Fig. S4-S14). This co-occurrence of nitrite accumulation and increased $N_2O$ saturation has been interpreted as signs for $N_2O$ production via denitrification (e.g. Wertz et al., 2018; Sharma et al., 2022). However, denitrification does not seem likely in this oxic water column. Such a succession of nitrite and ammonium peaks is also typical for remineralization and nitrification, and the slight decrease of oxygen concentrations around the higher $N_2O$ saturation (Fig. 2g and i) suggests oxygen consumption, possibly caused by these two processes. Sanders et al. (2018) measured small but detectable nitrification rates ($1 - 2$ µmol L$^{-1}$ d$^{-1}$) for this region of the Elbe Estuary, suggesting that $N_2O$ may be a side product of nitrification.

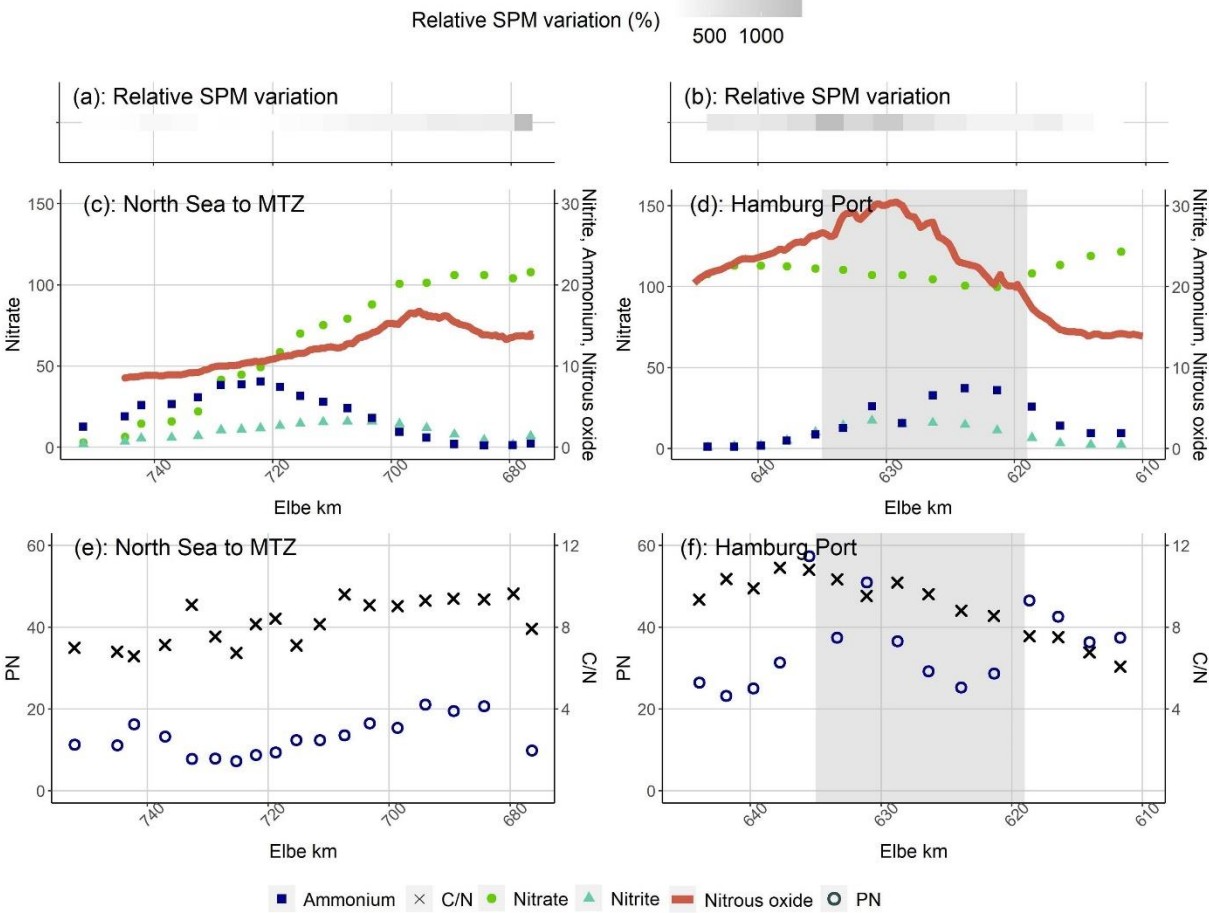

305

**Figure 4: Succession of N-bearing substances coming from the North Sea and in the Port of Hamburg in September 2020: Relative change of SPM concentrations (a) from the North Sea and (b) in the Port of Hamburg. Nitrate in $\mu mol\ L^{-1}$, nitrite in $\mu mol\ L^{-1}$, ammonium in $\mu mol\ L^{-1}$ and nitrous oxide concentrations in $nmol\ L^{-1}$ plotted against Elbe stream kilometers (c) from the North Sea and (d) in the Port of Hamburg. Particulate nitrogen concentrations in $\mu mol\ L^{-1}$ and C/N values plotted against stream kilometers (e) from the North Sea and (f) in the Port of Hamburg. The grey area in (d) and (f) shows the position of the Port of Hamburg.**

This succession of N-bearing substances (Fig. 4, Fig. S4-S14) suggests input of particulate matter from the North Sea and upstream particle transport towards the maximum turbidity zone of the estuary (MTZ). This transport mechanism is in line with Wolfstein and Kies (1999), who explained organic matter contents and chlorophyll a concentrations in the polyhaline part of the Elbe Estuary by input of freshly produced particulate matter of marine origin. Generally, maximum turbidity zones are generated by the balance between river-induced flushing and upstream transport of marine SPM, as a function of estuarine geomorphology, gravitational circulation and tidal flow, trapping the particles in the MTZ (Bianchi, 2007; Sommerfield and Wong, 2011; Winterwerp and Wang, 2013). Other studies detected $N_2O$ production from water column nitrification in estuarine MTZs (e.g. Barnes and Owens, 1999; de Wilde and de Bie, 2000; Bange, 2006; Barnes and Upstill-Goddard, 2011; Harley et al., 2015), caused by high bacterial numbers, particulate nitrogen availability and long residence times (Murray et al., 2015). For the selected dataset, we calculated a negative correlation between average SPM concentrations and $N_2O$ saturation (R = -0.81, Table 5), and found that the $N_2O$ peak was located downstream of the MTZ, and upstream of increasing nitrite and ammonium concentrations (Fig. 4a). This suggests that (1) the mere concentration of SPM is not the driving factor of nitrification as a source of $N_2O$, but that organic matter quality is key to biological turnover (Dähnke et al. 2022), and (2) the material transport from the North Sea upstream towards the MTZ

(Kappenberg and Fanger, 2007; Schoer, 1990) is a main mechanism for $N_2O$ generation. We find organic matter
with low C/N ratios, and with relatively high PN and PC contents in the outermost samples (ranging from 5.9 in
June 2020 to 8.8 August 2017), indicating fresh and easily degradable organic matter (Fig. S1, e.g. Redfield et al.
1963; Fraga et al. 1998; Middelburg and Herman 2007). Towards the MTZ, C/N values, PN and PC contents
decreased, indicating remineralization in the water column. This remineralization and subsequent nitrification can
then cause the observed succession of ammonium, nitrite and $N_2O$ peaks (Fig. 4a), contributing to the high nitrate
concentrations in the MTZ, where high C/N values (9 – 11/16) indicate low organic matter quality (e.g. Hedges
and Keil 1995; Middelburg and Herman 2007). Overall, we conclude that remineralization of marine organic
matter, followed by nitrification, produced the $N_2O$ peak in the salinity gradient of the Elbe Estuary. This
production was mainly fueled by fresh organic matter entering the estuary from the North Sea.
**4.3    Hamburg Port: $N_2O$ production in spring and summer**
During all cruises, we measured highest $N_2O$ saturation in the Port of Hamburg. These peaks can be caused by
input from a waste water treatment plant, by deepening and dredging operations, enhanced benthic production or
by in-situ production in the water column.
Point sources generally play a minor role in the Elbe Estuary (Hofmann et al., 2005; IKSE, 2018). We estimated
the wastewater discharge fraction of stream flow according to Büttner et al. (2020) for the waste water treatment
plant (WWTP) Köhlbrandhöft, which treats the waste water from the Hamburg metropolitan region, with less than
5 % even under low fresh water inflow. Thus, point sources seemed not to be the cause for the elevated $N_2O$
concentrations. However, discharge of WWTPs can potentially be important sources of $N_2O$ (Beaulieu et al., 2010;
Chun et al., 2020; Brown et al., 2022), and the effect of wastewater input on $N_2O$ concentrations and emissions
may change with altered river discharge, water temperature and riverine nitrogen loads in the future.
Dredging can be a potential source of $N_2O$ in the water column. The estuary is continuously deepened and dredged
to grant access for large container ships, which stirs up bottom sediments. Ammonium concentrations in the
sediment pore water are high (Zander et al., 2020, 2022) and $N_2O$ can be produced by nitrifier-denitrification in
the sediments (Deek et al., 2013). However, we found no correlation of high SPM concentrations and $N_2O$
saturation, indicating no major influence on $N_2O$ dynamics from channel dredging and deepening.
Several studies identified the Hamburg Port region as a hotspot of biogeochemical turnover: Deek et al. (2013)
showed denitrification, where Sanders et al. (2018) measured intense nitrification.. Norbisrath et al. (2022)
determined intense total alkalinity generation, and Dähnke et al. (2022) found that nitrogen turnover was driven
by high particulate organic matter in this region. Brase et al. (2017) identified the Hamburg port region as a hotspot
of $N_2O$ production and hypothesized that simultaneous nitrification and sediment denitrification were responsible.
We use our expanded dataset to further evaluate this hypothesis and to identify drivers for $N_2O$ production in the
port region.
During all cruises in spring and summer, we measured ammonium and nitrite peaks in the Hamburg Port region
(Fig. 2c and 2e, exemplary for September 2020 in Fig. 4b). Several researchers did address the nitrogen turnover
and this accumulation of nitrite and ammonium assuming that the sudden increase of water depth in the Port leads
to a light limitation and decomposition of riverine organic material (Schroeder, 1997; Schöl et al., 2014). This in
turn raises ammonium and nitrite concentrations and fosters nitrification in the port region (Sanders et al., 2018;
Dähnke et al., 2022).
High nitrite concentrations are favorable for $N_2O$ production by nitrifier-denitrification (Quick et al., 2019), while
low-oxygen conditions facilitate $N_2O$ production from both nitrification and denitrification. We found that $N_2O$
saturation increased with decreasing discharge (R = -0.48, Table 5) during spring and summer. This further points
towards in-situ $N_2O$ production, because longer residence times lead to a possible accumulation of $N_2O$ from
either nitrification or denitrification (e.g. Nixon et al. 1996; Pind et al. 1997; Silvennoinen et al. 2007; Gonçalves
et al. 2010). Overall, our data showed the succession of ammonium, nitrite and $N_2O$ production (Fig. 4b and
supplementary material S4-S14) as well as a breakup of the linear relation between AOU and $N_2O_{xs}$ in the Port
region (Fig. 3). In combination with previous nitrogen process studies performed in the Elbe Estuary (Deek et al.,
2013; Sanders et al., 2018; Dähnke et al., 2022), this supports simultaneous sedimentary denitrification and
nitrification in the water column as responsible pathways for $N_2O$ production in the Port of Hamburg (Brase et al.

376    2017).

In spring and summer, we found no linear relationship between $N_2O_{xs}$ and AOU in the Hamburg Port (Fig. 3). This
may result from combined $N_2O$ production by nitrification and denitrification. However, oxygen saturation and
$N_2O$ saturation were inversely correlated in Hamburg Port (Table 4 and 5), suggesting that $N_2O$ production was
controlled by oxygen concentrations, and thus was related to oxygen consumption in the port region. Most (75 %)
of this oxygen consumption is caused by respiration whereas the remaining 25 % stem from nitrification (Schöl et
al., 2014; Sanders et al., 2018). This respiration in turn is determined by remineralization of algal material from
the upstream river that is transported to and respired within the port region (Schroeder, 1997; Kerner, 2000; Schöl
et al., 2014), linking estuarine $N_2O$ production to river eutrophication. Fabisik et al. (2023) showed that algae could
additionally contribute to $N_2O$ production. In the Elbe, fresh organic matter from the river with low C/N values as
well as high PN and PC contents entered the estuary. This organic material was rapidly degraded in the Hamburg
Port region (Fig. S1). Dähnke et al. (2022) found that labile organic matter fueled nitrification but also
denitrification in the fresh water part of the Elbe Estuary, which, as shown in our study, results in high $N_2O$
production in the Hamburg Port, leading to the reported negative correlations of PC and PN content with $N_2O$
saturation.
Overall, oxygen conditions mainly controlled $N_2O$ production in the Hamburg Port region in spring and summer.
Since respiration of organic matter dominates oxygen drawdown in the port region, we deduce that $N_2O$ production
there is linked to the decomposition of phytoplankton produced in the upstream Elbe River regions.
**4.4    Hamburg Port: $N_2O$ production in winter**
In winter, low water temperature (< 6 °C) should hamper biological production (Koch et al., 1992; Halling-
Sorensen and Jorgensen, 1993). Indeed, we did not detect a $N_2O$ peak in the MTZ in winter, but we find high $N_2O$
concentrations in the port region. For March 2022, we found a linear increase of $N_2O_{xs}$ and AOU along with
oxygen consumption and increasing ammonium, nitrite and PN concentrations indicating nitrification in the
Hamburg Port producing $N_2O$. Unlike in summer, $N_2O$ concentrations showed a flat increase extending far into
the oligohaline section of the estuary (Fig. 2, Fig. S1).
However, in March 2021, we found a sharp and sudden increase in $N_2O$, with a peak concentration that by far
exceeded internal biological sources in summer (Fig. 2h). An ammonium peak in the water column coincided with
the $N_2O$ maximum (Fig. 2f and Fig. S12). If microbial activity is mostly temperature-inhibited, a local source of
$N_2O$ in the port seems the most likely cause.
We considered intensified deepening operations in the Port of Hamburg as one potential source of elevated $N_2O$
saturation. Deepening and dredging work occurred in the Hamburg Port region in 2021 (HPA, pers. Comm.,
Karrasch 2022), but, this also applied to 2022, when we saw no sharp $N_2O$ peak (Fig. 2h). Furthermore, the regions
of deepening and dredging did not match the region of high $N_2O$ concentrations, and turbidity at the time of
sampling did not change significantly compared to other cruises. Jointly, this suggests that channel dredging and
deepening was not the primary cause for the 2021 winter $N_2O$ peak.
Another possible source of $N_2O$ is the WWTP outflow in the Southern Elbe that joins the main estuary at stream
kilometer 626 (Fig. 1), matching the $N_2O$ peak at stream kilometer 627 (Fig. 2h). As explained above (section 4.3),
the effect of this WWTP on $N_2O$ saturations under normal conditions should be negligible. This peak can be the
result of an extraordinary event during our sampling. We indeed found that an extreme rain event occurred on
March 11[th] 2021 (HAMBURG WASSER, pers. Comm., Laurich 2022) with a statistical recurrence probability of
one to five years (https://sri.hamburgwasser.de/, last access: 04.04.2023). This rare event caused a temperature
drop in the WWTP due to high inflows of cold rainwater leading to aggravated operation conditions at the time of
sampling. While the operators could still meet the limits for the effluent levels of nitrate and ammonium, higher
than usual ammonium loads exited the treatment plant at this time. We hypothesize that these elevated ammonium
WWTP loads were rapidly converted to $N_2O$ as the warmer and biologically active waste water entered the Elbe
Estuary in March 2021. An important factor for aggravated conditions was a temperature drop in the WWTP
caused by cold rain water (HAMBURG WASSER, pers. Comm., Laurich 2022), we therefore hypothesize that a
similar rain event in warmer months would not have the same effect.
Therefore, we argue that our March 2021 cruise likely represents an exception due to an extreme weather situation,
whereas normal winter conditions in the estuary comply with the $N_2O$ production, like in March 2022.
**4.5    Seasonally varying $N_2O$:DIN dynamic**
We calculated annual $N_2O$ emissions of the Elbe Estuary ranging from $0.08 \pm 0.03$ Gg-$N_2O$ yr$^{-1}$ to
$0.25 \pm 0.16$ Gg-$N_2O$ yr$^{-1}$, which varied from recent $N_2O$ summer emission estimate of $0.18 \pm 0.01$ Gg-$N_2O$ yr$^{-1}$ by
Brase et al. (2017). Estuarine $N_2O$ emissions are affected by tides, diel variations and currents (Barnes et al., 2006;
Baulch et al., 2012; Gonçalves et al., 2015), all of which we did not address in our study. Range of possible
parametrizations of gas transfer coefficients further complicates a direct comparison of fluxes between studies
(Hall Jr. and Ulseth, 2020; Rosentreter et al., 2021), which was reflected in the big differences of our emission
estimates (Table 2). Therefore, a direct comparison to other studies is difficult.
In a more general approach, the relationship between $N_2O$ and DIN ($N_2O$:DIN) is used for global estimates of $N_2O$
emissions (Kroeze et al., 2005, 2010; Ivens et al., 2011; Hu et al., 2016). Using publicly available data (Table S4
and S5), we calculated the amount of the annual nitrogen load released as $N_2O$. Depending on the parametrization
used for the gas transfer coefficients, 0.14 % to 0.67 % of the annual DIN loads of the Elbe Estuary were released
as $N_2O$ (0.11 % to 0.57 % for TN loads). This is significantly less than the 1 % predicted by Kroeze et al. (2005),
but matches results from other estuaries with high agricultural input, e.g. Wells et al. (2018) with 0.3 % to 0.7 %
(0.1 % for TN loads) and Robinson et al. (1998) with 0.5 % (0.3 % for TN loads) as well as the 0.11 % to 0.37 %
estimated by Maavara et al. (2019), who used TN loads to predict global estuarine emissions. In general, $N_2O$:DIN
ratios vary widely (e.g., Baulch et al., 2012; Maavara et al., 2019; Smith and Böhlke, 2019). Wells et al. (2018)
even found a range from -25 % to 7 % of DIN was emitted as $N_2O$ in estuaries with low land-use intensity. At our
site, highest emissions were estimated in winter (Fig. 5b) along with highest DIN loads (Fig. 5c). In spring, summer
and late summer, N₂O emissions reduced along with DIN loads (Fig. 5b, c). However, N₂O release did not scale
with the seasonal change of DIN. In winter, 0.10 % to 0.32 % of DIN were released as N₂O, whereas during the
other seasons, up to 1.26 % were emitted. Thus, our results corroborate that there is a varying relationship between
DIN and N₂O (Borges et al., 2015; Marzadri et al., 2017; Wells et al., 2018) showing that this relationship even
varies seasonally on site due to changing drivers for N₂O production and emissions, e.g., temperature (Murray et
al., 2015; Quick et al., 2019) and oxygen levels (de Bie et al., 2002; Rosamond et al., 2012; Yevenes et al., 2017).
Next to DIN loads, we find that organic matter is an important driver for N₂O production by providing substrate
for nitrification. Furthermore, the comparison of results with previous measurements in the Elbe Estuary revealed
that N₂O saturation stopped to scale with DIN input after the 1990s (section 4.1). The significant regime change
after the 1990s enabled phytoplankton growth to reestablish in the river that had previously been inhibited by high
pollutant levels and low light availability (Kerner, 2000; Amann et al., 2012; Hillebrand et al., 2018; Rewrie et al.,
2023). The prevailing high nitrification rates in the estuary (Dähnke et al., 2008; Sanders et al., 2018) support an
overarching control of organic matter on N₂O production and emissions along the Elbe Estuary.

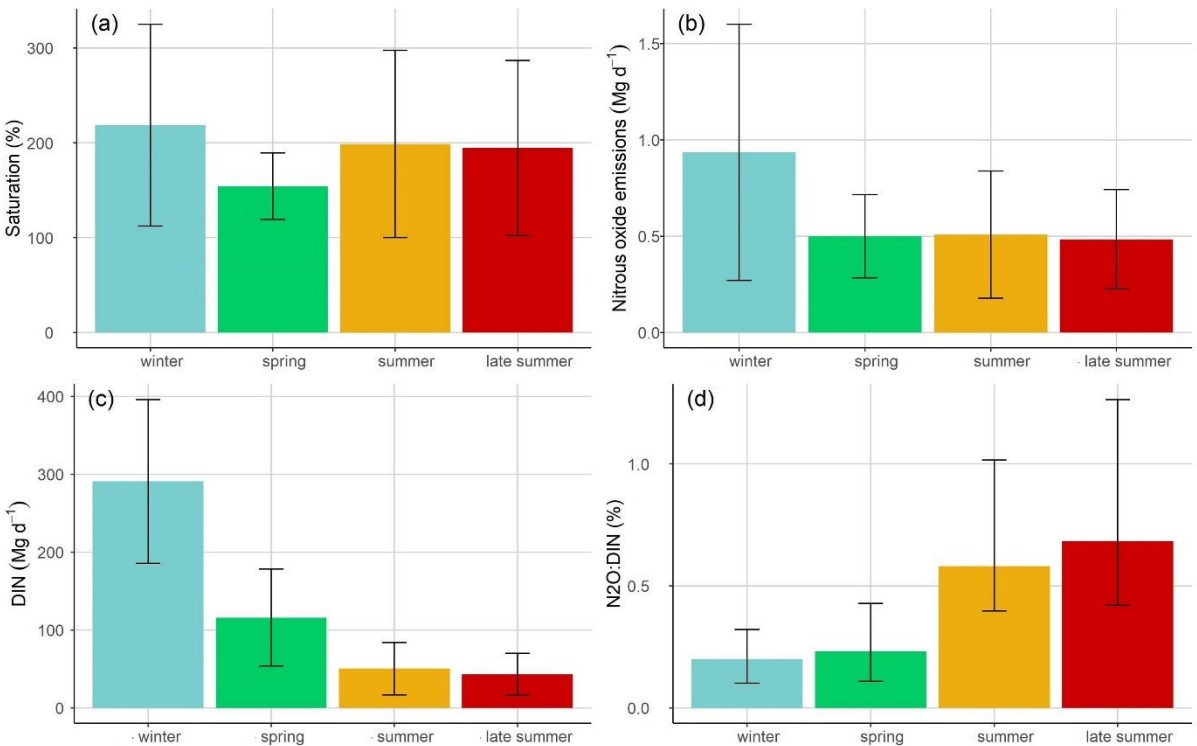


**Figure 5: (a) Average nitrous oxide saturation for each season, (b) average nitrous oxide emissions for each season**
**calculated after Borges et al. (2004), (c) average DIN loads for each season and (d) ratio of nitrous oxide emissions and**
**DIN loads (N₂O:DIN) for each season. The error bars represent the standard deviations for (a), (b) and (c). The**
**N₂O:DIN ratios is shown as average values calculated for each parametrization and wind speeds with error bars**
**representing their variability.**
**5    Conclusions**
Overall, the Elbe is a year-round source of N₂O to the atmosphere, with highest emissions occurring in winter,
along with high DIN loads and high wind speeds. However, summer N₂O saturation and emissions did not decrease
with lower riverine nitrogen input, suggesting variable relations of DIN and N₂O (Borges et al., 2004; Marzadri et
al., 2017; Wells et al., 2018), and seasonal variability of this ratio caused by changing drivers for N₂O production
and emissions. Two hot-spots of N₂O production were found in the Elbe Estuary: the Port of Hamburg and the
mesohaline estuary near the estuarine turbidity maximum. Biological N₂O production was fueled by riverine
organic matter in the Hamburg Port or marine organic matter in the MTZ. A comparison with historical $N_2O$
measurements in the Elbe Estuary revealed that $N_2O$ saturation did not decrease with DIN input after the 1990s.
The improvement of water quality in the Elbe Estuary allowed phytoplankton growth after the reunification of
Germany in 1990s (Kerner, 2000; Amann et al., 2012; Hillebrand et al., 2018; Rewrie et al., 2023) and led to a
switch from dominant denitrification to high nitrification (Dähnke et al., 2008; Sanders et al., 2018), supporting
the overarching control of organic matter on $N_2O$ production along the Elbe Estuary. Thus, our findings indicate
that DIN availability is not the sole control of $N_2O$ production in estuaries with high agricultural input.
High organic matter availability due to phytoplankton blooms driven by river eutrophication fuels nitrification and
subsequent $N_2O$ emissions, causing a decoupling of the $N_2O$:DIN ratio. Therefore, $N_2O$ emissions in heavily
managed estuaries with high agricultural loads are clearly linked to eutrophication. A reduced nitrogen input would
reduce phytoplankton growth and thus also $N_2O$ emissions. However, the development of phytoplankton blooms
is not solely controlled by nutrient inputs, but also by e.g., temperature, residence time, water depth and grazing.
Thus, complex biological and chemical processes control phytoplankton dynamics (Scharfe et al., 2009; Dijkstra
et al., 2019; Kamjunke et al., 2021), which will change significantly in the future due to the effects of climate
change (IPCC, 2022). A holistic approach to water quality mitigation and climate change adaptation is needed to
prevent high $N_2O$ emissions.

**Data availability**

The dataset generated and/or analyzed in this study are currently available upon request from the corresponding
author and will be made publicly available under coastMap Geoportal ([www.coastmap.org](www.coastmap.org)) connecting to
PANGAEA. ([https://www.pangaea.de/](https://www.pangaea.de/)) with DOI availability in the near future.

**Authors contribution**

GS, TS and KD designed this study. GS did the sampling and measurements for cruises from 2020 to 2022 as well
as the data interpretation and evaluation. TS was responsible for the sampling and measurements for cruises done
in 2017 and 2019. YGV provided the oxygen data correction from the FerryBox data. KD, HWB, YGV and TS
contributed with scientific and editorial recommendations. GS prepared the manuscript with contributions of all
co-authors.

**Competing interest**

The authors declare that they have no conflict of interest.

**Acknowledgments**

This study was funded by the Deutsche Forschungsgemeinschaft (DFG, German Research Foundation) under
Germany's Excellence Strategy – EXC 2037 "CLICCS - Climate, Climatic Change, and Society" – Project
Number: 390683824, contribution to the Center for Earth System Research and Sustainability (CEN) of Universität
Hamburg. Parts of the study were done in the framework of the cross-topic activity MOSES (Modular Observation
Solutions for Earth Systems) within the Helmholtz program Changing Earth (Topic 4.1). We thank the crew of

R/V Ludwig Prandtl for the great support during the cruises. Thanks to Leon Schmidt and the entire working group "Aquatic nutrients cycles" for measuring nutrients and the support during the campaigns. We are thankful for the Hereon FerryBox Team for providing the FerryBox data. Thanks to the working group of Biogeochemistry at the Institute for Geology of the University Hamburg for measuring C/N ratios, PC and PN fractions. We thank Frank Laurich (HAMBURG WASSER) and Dr. Maja Karrasch (Hamburg Port Authority) for their interest in our $N_2O$ measurements and their willingness to provide information. Thanks to Victoria Oritz (Federal Waterways and Engineering and Research Insitute) for providing the respective areas of the Elbe Estuary. Thanks to the NOAA ESRL GML CCGG Group for providing high quality, readily accessible atmospheric $N_2O$ data.

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
