# Peer review of "Seasonal variability of nitrous oxide concentrations and emissions in a temperate estuary"

_Biogeosciences, 2023_

## Referee Comment (RC1)

Schulz et al., 2023 conducted underway surface measurements of N2O concentration in the Elbe estuary on 9 cruises across spring, summer and winter. They showed a large spatial variation of N2O concentration and flux, and identified two hotspots of N2O production including the Port of Hamburg and region near the estuarine turbidity maximum. They argued that there is not much seasonal variation in N2O flux because high riverine N2O concentration in winter may compensate for its low biological N2O production compared to summer. This manuscript presents a valuable dataset of N2O concentration and flux from a European estuary. Especially, the seasonal pattern of estuarine N2O flux has been poorly observed and understood. However, there are some points needed to be considered/corrected such as the interpretation of the relationship between excess N2O and AOU, evaluation of environmental controls on N2O concentration/flux, calculation of annual N2O flux and uncertainty estimate. See detailed comments below.

Line 17-18: what do you mean by "compensated the effect of decreasing dissolved inorganic nitrogen (DIN) loads"?

Line 25: How does 0.24±0.06 Gg N2O y-1 emission compare to global estuarine N2O emission?

Lines 40-42: Denitrification could also occur in anoxic water column contributing to N2O production (Ji et al., 2018; Tang et al., 2022).

Line 44: specify Port Hamburg as the third largest port in Europe.

Line 70: how deep is the Elbe estuary? This gives an idea if sedimentary processes (e.g., N2O production) may affect N2O concentration in the surface water column.

Figure 1: There are too many city names on the map, which is distractive. It may be clearer to label only the key cities like Cuxhaven or island Scharhorn where the Elbe River enters the North Sea or Oortkaten.

Lines 85-87: Why transect sampling was performed after high tides? What's the effect of tides on N2O concentration? Tidal cycles of N2O concentration have been observed in other estuaries (Goncalves et al., 2015; Barnes et al., 2006).

Line 116 in Equation 1: is N2Ocw the partial pressure of N2O in water? Otherwise, the saturation should be calculated as the $\frac{N2Ocw}{N2Oeq}*100$ where N2Oeq is the equilibrated N2O concentration with atmosphere. Similarly in Equation 3. N2Oair should be N2Oeq.

Line 143: Why nitrate concentration increased at 700 km? Are there tributaries or point sources?

Lines 148-149: Why ammonium and nitrite concentration increased near Hamburg Port? Is it due to internal organic matter remineralization or point sources or sedimentary flux?

Figure 2: It is hard to tell the difference among each cruise with so many colored lines. How about presenting data from the same season using the same color to illustrate the seasonality as a supplementary figure?

Lines 211-218 and lines 232-234: Figure 4 a and b are both from June, summer. The linear positive relationship between AOU and excess N2O suggests N2O production from nitrification (e.g., Nevison et al., 2003). The increase in the slope should be interpreted as an increase in the N2O production yield or external N2O input (e.g., point source).

Figure 4: It would be interesting to systematically/statistically assess the relations between excess N2O and environmental factors like salinity (non-conservative behavior of N2O) or dissolved inorganic nitrogen (infer N2O production pathways), PN, PC, and SPM. There seems to be a good relation between N2O and ammonium/nitrite concentration shown in Figure 2.

Lines 242-243 and Figure 5: What about the variations of the N2O%, oxygen and total nitrogen concentration? The riverine N concentration is decreasing, what about the changes in other point sources of N input along the estuary (e.g., from wastewater treatment plants) or concentration in the estuary?

Line 272: "this suggests"

Line 273: how is MTZ defined? What threshold of suspended particle material is used to define the MTZ?

Line 287 and 296-297: clarify the reference: Kappenberg and Fanger, 2007 (German?) and source of organic matter from the North Sea into the Elbe estuary.

Lines 311-313: How about showing the relations between ammonium, nitrite and N2O in figures?

Line 315: What are R values? R is positive for nitrite concentration.

Line 320-321: Is nitrification responsible for the remaining oxygen consumption?

Line 326 and Figure S1: why C/N ratio was so high in 2021 March?

Line 345-347: "Ammonium and N2O concentrations are high in the pore water of underlying sediments". Reference or example of the concentration. What about the timing of deepening and dredging works in the Hamburg Port compared to the cruise periods?

Lines 360-361: Has there been any N2O measurement from this wastewater treatment plant (WWTP) Köhlbrandhöft? The ammonium concentration in 2021/03 is not exceptionally high compared to previous cruises (e.g., 2020/06). What about the direct N2O output from the wastewater treatment plant?

Figure 7. Use month or season as the x axis instead of cruise number? Add description of the boxplot. Why not adding error bars for emissions?

Table 3. How is annual N2O emission calculated? Since there is a seasonal variation in the N2O flux, monthly or seasonal N2O emission may be more representative. Because N2O flux was measured at a high spatial resolution, it may be useful to calculate the N2O flux across the whole estuary by integrating the flux and area section by section (e.g., River section, Hamburg port, Oligohaline section) instead of multiplying the average N2O flux by the whole area of Elbe estuary.

Line 406: Why do you think there is no seasonality in N2O emission? N2O flux is different comparing spring, summer and winter shown in Table 3.

References:

Barnes, J., Ramesh, R., Purvaja, R., Nirmal Rajkumar, A., Senthil Kumar, B., Krithika, K., ... & Upstill-Goddard, R. (2006). Tidal dynamics and rainfall control N2O and CH4 emissions from a pristine mangrove creek. *Geophysical Research Letters*, *33*(15).

Gonçalves, C., Brogueira, M. J., & Nogueira, M. (2015). Tidal and spatial variability of nitrous oxide (N2O) in Sado estuary (Portugal). *Estuarine, Coastal and Shelf Science*, *167*, 466-474.

Ji, Q., Frey, C., Sun, X., Jackson, M., Lee, Y. S., Jayakumar, A., ... & Ward, B. B. (2018). Nitrogen and oxygen availabilities control water column nitrous oxide production during seasonal anoxia in the Chesapeake Bay. *Biogeosciences*, *15*(20), 6127-6138.

Nevison, C., Butler, J. H., & Elkins, J. W. (2003). Global distribution of N2O and the ΔN2O-AOU yield in the subsurface ocean. *Global Biogeochemical Cycles*, *17*(4).

Tang, W., Tracey, J. C., Carroll, J., Wallace, E., Lee, J. A., Nathan, L., ... & Ward, B. B. (2022). Nitrous oxide production in the Chesapeake Bay. *Limnology and Oceanography*, *67*(9), 2101-2116.

---

## Referee Comment (RC2)

**Review of 'Seasonal variability of nitrous oxide concentrations and emissions along the Elbe estuary' 12/03/2023**

This work presents five years (two years of new data) of seasonal (winter v summer) data on dissolved N2O concentrations in the Elbe River Estuary. The insights this can provide into interannual variations in aquatic N2O is relatively unique. The site of the study itself, which encompasses a large industrial port, is also important in terms of better understanding anthropogenic impacts on aquatic N2O emissions. The authors show that, even though the source of N2O seems to be strongly seasonal, emissions remain relatively consistent over the year. This is a new and interesting finding. Overall this is a nice study with the potential to be a useful contribution to both the journal and scientific understanding of aquatic N2O.

However, there are a few weaknesses with the data analysis and discussion that need to be addressed to ensure that the emissions are accurately represented and the findings are clearly conveyed.

**Data analysis:**

1. Converting dissolved concentrations to emissions: Like many studies, here the authors measured the dissolved concentration of the gas (N2O), and then converted this into water-air emissions based on a gas transfer velocity (k). Gas transfer velocities can be highly variable, especially in estuaries where the importance (and magnitude) of factors like wind, flow velocity, and water depth can all vary a lot over space and time. This complexity is reflected in the wide range of empirical k value parameterisations that have been developed for estuaries (see e.g., Rosentreter et al. (2021), also Hall and Ulseth (2019) for a good review of the topic, albeit for freshwater systems). However, here the authors convert measured concentrations to emissions using a single parameterisation (L116-125). This creates considerable uncertainty, which is not reflected in the reported estuary emissions estimates. Emissions should be recalculated using 3-5 k parameterisations, and the variability of these outputs reported in the results / figures. More information should also be supplied on the wind speed data used in the parameterisations. It is important to understand how the values measured during the campaigns compare to 'average' conditions around the estuary when considering the upscaled seasonal emissions values (e.g., are emissions estimates likely to be on the low side because cruises were only done on low-wind days?).

2. Relationship between N2O and N inputs: As discussed in the paper intro here, aquatic N2O emissions are generally predicted based on N loads to the system (i.e., leaching of N, inputs from WWTPs, etc). While here N2O emissions are discussed and presented, the N inputs side of the equation is not clear to me. In the site description it says that annual N load were ~80 Gg y-1 (L67) – but does this mean the estuary *receives* this much N, or discharges this much N? And how does this break down between sources (WWTPs v river discharge)? On L231 it says that N2O emissions were low relative to other high N input estuaries. But how do N inputs into the Elbe stack up compare to these other estuaries? I particularly wonder how the 'point source' N loads around the port might stack up with those in other urban estuaries where N2O emissions have been measured, e.g., (Wells et al., 2018). Constraining the other side of the N2O emissions v N inputs equations is critical for placing these findings into a more global context. Within the study, more information on N loads will also be important for picking apart the seasonal emissions drivers. How much N enters the estuary at the port? Is this input seasonally variable? Did it vary between the sampled years? Do these variations correspond with variations in emissions (particularly the size of the winter N2O-excess excursion)?

**Paper structure:**

1. Introduction: It is not entirely clear how studying N2O in the Elbe estuary will advance understanding of aquatic N2O emissions / fill a needed research gap. A stronger transition between the penultimate and last paragraphs of the discussion is needed (how does the present study relate to the broader literature). Stating a testable hypothesis, rather than just site-specific study objectives, in the last paragraph may also help make the study more clearly relevant to the broader scientific community. Is this just a case study or will the data help us understand estuary N cycling and gaseous emissions in a more fundamental way?

2. Discussion: While I think overall the data interpretation makes sense, the discussion section currently reads as a bit descriptive and could go further to place these findings in a broader context (rather than just the context of how we understand the Elbe River Estuary). This could include in particular more discussion of N cycling in urban estuaries / where there are point N pollution. Where else in the world would the observed seasonal patterns be expected to be found? I also think there is missing some discussion of 'alternative hypotheses' – work through the logic of why denitrification is not thought to be the primary driver of N2O in the estuary, and why benthic production (e.g., (Chen et al., 2022)) is also ruled out. Also please carefully edit to ensure that you are not repeating results in this section.

3. Conclusion: This is currently very focused on untangling what exactly is happening within the Elbe River Estuary, but the implications for broader understanding of aquatic N2O production and emissions are not clear.

Line comments

L17-19: This sentence is not clear (how does N2O 'compensate' for decreasing N loads?), please reword.

L22-24: "In winter, high riverine N2O concentrations led to high N2O emissions from the estuary, whereas in summer, estuarine biological N2O production led to equally high N2O emissions." This is I think getting at a crucial point (that although seasonal magnitude of N2O fluxes did not differ the drivers of these fluxes did), the meaning is not clear. What is the difference between winter 'high N2O concentrations' and summer 'high N2O production'? Reword to be more precise about these differences.

L70: How often is 'on a regular basis'? e.g., weekly, yearly, every three years?

L86: Suggest changing 'steaming upstream' to 'travelling upstream' (steaming sounds a bit antiquated)

L101-104: More information on number of nutrient samples collected per survey, as well as method detection limits and precision, would be useful.

L109: How often was 'regularly'? e.g., before each cruise?

L116: How often, and how, was dry air sampled during each cruise?

L122: The term 'flux densities' is not one I'm familiar with – more common to see something like 'water-air fluxes' or 'evasion'.

L123-125: Please provide some clarification on the upscaling approach used to calculate whole-estuary emissions. From the description it sounds like the mean flux was multiplied by the estuary surface area? Or were these calculations area-weighted, and if so at what resolution?

L127-128: Citation?

L148: Low relative to what?

L163-189: Separating the N2O data into different sections for the different units (molar concentrations, % saturation, water-air fluxes) is confusing as these are all inter-related. For instances, it is hard to make sense of the meaning of the molar concentrations without also considering whether these reflect changes in percent saturation (i.e., changes due to water temperature / salinity v source / production). I suggest integrating these lines of data (and thinking) to provide a clearer picture of estuary N2O patterns.

L204: High relative to what?

L209-218: The AOU v N2O-excess relationship really highlights the importance, and seasonality, of the port for estuary N2O emissions, with distinct peaks in the winter and consumption in the summer. Given that this underpins the discussion around seasonal N2O source switching, I wonder if there is a way to include more than just these 'representative' plots in the main text. For instance, a table with info on AOU v N2O-excess slopes, and min-max range for the port? I think if the port data is excluded something like an ANCOVA could be used to compare shifts in slope relationships.

L256-260: Interesting relationship between NO2- and N2O. This could be connected to previous work, e.g., (Sharma et al., 2022; Smith and Bohlke, 2019; Wertz et al., 2018)

L314-316: This should be in the results section

L318-324: Interesting! I wonder if the algae themselves could also be contributing to the N2O production, e.g., (Fabisik et al., 2023)

L330-332: This makes sense, but is this the only possible explanation for high emissions around the port area? What about wastewater inputs, enhanced benthic production, and/or enhanced groundwater connectivity due to dredging? Some discussion of these points will make this conclusion stronger.

L357-358: How extreme was this rain event, i.e., was it more extreme than any rainfalls over the other five years of sampling? This will help verify the attribution, and also put the pulse into context. It would then be instructive to recalculate the seasonal budget with and without this pulse.

L392: If large riverine loads were the main driver, wouldn't there be a ~continuous decrease in concentration over distance? But instead emissions peak in the port.

Table 2: Standard deviations for the air N2O concentrations would be helpful

Fig. 1: The most important pieces of info in this map (where sampling points are, where the port is, where the MTZ is) don't really stand out. Can you adjust colours, font size, etc to better highlight these key features? A scale bar for the main map would also be helpful.

Fig. 2: I'm not sure that there is much value in showing N2O concentrations (in nM) here – the % saturation information in the subsequent figure is much more effective for showing fluctuations between seasons and over the salinity gradient, given the relatively low concentrations and the impact of both temperature and salinity on N2O solubility. It would also be helpful to have 'summer' and 'winter' headings at the top of the two columns to make the point of difference more immediately obvious.

Fig. 3: A unified y axis scale would be helpful for picking out seasonal differences

Fig. 4: As above, unified axes scales would make differences between sampling dates much clearer.

Fig. 5: Different y axes are needed for the different variables (N2O, O2, TN), if not different plot panels

Fig. 6: I found this to be too many variables on the same plot to make much logical sense out of. I suggest separating into two panels, one for all of the N species (y axis unit is uM N), and then another with two y axes, one for PN and one for C/N.

Fig. 7: It would be helpful to use a different pattern or colour scheme to distinguish the winter v summer cruises.

**References** (included to be helpful, not as required citations)

Chen, J.-J., Wells, N.S., Erler, D.V. and Eyre, B.D. (2022) Land-use intensity increases benthic $N_2O$ emissions across three sub-tropical estuaries. J. Geophys. Res.: Biogeosci. 127, e2022JG006899.
Fabisik, F., Guieysse, B., Procter, J. and Plouviez, M. (2023) Nitrous oxide (N2O) synthesis by the freshwater cyanobacterium Microcystis aeruginosa. Biogeosci. 20, 687-693.
Hall, R.O. and Ulseth, A.J. (2019) Gas exchange in streams and rivers. WIREs Water 0, e1391.
Rosentreter, J.A., Wells, N.S., Ulseth, A.J. and Eyre, B.D. (2021) Divergent gas transfer velocities of $CO_2$, $CH_4$, and $N_2O$ over spatial and temporal gradients in a subtropical estuary. J. Geophys. Res.: Biogeosci. 126, e2021JG006270.
Sharma, N., Flynn, E.D., Catalano, J.G. and Giammar, D.E. (2022) Copper availability governs nitrous oxide accumulation in wetland soils and stream sediments. Geochim. Cosmochim. Acta 327, 96-115.
Smith, R.L. and Bohlke, J.K. (2019) Methane and nitrous oxide temporal and spatial variability in two midwestern USA streams containing high nitrate concentrations. Sci. Total Environ. 685, 574-588.
Wells, N.S., Erler, D.V., Maher, D.T., Rosentreter, J., Hipsey, M.R. and Eyre, B.D. (2018) Estuaries as sources and sinks of $N_2O$ across a land-use gradient in subtropical Australia Global Biogeochemical Cycles 32, 877-894.
Wertz, S., Goyer, C., Burton, D.L., Zebarth, B.J. and Chantigny, M.H. (2018) Processes contributing to nitrite accumulation and concomitant $N_2O$ emissions in frozen soils. Soil Biol. Biochem. 126, 31-39.

---

## Author Comment (AC1)

*Schulz et al., 2023 conducted underway surface measurements of N2O concentration in the Elbe estuary on 9 cruises across spring, summer and winter. They showed a large spatial variation of N2O concentration and flux, and identified two hotspots of N2O production including the Port of Hamburg and region near the estuarine turbidity maximum. They argued that there is not much seasonal variation in N2O flux because high riverine N2O concentration in winter may compensate for its low biological N2O production compared to summer. This manuscript presents a valuable dataset of N2O concentration and flux from a European estuary. Especially, the seasonal pattern of estuarine N2O flux has been poorly observed and understood. However, there are some points needed to be considered/corrected such as the interpretation of the relationship between excess N2O and AOU, evaluation of environmental controls on N2O concentration/flux, calculation of annual N2O flux and uncertainty estimate. See detailed comments below.*

We thank the reviewer for their constructive and helpful comments and suggestions about our paper. Following, we reply to each issue individually, and explain the changes we will make to the revised manuscript to meet the reviewers criticism. Reviewer comments are written in bold italics, our answers are kept in plain font.

**Line 17-18: what do you mean by "compensated the effect of decreasing dissolved inorganic nitrogen (DIN) loads"?**

This statement was supposed to summarize our findings of section 4.1, where we complemented observations done by Brase et al. (2017) with our new data sets: Previous $N_2O$ measurements done in 1980s and 1990s showed a significant reduction of $N_2O$ saturation due to reduced riverine nitrogen loads and higher dissolved oxygen conditions. Water quality in the Elbe estuary improved significantly after the reunification and collapse of East German industries (e.g. Guhr et al., 2000). In the 1980s, high nitrogen loads and low oxygen conditions favored denitrification leading to high $N_2O$ saturations (Hanke and Knauth, 1990). With improving water quality, the importance of denitrification decreased (Dähnke et al., 2008), which probably led to the decrease in $N_2O$ saturations (Brase et al., 2017). However, compared to the study in 1997 (Barnes and Upstill-Goddard, 2011) our measured $N_2O$ saturations did not further decreased despite a continuous reduction of nitrogen loads entering the estuary form the upstream river (e.g. Figure 5, FGG Elbe, 2018; Radach and Pätsch, 2007). These findings suggesting that in-situ $N_2O$ production along the estuary is important and compensates the overall effect of decreasing nitrogen loads in the last decades.

Also considering the comments of reviewer 2, we will revise the manuscript to highlight the relevance of our research for a broader audience. Thus, we will rephrase or remove this finding from the abstract, and will focus on a comparison of our results with other research in heavily managed estuaries, highlighting that we found seasonal varying drivers of $N_2O$ emissions that did not scale with DIN loads and were directly linked to eutrophication phenomena.

**Line 25: How does 0.24±0.06 Gg N2O y-1 emission compare to global estuarine N2O emission?**

As mentioned above we will address the relation to other estuaries and global scale in the abstract of the revised version. Also, as we will change the emission estimation by using several parametrizations to calculate the k value (as suggested by the other reviewer) and use a different upscale approach as suggested by this reviewer in a comment below. Thus, the $N_2O$ emission estimate will probably change.

**Lines 40-42: Denitrification could also occur in anoxic water column contributing to N2O production (Ji et al., 2018; Tang et al., 2022).**

We will list water column denitrification as possible $N_2O$ production pathways in this section of the text.

*Line 44: specify Port Hamburg as the third largest port in Europe.*

We will change "biggest" to "largest" port in Europe.

*Line 70: how deep is the Elbe estuary? This gives an idea if sedimentary processes (e.g., N2O production) may affect N2O concentration in the surface water column.*

We will add information about the depth of the Elbe estuary in our study site description. The Elbe estuary has a depth of 15 – 20 m, which is maintained by regular deepening and dredging operations. Upstream of the Port of Hamburg, the water depths is less than 10 m (Hein et al., 2021).

*Figure 1: There are too many city names on the map, which is distractive. It may be clearer to label only the key cities like Cuxhaven or island Scharhorn where the Elbe River enters the North Sea or Oortkaten.*

We will change the map and label only relevant cities as suggested.

*Lines 85-87: Why transect sampling was performed after high tides? What's the effect of tides on N2O concentration? Tidal cycles of N2O concentration have been observed in other estuaries (Goncalves et al., 2015; Barnes et al., 2006).*

We chose our sampling strategy (upstream against the outgoing tide) to prevent interference of tidal effects on our measurements. Aim was to have comparable data for each cruise at similar tidal states with comparable current and mixing conditions. We started after high-tide and travelled against the outgoing tide to make sure that we did not move with the same water masses while travelling upstream.  We will explain our chosen sampling strategy in a revised manuscript version.

Tidal effects will very likely affect nitrous oxide concentrations in the Elbe estuary, but this is not the focus of the present manuscript. We will briefly address possible tidal effects in a revision.

*Line 116 in Equation 1: is N2Ocw the partial pressure of N2O in water? Otherwise, the saturation should be calculated as the N2Ocw/N2Oeq*100 where N2Oeq is the equilibrated N2O concentration with atmosphere. Similarly in Equation 3. N2Oair should be N2Oeq.*

For our calculations, we used the average atmospheric $N_2O$ concentrations measured on each specific day of the cruise to calculate expected atmospheric equilibrium concentration considering the solubility function of Weiss and Price (1980) and atmospheric pressure. We will change $N_2O_{air}$ to $N_2O_{eq}$ to prevent confusion and add a short description about the calculation in the method section.

*Line 143: Why nitrate concentration increased at 700 km? Are there tributaries or point sources?*

We regard this as a result of nitrification, rather than a point source. Dähnke et al. (2008) identified the Elbe estuary along its salinity gradient as a significant source of nitrate with high nitrate production in the maximum turbidity zone (MTZ). Sanders et al. (2018) measured highest nitrification in the Hamburg port region, which was not covered in the research done by Dähnke et al. (2008). Both studies highlighted the importance of nitrification along the Elbe estuary. Further, our results in section 4.2 showed ongoing nitrification fueled by marine organic matter along the mesohaline estuary and indicated that nitrate is produced by coupled remineralization with nitrification from both the riverine and marine site of the estuary.

In general, point sources play a subordinate role in the nitrogen input of the Elbe estuary (Hofmann et al., 2005; IKSE, 2018) with dominating agricultural sources in the upper and middle Elbe River. We will add this information to the study site description We believe the offset between region of highest nitrification rates (Port of Hamburg) and nitrate peak in the estuary (stream kilometer) is a result of different spatiotemporal scales, as suggested by  Sanders et al. (2018). They found that the position of

the nitrate maximum and nitrate gain over the entire estuary depended on the processing rates and was thus coupled to discharge conditions. We will briefly address this in a revised version of our manuscript.

***Lines 148-149: Why ammonium and nitrite concentration increased near Hamburg Port? Is it due to internal organic matter remineralization or point sources or sedimentary flux?***

Ammonium and nitrite concentration increases in the Hamburg port due to remineralization of organic material coming from upstream regions of the estuary. The sudden increase of water depth in the Port leads to a light limitation and decomposition of riverine organic material. As a result, ammonium and nitrite can accumulate leading to measurable peaks of both nitrogen forms in the Hamburg port region with the ammonium peak usually occurring upstream of the nitrite peak. At the same time, the decomposition of phytoplankton leads to increasing C/N values. Respiration processes predominate causing along with nitrification intense oxygen depletion in the Port of Hamburg. This succession of biogeochemical turnover has been addressed previously (e.g. Schroeder, 1997; Kerner and Spitzy, 2001; Schlarbaum et al., 2010; Schöl et al., 2014; Sanders et al., 2018; Dähnke et al., 2022). We addressed the succession of nitrogen turnover in the Port of Hamburg in section 4.3 of the discussion (L324 – 329). We will revise this section to clarify the state of research.

This also led us to one of our main conclusion, as riverine organic material is not only lead to remineralization/nitrification but also to intense oxygen consumption and therefore possible denitrification in the Hamburg port region fueling $N_2O$ production in this area of the Elbe estuary. Further, we will discuss possible nitrogen turnover processes and benthic-pelagic coupling in more detail as also suggested by reviewer 2.

***Figure 2: It is hard to tell the difference among each cruise with so many colored lines. How about presenting data from the same season using the same color to illustrate the seasonality as a supplementary figure?***

Thanks for this suggestion, we will add a figure like suggested by the reviewer to the supplements and will refer to it in the text.

***Lines 211-218 and lines 232-234: Figure 4 a and b are both from June, summer. The linear positive relationship between AOU and excess N2O suggests N2O production from nitrification (e.g., Nevison et al., 2003). The increase in the slope should be interpreted as an increase in the N2O production yield or external N2O input (e.g., point source).***

Thanks to the reviewer for this comment. We will include figures from all cruises in the revised version of the manuscript.

We indeed identified nitrification as responsible production processes both in the mesohaline estuary (section 4.2) and the Hamburg port region (section 4.3). In a revised manuscript version, we will clarify that our data suggest that $N_2O$ yield varies due to changes in nitrification dynamics, rather than point sources. As stated above, we will also address the role of point sources in the study site description.

Further we will also correct the statement in L232-234:

"As shown in Fig. 4, the relationship between $N_2O_{xs}$ and AOU were influenced by either initial riverine $N_2O$ production, or in-situ production along the estuary."

*Figure 4: It would be interesting to systematically/statistically assess the relations between excess N2O and environmental factors like salinity (non-conservative behavior of N2O) or dissolved inorganic nitrogen (infer N2O production pathways), PN, PC, and SPM. There seems to be a good relation between N2O and ammonium/nitrite concentration shown in Figure 2.*

We understand that a systematically and statistical assessment of the relations would help the reader to follow our discussion. We did assess the statistical relations in sections of our discussions (e.g. L:283, L:309, L:314-316, L319), but for clarity, we will we will add a section regarding the statically analysis in the results chapter in the revised manuscript.

During data interpretation, we tested diverse presentation and analysis methods and found correlations not necessarily the best suited outlet to describe and analyze our data. Correlations were distorted by the spatial offset between the ammonium, nitrite and $N_2O$ peaks, which we attribute to a succession of nitrogen bearing substances during turnover processes like nitrification (Sanders et al., 2018 and section 4.2). Thus, we chose another way to visualize the data by inserting the Figure 6 (L268) and S3 of the supplementary material. Further, we addressed the relations of $N_2O$ and various forms of nitrogen to identify $N_2O$ production processes and their controls in the section 4.2 (e.g. L258-259, L283-285, L310-317). For the Hamburg Port region, we decided against a detailed derivation of $N_2O$ production processes as this was already done by Brase et al. (2017). Therefore, we added the statement in L303-305.

We hope that a section on statistics in the results section, as mentioned above, and an earlier reference to the supplementary material in *Section 3.1* is sufficient to meet this criticism.

*Lines 242-243 and Figure 5: What about the variations of the N2O%, oxygen and total nitrogen concentration? The riverine N concentration is decreasing, what about the changes in other point sources of N input along the estuary (e.g., from wastewater treatment plants) or concentration in the estuary?*

We agree with the reviewer that a more detailed analysis of a long-term trend of $N_2O$ concentrations and reasons for changes would be very interesting. However, for our study we focus on seasonal variations rather than a long-term trend analysis. With section 4.1 we aimed to compare our results with a broader spatial and temporal scale by including a short comparison to other estuaries as well as with previous measurements from the Elbe estuary. This gives a hint towards temporal trends, but seasonal variability and data coverage make the long-term data difficult to interpret.

Briefly, the biogeochemical processes occurring in the Elbe estuary have drastically changed over the last 50 years: (1) the reunification of Germany and the collapse of East-German industry had led to significant improvements of water quality (e.g. Guhr et al., 2000), (2) decision to combat eutrophication in the North Sea in the 1980s and (3) improved waste water management resulted in a significant reduction of riverine nutrient loads (de Jong, 2007; Van Beusekom et al., 2019; Bergemann and Gaumert, 2010). Dähnke et al. (2008) showed that this led to a change of dominating denitrification towards significant nitrification in the Elbe estuary. Thus, a profound long-term analysis would be in need of its own paper, for which we think our data coverage is insufficient given the seasonal variability. As an example, measurements from only one cruise are available for the 1990s.

However, due to this comment as well as from other comments of the reviewer, we see the need to address the influence of point sources on nitrous oxide concentrations and nitrogen turnover in the Elbe estuary, which we already elaborated in a reply of an early comment above and which we will include in a reviewed version of our manuscript.

**Line 272: "this suggests"**

We will add the "s".

**Line 273: how is MTZ defined? What threshold of suspended particle material is used to define the MTZ?**

Generally, the occurrence of an MTZ is unique to each estuary and is generated by the balance between river-induced flushing and upstream transport of marine SPM as well as a function of estuarine geomorphology, gravitational circulation and tidal flow, trapping the particles in the MTZ (Bianchi, 2007; Sommerfield and Wong, 2011; Winterwerp and Wang, 2013). Thus, the MTZ is usually located in the onset of the salinity gradient of an estuary (Burchard et al., 2018).

The MTZ is – also in literature – often assessed based on relative changes in SPM or turbidity, and is mostly located between stream km 670 and 710 (e.g. Bergemann, 2004) in the Elbe estuary. During our cruises, SPM and turbidity were not always measured consistently, depending on instrument and personnel availability – during some cruises suspended particulate matter concentrations were measured using filtration techniques, during some cruises we obtained the data from turbidity sensors, which are not entirely intercomparable. Therefore, we did not defined a threshold of suspended particulate matter to define the MTZ, but also used relative change during each cruise to define it.

We will add colorbars indicating the relative change for SPM concentrations to the figure 6 and supplement material S3.

**Line 287 and 296-297: clarify the reference: Kappenberg and Fanger, 2007 (German?) and source of organic matter from the North Sea into the Elbe estuary.**

Kappenberg and Fanger (2007) is a German report from the former research center GKSS studying the sediment transport events in the tidally influenced Elbe estuary, the German Bight and the North Sea. The report (including a short summery in English) is available at https://www.hereon.de/imperia/md/content/hzg/zentrale_einrichtungen/bibliothek/berichte/gkss_berichte_2007/gkss_2007_20.pdf, last accessed: 21.03.2023.

We will include a peer-review reference that show a upstream transport of suspended matter into the Elbe estuary due to tidal transports in the Elbe estuary (Schoer, 1990). For the Ems estuary, Schulz and Umlauf (2016) showed an upslope transport of suspended matter due to tidal pumping is possible.

**Lines 311-313: How about showing the relations between ammonium, nitrite and N2O in figures?**

We decided to show the relation between ammonium, nitrite and $N_2O$ plotted against stream kilometers in Figure 6b and supplementary material S3. We found that the spatial progression of nitrogen containing substances were more illustrative than scatter plots or correlations for each substance, cruise and production areas, which would lead to a lot of figures added in the supplements. These relations were distorted by the spatial offset between the occurring ammonium, nitrite and $N_2O$ peaks, which we explained by a succession of nitrogen bearing substances during turnover processes like nitrification (e.g. section 4.2). Therefore, we found our choice of presentation better suited. We will add a section about the statistical analysis and relations of individual parameters in the results section. We will further rephrase the statement (L311-313) and add a reference to Figure 6 and the supplementary material.

*Line 315: What are R values? R is positive for nitrite concentration.*

R is the Pearson correlation coefficient. We will clarify this in the text by adding a short section about our statistical analysis in the *Methods* section of our manuscript. Nitrite concentrations correlated positive with $N_2O$ leading to a positive correlation coefficient.

*Line 320-321: Is nitrification responsible for the remaining oxygen consumption?*

Yes, nitrification is responsible for the remaining oxygen consumption. Sanders et al. (2018) assessed 25 % of the oxygen consumption in the Hamburg Port region were caused by nitrification, which was in line with results from Schöl et al. (2014). We will clarify this in the text.

*Line 326 and Figure S1: why C/N ratio was so high in 2021 March?*

Thanks for this good observation from the reviewer. We double checked our measurements, and the data appear correct and sound. We have no easy explanation at hand, but speculate that a calcareous algae bloom in the North Sea might be a potential cause. However, we have no further evidence for this hypothesis. We will not address this further, as the C/N ratios are not a crucial parameter for our discussion. However, we would like to keep the data in the manuscript as they might be insightful for later research and other researchers.

*Line 345-347: "Ammonium and N2O concentrations are high in the pore water of underlying sediments". Reference or example of the concentration. What about the timing of deepening and dredging works in the Hamburg Port compared to the cruise periods?*

We will add references for this statement (Zander et al., 2020) and rephrase the sentence as we only assumed elevated $N_2O$ concentrations in pore waters, eg.: "Ammonium concentrations are high in the pore water of underlying sediments (Zander et al., 2020), which we assume lead also to elevated $N_2O$ concentrations due to occurring nitrifier-denitrification in the sediments that was found by Deek et al. (2013)."

The Elbe estuary is constantly deepened and dredged along the entire transect to grant access for big container ships. However, we did not compare operation locations with the measured $N_2O$ concentrations except for our March cruises as we did not see big differences in the spatial variation of the $N_2O$ profiles, which were not explainable by in-situ production, nor found a strong correlations between $N_2O$ and suspended particulate matter concentrations. However, the effect of dredging and deepening operations as well as the effect of higher suspended particulate matter concentrations in the water column are a very interesting research question that should be worked on in the future. Ideally, in cooperation with the Hamburg Port Authority, who is responsible for maintain the water depth in the Elbe estuary. We will address this in a bit more detail in section 4.3 while discussion possible effects on $N_2O$ production in the Hamburg Port region as also suggested by the second reviewer.

*Lines 360-361: Has there been any N2O measurement from this wastewater treatment plant (WWTP) Köhlbrandhöft? The ammonium concentration in 2021/03 is not exceptionally high compared to previous cruises (e.g., 2020/06). What about the direct N2O output from the wastewater treatment plant?*

The operators of the WWTP measured $N_2O$ concentrations during our cruise, but did not detect elevated $N_2O$ concentrations. However, direct $N_2O$ output from the WWTP was not measured and the increased ammonium loads leaving the WWTP corresponded to the increased $N_2O$ concentration measured. We assume that the $N_2O$ is not produced within WWTP itself, but from elevated ammonium concentrations in the Elbe in warmer waste water.

We assume that we did not see an extraordinary ammonium peak due to the distance the WWTP outflow and our measurement transect, ~ 2 km. The outflow of the WWTP is located in the Southern Elbe, which joins the sample stretch at stream kilometer 626. We assumed that ammonium were rapidly converted to $N_2O$ as the warmer and biological active waste water entered the Elbe estuary before it reached the Northern Elbe.

We were admittedly surprised by the extraordinary $N_2O$ concentrations in March 2021, and even more so when the results were not reproduced in the following year. Consequently, we concluded that an extraordinary event must have caused the $N_2O$ peak. Since we could not detect any relation to the deepening and dredging work in the Port area and our measurements fitted to extraordinary operation condition in the WWTP, we assumed this might be the source, especially as our hypothesis was confirmed by the WWTP operators. Additional measurements confirmed that $N_2O$ concentration was not elevated near the WWTP outlet under normal conditions.

We will carefully revise this discussion section for clarity.

***Figure 7. Use month or season as the x axis instead of cruise number? Add description of the boxplot. Why not adding error bars for emissions?***

We will add error bars to the emissions plot Fig. 7b.

Figure 7 already shows flux densities and emissions plotted against months not cruise numbers. We will adapt this plot by separating the data into seasons (winter: March), spring (April and May), summer (June and July) and late summer (August and September). We will change the color scheme to better distinguish the winter season.

***Table 3. How is annual N2O emission calculated? Since there is a seasonal variation in the N2O flux, monthly or seasonal N2O emission may be more representative. Because N2O flux was measured at a high spatial resolution, it may be useful to calculate the N2O flux across the whole estuary by integrating the flux and area section by section (e.g., River section, Hamburg port, Oligohaline section) instead of multiplying the average N2O flux by the whole area of Elbe estuary.***

We described our annual emission calculations in the method section 2.4 (L124-126). We used this annual average for comparability with other studies, but will revise this calculation, to individually address sections and seasons. We found respective areas for sections from stream kilometer 610 - 632, 632 - 704 and 704 – 750 (Geerts et al., 2012), which fit reasonably well with our described sections. We will probably use these to calculate the fluxes and emissions as suggested by the reviewer. Further, we will differentiate into more seasons between winter (March), spring (April and May), summer (June to July) and last summer (August and September) to better reflect seasonality.

***Line 406: Why do you think there is no seasonality in N2O emission? N2O flux is different comparing spring, summer and winter shown in Table 3.***

We were carried away by comparing summer and winter emissions, which were (for us surprisingly) similar and missed to discuss the effects of spring and late summer on $N_2O$ emissions. As a starting hypothesizes we considered winter emissions significant lower than summer emissions due to missing in-situ production. However, we found that high nitrogen loads coming from the river leading to high emissions in winter comparable to emissions in summer, which were driven by production at two production sites.

For the revised manuscript, we calculated emissions as suggested by the reviewer in a comment above by separating our data into more seasons (winter: March, spring: April to May, summer: June to July, late summer: August to September). With the new upscaling technique we found highest emissions in

winter. Thus, we will restructure our last section of the discussion in line with suggestions of the other reviewer focusing on the relevance of our research for a broader audience.

**References**

Barnes, J. and Upstill-Goddard, R. C.: N2O seasonal distributions and air-sea exchange in UK estuaries: Implications for the tropospheric N2O source from European coastal waters, J. Geophys. Res. Biogeosciences, 116, https://doi.org/10.1029/2009JG001156, 2011.

Bergemann, M.: Die Trübungszone in der Tideelbe - Beschreibung der räumlichen und zeitlichen Entwicklung, Wassergütestelle Elbe, 2004.

Bergemann, M. and Gaumert, T.: Elbebericht 2008, Flussgebietsgemeinschaft Elbe, Hamburg, 2010.

Bianchi, T. S.: Biogeochemistry of Estuaries, Oxford University Press, New York, 706 pp., 2007.

Brase, L., Bange, H. W., Lendt, R., Sanders, T., and Dähnke, K.: High Resolution Measurements of Nitrous Oxide (N2O) in the Elbe Estuary, Front. Mar. Sci., 4, 162, https://doi.org/10.3389/fmars.2017.00162, 2017.

Burchard, H., Schuttelaars, H. M., and Ralston, D. K.: Sediment Trapping in Estuaries, Annu. Rev. Mar. Sci., 10, 371–395, https://doi.org/10.1146/annurev-marine-010816-060535, 2018.

Dähnke, K., Bahlmann, E., and Emeis, K.-C.: A nitrate sink in estuaries? An assessment by means of stable nitrate isotopes in the Elbe estuary, Limnol. Oceanogr., 53, 1504–1511, https://doi.org/10.4319/lo.2008.53.4.1504, 2008.

Dähnke, K., Sanders, T., Voynova, Y., and Wankel, S. D.: Nitrogen isotopes reveal a particulate-matter-driven biogeochemical reactor in a temperate estuary, Biogeosciences, 19, 5879–5891, https://doi.org/10.5194/bg-19-5879-2022, 2022.

Deek, A., Dähnke, K., van Beusekom, J., Meyer, S., Voss, M., and Emeis, K.-C.: N2 fluxes in sediments of the Elbe Estuary and adjacent coastal zones, Mar. Ecol. Prog. Ser., 493, 9–21, https://doi.org/10.3354/meps10514, 2013.

FGG Elbe, Flussgebietsgemeinschaft Elbe: Naehrstoffminderungsstrategie für die Flussgebietsgemeinschaft Elbe, Gemeinsamer Bericht der Bundesländer und der Flussgebietsgemeinschaft Elbe, 2018.

Geerts, L., Wolfstein, K., Jacobs, S., van Damme, S., and Vandenbruwaene, W.: Zonation of the TIDE estuaries, TIDE toolbox, 2012.

Guhr, H., Karrasch, B., and Spott, D.: Shifts in the Processes of Oxygen and Nutrient Balances in the River Elbe since the Transformation of the Economic Structure, Acta Hydrochim. Hydrobiol., 28, 155–161, https://doi.org/10.1002/1521-401X(200003)28:3<155::AID-AHEH155>3.0.CO;2-R, 2000.

Hanke, V.-R. and Knauth, H.-D.: N2O-Gehalte in Wasser-und Luftproben aus den Bereichen der Tideelbe und der Deutschen Bucht, GKSS-Forschungszentrum, Weinheim, 1990.

Hein, S. S. V., Sohrt, V., Nehlsen, E., Strotmann, T., and Fröhle, P.: Tidal Oscillation and Resonance in Semi-Closed Estuaries—Empirical Analyses from the Elbe Estuary, North Sea, Water, 13, 848, https://doi.org/10.3390/w13060848, 2021.

Hofmann, J., Behrendt, H., Gilbert, A., Janssen, R., Kannen, A., Kappenberg, J., Lenhart, H., Lise, W., Nunneri, C., and Windhorst, W.: Catchment–coastal zone interaction based upon scenario and model analysis: Elbe and the German Bight case study, Reg. Environ. Change, 5, 54–81, https://doi.org/10.1007/s10113-004-0082-y, 2005.

IKSE, I. K. zur S. der E.: Strategie zur Minderung der Nährstoffeinträge in Gewässer in der internationalen Flussgebietsgemeinschaft Elbe, Internationale Kommission zur Schutz der Elbe, Magdeburg, 2018.

de Jong, F.: Marine Eutrophication in Perspective, Springer, Berlin, Heidelberg, 335 pp., https://doi.org/10.1007/3-540-33648-6, 2007.

Kerner, M. and Spitzy, A.: Nitrate Regeneration Coupled to Degradation of Different Size Fractions of DON by the Picoplankton in the Elbe Estuary, Microb. Ecol., 41, 69–81, https://doi.org/10.1007/s002480000031, 2001.

Radach, G. and Pätsch, J.: Variability of continental riverine freshwater and nutrient inputs into the North Sea for the years 1977–2000 and its consequences for the assessment of eutrophication, Estuaries Coasts, 30, 66–81, https://doi.org/10.1007/BF02782968, 2007.

Sanders, T., Schöl, A., and Dähnke, K.: Hot Spots of Nitrification in the Elbe Estuary and Their Impact on Nitrate Regeneration, Estuaries Coasts, 41, 128–138, https://doi.org/10.1007/s12237-017-0264-8, 2018.

Schlarbaum, T., Daehnke, K., and Emeis, K.: Turnover of combined dissolved organic nitrogen and ammonium in the Elbe estuary/NW Europe: Results of nitrogen isotope investigations, Mar. Chem., 119, 91–107, https://doi.org/10.1016/j.marchem.2009.12.007, 2010.

Schoer, J. H.: Determination of the origin of suspended matter and sediments in the Elbe estuary using natural tracers, Estuaries, 13, 161–172, https://doi.org/10.2307/1351585, 1990.

Schöl, A., Hein, B., Wyrwa, J., and Kirchesch, V.: Modelling Water Quality in the Elbe and its Estuary – Large Scale and Long Term Applications with Focus on the Oxygen Budget of the Estuary, Küste 81 Model., 203–232, 2014.

Schroeder, F.: Water quality in the Elbe estuary: Significance of different processes for the oxygen deficit at Hamburg, Environ. Model. Assess., 2, 73–82, https://doi.org/10.1023/A:1019032504922, 1997.

Schulz, K. and Umlauf, L.: Residual Transport of Suspended Material by Tidal Straining near Sloping Topography, J. Phys. Oceanogr., 46, 2083–2102, https://doi.org/10.1175/JPO-D-15-0218.1, 2016.

Sommerfield, C. K. and Wong, K.-C.: Mechanisms of sediment flux and turbidity maintenance in the Delaware Estuary, J. Geophys. Res. Oceans, 116, https://doi.org/10.1029/2010JC006462, 2011.

Van Beusekom, J. E. E., Carstensen, J., Dolch, T., Grage, A., Hofmeister, R., Lenhart, H., Kerimoglu, O., Kolbe, K., Pätsch, J., Rick, J., Rönn, L., and Ruiter, H.: Wadden Sea Eutrophication: Long-Term Trends and Regional Differences, Front. Mar. Sci., 6, https://doi.org/10.3389/fmars.2019.00370, 2019.

Weiss, R. F. and Price, B. A.: Nitrous oxide solubility in water and seawater, Mar. Chem., 8, 347–359, https://doi.org/10.1016/0304-4203(80)90024-9, 1980.

Winterwerp, J. C. and Wang, Z. B.: Man-induced regime shifts in small estuaries—I: theory, Ocean Dyn., 63, 1279–1292, https://doi.org/10.1007/s10236-013-0662-9, 2013.

Zander, F., Heimovaara, T., and Gebert, J.: Spatial variability of organic matter degradability in tidal Elbe sediments, J. Soils Sediments, 20, 2573–2587, https://doi.org/10.1007/s11368-020-02569-4, 2020.

---

## Author Comment (AC2)

*This work presents five years (two years of new data) of seasonal (winter v summer) data on dissolved N2O concentrations in the Elbe River Estuary. The insights this can provide into interannual variations in aquatic N2O is relatively unique. The site of the study itself, which encompasses a large industrial port, is also important in terms of better understanding anthropogenic impacts on aquatic N2O emissions. The authors show that, even though the source of N2O seems to be strongly seasonal, emissions remain relatively consistent over the year. This is a new and interesting finding. Overall this is a nice study with the potential to be a useful contribution to both the journal and scientific understanding of aquatic N2O.*

*However, there are a few weaknesses with the data analysis and discussion that need to be addressed to ensure that the emissions are accurately represented and the findings are clearly conveyed.*

We thank the reviewer for their constructive and helpful review of our paper as well as for the many great ideas to improve our research.

Following, we reply to each issue individually, and explain the changes we will make to the revised manuscript to meet the reviewers criticism. Reviewer comments are written in bold italics, our answers are kept in plain font.

*1. Converting dissolved concentrations to emissions: Like many studies, here the authors measured the dissolved concentration of the gas (N2O), and then converted this into water-air emissions based on a gas transfer velocity (k). Gas transfer velocities can be highly variable, especially in estuaries where the importance (and magnitude) of factors like wind, flow velocity, and water depth can all vary a lot over space and time. This complexity is reflected in the wide range of empirical k value parameterisations that have been developed for estuaries (see e.g., Rosentreter et al. (2021), also Hall and Ulseth (2019) for a good review of the topic, albeit for freshwater systems). However, here the authors convert measured concentrations to emissions using a single parameterisation (L116-125). This creates considerable uncertainty, which is not reflected in the reported estuary emissions estimates. Emissions should be recalculated using 3-5 k parameterisations, and the variability of these outputs reported in the results / figures. More information should also be supplied on the wind speed data used in the parameterisations. It is important to understand how the values measured during the campaigns compare to 'average' conditions around the estuary when considering the upscaled seasonal emissions values (e.g., are emissions estimates likely to be on the low side because cruises were only done on low-wind days?).*

Thanks for this helpful comment. As suggested we will include calculations using two other parameterizations and report the variability. We will calculate and discuss the effect of wind speeds in relation to average conditions along the estuary as well as add the information about average wind speed. In general, long-term average wind speeds along the Elbe estuary range between 2.8 and 5.8 m s$^{-1}$ (https://www.dwd.de/DE/leistungen/windkarten/deutschland_und_bundeslaender.html, last accessed: 05.04.2023).

*2. Relationship between N2O and N inputs: As discussed in the paper intro here, aquatic N2O emissions are generally predicted based on N loads to the system (i.e., leaching of N, inputs from WWTPs, etc). While here N2O emissions are discussed and presented, the N inputs side of the equation is not clear to me. In the site description it says that annual N load were ~80 Gg y-1 (L67) – but does this mean the estuary receives this much N, or discharges this much N? And how does this break down between sources (WWTPs v river discharge)? On L231 it says that N2O emissions were low relative to other high N input estuaries. But how do N inputs into the Elbe stack up compare to these other estuaries? I particularly wonder how the 'point source' N loads around the port might stack up with those in other urban estuaries where N2O emissions have been measured, e.g., (Wells*

*et al., 2018). Constraining the other side of the N2O emissions v N inputs equations is critical for placing these findings into a more global context. Within the study, more information on N loads will also be important for picking apart the seasonal emissions drivers. How much N enters the estuary at the port? Is this input seasonally variable? Did it vary between the sampled years? Do these variations correspond with variations in emissions (particularly the size of the winter N2O-excess excursion)?*

The N-loads of 80 Gg yr$^{-1}$ are calculated from concentrations data at a station "Seemanshoeft", which is located at the Hamburg Port (stream kilometer 628.9). In general, point sources play a subordinate role in the nitrogen input of the Elbe estuary (Hofmann et al., 2005; IKSE, 2018) with dominating agricultural sources in the upper and middle Elbe River (Hofmann et al., 2005; Johannsen et al., 2008). We will add this information to the study site description. Further, we will calculate annual varying DIN and total nitrogen (TN) loads for our observation period and list the results in the supplements as recent TN loads were lower varying between 40.67 kt-N yr$^{-1}$ and 60.08 kt-N yr$^{-1}$ from 2015 to 2021 (FGG, 2021).

We will restructure our section 4.5 of the discussion: We will address the N$_2$O emissions and N inputs relation based on a comparison of the amount of DIN released as N$_2$O for annual loads and for each cruise separately. In a revised version, we will compare flux densities and the N$_2$O emission versus N input equation to a wider set of literature data and further estuaries. We will more clearly refer to a change of drivers for N$_2$O emissions in winter (high riverine input and nitrification) versus spring and summer (organic matter). Finally, we will highlight the link of N$_2$O emissions to eutrophication phenomena to broaden the scope of our paper.

*1. Introduction: It is not entirely clear how studying N2O in the Elbe estuary will advance understanding of aquatic N2O emissions / fill a needed research gap. A stronger transition between the penultimate and last paragraphs of the discussion is needed (how does the present study relate to the broader literature). Stating a testable hypothesis, rather than just site-specific study objectives, in the last paragraph may also help make the study more clearly relevant to the broader scientific community. Is this just a case study or will the data help us understand estuary N cycling and gaseous emissions in a more fundamental way?*

Thanks for this great comment! We will focus on the connection to a broader scientific audience, likely by stressing the general link of N$_2$O emissions and eutrophication phenomena in heavily managed estuaries around the world. So that overall, we provide a better insight of controls on seasonal varying N$_2$O production and emissions from heavily anthropogenic impacted estuaries.

*2. Discussion: While I think overall the data interpretation makes sense, the discussion section currently reads as a bit descriptive and could go further to place these findings in a broader context (rather than just the context of how we understand the Elbe River Estuary). This could include in particular more discussion of N cycling in urban estuaries / where there are point N pollution. Where else in the world would the observed seasonal patterns be expected to be found? I also think there is missing some discussion of 'alternative hypotheses' – work through the logic of why denitrification is not thought to be the primary driver of N2O in the estuary, and why benthic production (e.g., (Chen et al., 2022)) is also ruled out. Also please carefully edit to ensure that you are not repeating results in this section.*

We will discuss the alternative hypotheses in more detailed as we described below in the replies for the individual comments. We want to clarify that we found both denitrification in the sediments as well as production in the water column responsible processes for N$_2$O production in the Port of Hamburg. However, we will discuss the effects of benthic fluxes and production in more detail (see specific comment below). In a revision, we will carefully remove results bits from the discussion

section, and will focus more on comparing our finding with research from other estuaries to address a broader audience.

**3. Conclusion: This is currently very focused on untangling what exactly is happening within the Elbe River Estuary, but the implications for broader understanding of aquatic N2O production and emissions are not clear.**

We will add a section about the implications for a broader understanding by (1) addressing our newly formulated hypothesizes as well as (2) summarize the findings in a revised discussion section that will compare our findings to other estuaries.

**L17-19: This sentence is not clear (how does N2O 'compensate' for decreasing N loads?), please reword.**

This statement was supposed to summarize our findings of section 4.1, where we complemented observations done by Brase et al. (2017) with our new data sets: Previous $N_2O$ measurements done in 1980s and 1990s showed a significant reduction of $N_2O$ saturation due to reduced riverine nitrogen loads and higher dissolved oxygen conditions. Water quality in the Elbe estuary improved significantly after the reunification and collapse of East German industries (e.g. Guhr et al., 2000). In the 1980s, high nitrogen loads and low oxygen conditions favored denitrification leading to high $N_2O$ saturations (Hanke and Knauth, 1990). With improving water quality, the importance of denitrification decreased (Dähnke et al., 2008), which probably led to the decrease in $N_2O$ saturations (Brase et al., 2017). However, compared to the study in 1997 (Barnes and Upstill-Goddard, 2011) our measured $N_2O$ saturations did not further decreased despite a continuous reduction of nitrogen loads entering the estuary form the upstream river (e.g. Figure 5, FGG Elbe, 2018; Radach and Pätsch, 2007). These findings suggesting that in-situ $N_2O$ production along the estuary is important and compensates the overall effect of decreasing nitrogen loads in the last decades.

Also considering previous reviewer comments we will address the importance of our research for a broader audience in the abstract and thus, rephrase or remove this finding from the abstract. Instead, we will probably address the results from the comparison of our results with other research and the implications for other heavily managed estuaries (see comment above).

**L22-24: "In winter, high riverine N2O concentrations led to high N2O emissions from the estuary, whereas in summer, estuarine biological N2O production led to equally high N2O emissions." This is I think getting at a crucial point (that although seasonal magnitude of N2O fluxes did not differ the drivers of these fluxes did), the meaning is not clear. What is the difference between winter 'high N2O concentrations' and summer 'high N2O production'? Reword to be more precise about these differences.**

We wanted to address the seasonal varying drivers leading to high $N_2O$ emissions along the Elbe estuary. We will rephrase the section of the abstract.

**L70: How often is 'on a regular basis'? e.g., weekly, yearly, every three years?**

Deepening and dredging operations in the Elbe estuary are performed year-round, if necessary every couple of days. We will clarify this.

**L86: Suggest changing 'steaming upstream' to 'travelling upstream' (steaming sounds a bit antiquated)**

We will change the wording as suggested by the reviewer.

***L101-104: More information on number of nutrient samples collected per survey, as well as method detection limits and precision, would be useful.***

We usually took about 30 to 40 samples during each cruise. We will include the detection limits and also add a short description addressing the range of samples numbers.

***L109: How often was 'regularly'? e.g., before each cruise?***

We measured standard gas mixtures before and after each day of our campaigns that usually lasted from two to three days. We will clarify this in a revision.

***L116: How often, and how, was dry air sampled during each cruise?***

We measured dry air before and after each day, as with standard gas mixtures. Also, we continuously measured dry air overnight in between our sampling days. We will include this in the text. We used an air duct from the deck of our research vessel into our analyzer.

***L122: The term 'flux densities' is not one I'm familiar with – more common to see something like 'water-air fluxes' or 'evasion'.***

We used the term "flux densities", because we had both positives and negative fluxes. Therefore, we would like to stick to this term, which has been used previously (e.g. Brase et al., 2017; Bange et al., 2019; Morgan et al., 2019; Forster et al., 2009). However, we will insert a brief definition of the term.

***L123-125: Please provide some clarification on the upscaling approach used to calculate whole-estuary emissions. From the description it sounds like the mean flux was multiplied by the estuary surface area? Or were these calculations area-weighted, and if so at what resolution?***

We multiplied the mean flux with the estuary surface area. Due to the comment of the other reviewer, we decided to change our calculation approach and to calculate mean fluxes for three different regions: 1. Limnic and Hamburg Port region, 2. Oligohaline section and 3. Mesohaline/Polyhaline section while separating our data set into seasons (winter: March, spring: April and May, summer: June and July, late summer: August and September). We will use this data to calculate annual emissions from the entire estuary. Further, we will calculate flux densities and emissions for various wind speeds (wind during our cruises, annual and seasonal average wind speeds). We will clarify this method in the reviewed version of our manuscript.

***L127-128: Citation?***

We will add a citation (e.g. Nevison et al., 2004; Walter et al., 2004).

***L148: Low relative to what?***

"low" is less than 1 µmol $L^{-1}$ – we will specify this.

***L163-189: Separating the N2O data into different sections for the different units (molar concentrations, % saturation, water-air fluxes) is confusing as these are all inter-related. For instances, it is hard to make sense of the meaning of the molar concentrations without also considering whether these reflect changes in percent saturation (i.e., changes due to water temperature / salinity v source / production). I suggest integrating these lines of data (and thinking) to provide a clearer picture of estuary N2O patterns.***

We will change $N_2O$ concentrations to $N_2O$ saturations in Fig. 2. Thus, we will remove Fig. 3 in the text. As suggested by the reviewer in a comment below, we will focus on describing $N_2O$ saturations rather than concentrations to exclude effects on $N_2O$ solubility.

*L204: High relative to what?*

"High" is more than 100 % – we will specify this.

*L209-218: The AOU v N2O-excess relationship really highlights the importance, and seasonality, of the port for estuary N2O emissions, with distinct peaks in the winter and consumption in the summer. Given that this underpins the discussion around seasonal N2O source switching, I wonder if there is a way to include more than just these 'representative' plots in the main text. For instance, a table with info on AOU v N2O-excess slopes, and min-max range for the port? I think if the port data is excluded something like an ANCOVA could be used to compare shifts in slope relationships.*

In the Port region, $N_2O_{xs}$ and AOU had no linear relation during most of our cruises (e.g. June 2015 and August 2017). Therefore, a table with slopes and min-max ranges would miss crucial information. However, we will include figure S2 from the supplement in the main text so that we do not only show representative plots but all cruises. We will further test whether an individual (i.e., over distinct sections) assessment of slopes aids the discussion.

*L256-260: Interesting relationship between NO2- and N2O. This could be connected to previous work, e.g., (Sharma et al., 2022; Smith and Bohlke, 2019; Wertz et al., 2018)*

Thanks for these suggestion. We will address previous works that found a relation between $N_2O$ and nitrite in a revised version of our manuscript. In our data set, the accumulation of nitrite is a sign for stepwise nitrification rather than denitrification, which is in contrast to findings from e.g. Wertz et al. (2018) and Sharma et al. (2022).

*L314-316: This should be in the results section*

We will add a section about the statistical analysis in the results section and will move this bit into this new section.

*L318-324: Interesting! I wonder if the algae themselves could also be contributing to the N2O production, e.g., (Fabisik et al., 2023)*

Thanks for this great literature suggestion. We will address their findings briefly in a revised version of our manuscript.

*L330-332: This makes sense, but is this the only possible explanation for high emissions around the port area? What about wastewater inputs, enhanced benthic production, and/or enhanced groundwater connectivity due to dredging? Some discussion of these points will make this conclusion stronger.*

Thanks to the reviewer for this helpful comment. We largely ruled out most of these sources, but will certainly discuss this in more detail in our revised version. First, point sources (including the waste water treatment plant, WWTP) are considered to be from only minor importance in the Elbe estuary (Hofmann et al., 2005; IKSE, 2018). Further, we estimated the impact of the WWTP as less than 5 % even under low fresh water inflow. We will address this in the study site description.

We found no general trend between turbidity values and $N_2O$ saturation. Thus, we excluded an overarching effect of deeping and dredging operations on $N_2O$ levels, which we discussed later in the text (L:349 – 352).

We considered both sediment denitrification and water column nitrification responsible for the elevated $N_2O$ saturations. We will discuss the different explanations for high emissions in the port and partly restructure the discussion in section 4.3 in line with the suggestion of the reviewer.

***L357-358: How extreme was this rain event, i.e., was it more extreme than any rainfalls over the other five years of sampling? This will help verify the attribution, and also put the pulse into context. It would then be instructive to recalculate the seasonal budget with and without this pulse.***

The rain event had a statistical recurrence probability of one to five years (https://sri.hamburgwasser.de/, last access: 04.04.2023). We will check the effect of this rain event on the seasonal budget. We will probably only present the data in the supplements, but we will address the resulting differences in the main text. Considering the statistical recurrence probability, it is likely that similar events occurred during the last five years, but so far not during a comparable sampling cruise. Moreover, we assume that temperature was equally important as water mass, because cold rain water in the waste water treatment plant led to aggravated operation conditions. Thus, in warmer months the effect might be different. We will discuss this in more detail in a revised version of our manuscript.

***L392: If large riverine loads were the main driver, wouldn't there be a continuous decrease in concentration over distance? But instead emissions peak in the port.***

$N_2O$ concentrations were affected by a combined effect of river nitrate loads, and of additional production in the port region (L337-341). However, this production is also driven by river nitrogen loads, so that overall, river nitrogen or nitrate loads are crucial drivers of eutrophication as well as of $N_2O$ production in winter. We will clarify this in this section of the text.

***Table 2: Standard deviations for the air N2O concentrations would be helpful***

We will include standard deviations.

***Fig. 1: The most important pieces of info in this map (where sampling points are, where the port is, where the MTZ is) don't really stand out. Can you adjust colours, font size, etc to better highlight these key features? A scale bar for the main map would also be helpful.***

We will change the style of map to highlight key features and important locations as suggested by the reviewer.

***Fig. 2: I'm not sure that there is much value in showing N2O concentrations (in nM) here – the % saturation information in the subsequent figure is much more effective for showing fluctuations between seasons and over the salinity gradient, given the relatively low concentrations and the impact of both temperature and salinity on N2O solubility. It would also be helpful to have 'summer' and 'winter' headings at the top of the two columns to make the point of difference more immediately obvious.***

We will include a heading to the plot and change $N_2O$ concentrations to $N_2O$ saturations. Thus, we will remove Fig. 3 from the text. We will move the plot and description of $N_2O$ concentrations to the supplements.

***Fig. 3: A unified y axis scale would be helpful for picking out seasonal differences***

We decided against unified y-axis. Truly, it would help to visualize seasonal differences, but it would be hard to differentiate between the different cruises in spring and summer.

***Fig. 4: As above, unified axes scales would make differences between sampling dates much clearer.***

We decided against unified y-axis. The March 2021 cruise differs so much that it's hard to see variabilities in the other cruises if we use the same y-axis. However, we will use unified axes scales for all other cruises

***Fig. 5: Different y axes are needed for the different variables (N2O, O2, TN), if not different plot panels***

We will change the plot to include different plot panels for each variable.

***Fig. 6: I found this to be too many variables on the same plot to make much logical sense out of. I suggest separating into two panels, one for all of the N species (y axis unit is uM N), and then another with two y axes, one for PN and one for C/N.***

We will adapt the plot as suggested by the reviewer.

***Fig. 7: It would be helpful to use a different pattern or colour scheme to distinguish the winter v summer cruises.***

As we are planning to restructure this part of the discussion Figure 7 will most likely change. We will probably insert anew figure showing seasonal varying $N_2O$ saturation, emission estimates, DIN loads and $N_2O$/DIN ratios to highlight the seasonal varying dynamic of these parameters and that $N_2O$ emissions did not scale with DIN loads. Referring to a comment from the other reviewer, we will adapt this plot by separating the data into seasons (winter: March), spring (April and May), summer (June and July) and late summer (August and September). We will change the color scheme to better distinguish the winter season.

**References**

Bange, H. W., Sim, C. H., Bastian, D., Kallert, J., Kock, A., Mujahid, A., and Müller, M.: Nitrous oxide ($N_2O$) and methane ($CH_4$) in rivers and estuaries of northwestern Borneo, Biogeosciences, 16, 4321–4335, https://doi.org/10.5194/bg-16-4321-2019, 2019.

Brase, L., Bange, H. W., Lendt, R., Sanders, T., and Dähnke, K.: High Resolution Measurements of Nitrous Oxide (N2O) in the Elbe Estuary, Front. Mar. Sci., 4, 162, https://doi.org/10.3389/fmars.2017.00162, 2017.

FGG, F. E.: FIS der FGG Elbe - Physikalisch-chemische Qualitätskomponenten: Elbe - Bunthaus (Strom-km 609,8) - Wassertemperatur - Tagesmittelwert, 2021.

Forster, G., Upstill-Goddard, R., Gist, N., Robinson, C., Uher, G., and Woodward, E.: Nitrous oxide and methane in the Atlantic Ocean between 50°N and 52°S: Latitudinal distribution and sea-to-air flux, Deep Sea Res. Part II Top. Stud. Oceanogr., 964–976, https://doi.org/10.1016/j.dsr2.2008.12.002, 2009.

Hofmann, J., Behrendt, H., Gilbert, A., Janssen, R., Kannen, A., Kappenberg, J., Lenhart, H., Lise, W., Nunneri, C., and Windhorst, W.: Catchment–coastal zone interaction based upon scenario and model analysis: Elbe and the German Bight case study, Reg. Environ. Change, 5, 54–81, https://doi.org/10.1007/s10113-004-0082-y, 2005.

IKSE, I. K. zur S. der E.: Strategie zur Minderung der Nährstoffeinträge in Gewässer in der internationalen Flussgebietsgemeinschaft Elbe, Internationale Kommission zur Schutz der Elbe, Magdeburg, 2018.

Johannsen, A., Dähnke, K., and Emeis, K.: Isotopic composition of nitrate in five German rivers discharging into the North Sea, Org. Geochem., 39, 1678–1689, https://doi.org/10.1016/j.orggeochem.2008.03.004, 2008.

Morgan, E. J., Lavric, J. V., Arévalo-Martínez, D. L., Bange, H. W., Steinhoff, T., Seifert, T., and Heimann, M.: Air–sea fluxes of greenhouse gases and oxygen in the northern Benguela Current region during upwelling events, Biogeosciences, 16, 4065–4084, https://doi.org/10.5194/bg-16-4065-2019, 2019.

Nevison, C., Lueker, T., and Weiss, R. F.: Quantifying the nitrous oxide source from coastal upwelling, https://doi.org/10.1029/2003GB002110, 2004.

Sharma, N., Flynn, E. D., Catalano, J. G., and Giammar, D. E.: Copper availability governs nitrous oxide accumulation in wetland soils and stream sediments, Geochim. Cosmochim. Acta, 327, 96–115, https://doi.org/10.1016/j.gca.2022.04.019, 2022.

Walter, S., Bange, H. W., and Wallace, D. W. R.: Nitrous oxide in the surface layer of the tropical North Atlantic Ocean along a west to east transect, Geophys. Res. Lett., 31, L23S07, https://doi.org/10.1029/2004GL019937, 2004.

Wertz, S., Goyer, C., Burton, D. L., Zebarth, B. J., and Chantigny, M. H.: Processes contributing to nitrite accumulation and concomitant N2O emissions in frozen soils, Soil Biol. Biochem., 126, 31–39, https://doi.org/10.1016/j.soilbio.2018.08.001, 2018.

---

## Author Response (AR1)

**1 Response letter**

We thank the editor and reviewers for their constructive and helpful comments and suggestions. In the following sections, we reply to each comment individually, and explain the changes we have made to the revised manuscript. Note that we also slightly corrected the revised manuscript for stylistic issues and minor mistakes (grammar mistakes, recalculation of $N_2O$ flux densities, etc.). These changes do not affect the conclusion of the manuscript and are shown in the marked-up manuscript version. We address all comments in detail below, but would like to highlight some major changes:

(1)   In accordance with comments from the editor and reviewers, we adapted the manuscript to put our research in a wider

 scientific context, see response below. Furthermore, we changed the title to "Seasonal variability of nitrous oxide

 concentrations and emissions in a temperate estuary" to address a broader audience.

(2)   We recalculated $N_2O$ flux densities and emissions using four different parametrizations for the gas transfer coefficient

 and different wind speeds.

(3)   We adapted most of our figures according to the reviewers suggestions.

Reviewer comments are written in bold italics, our answers are kept in plain font.

**General remarks from the editor**

***Thank you for submitting your paper to Biogeosciences. Two referees have evaluated your paper and provided detailed***

***feedback, in particular regarding the need to improve the presentation and to better put your results in a wider context***

***(beyond a case study). In your detailed rebuttal, you indicate that you will be able to resolve most issues and I therefore***

***believe that a revised paper might be qualified for publication in Biogeosciences. Your revised version will likely be***

***evaluated again by one or both referees.***

We understand the need to put our results into a wider context. To accomplish this, we focussed more on the relation between

DIN and $N_2O$ as suggested by reviewer 2. In line with several other researchers (Borges et al., 2015; Marzadri et al., 2017;

Wells et al., 2018), we found a limited relation between both parameters and thus, we focused on understanding the drivers for this discrepancy. Since we identified organic matter availability as a main driver for $N_2O$ production in the Elbe Estuary, we concluded that in heavily managed estuaries with high agricultural loads, $N_2O$ emissions are clearly linked to eutrophication phenomena as already proposed by Wells et al. (2018). Therefore, we rewrote and restructured parts of our abstract and introduction towards a broader research question centered on N loads as drivers of $N_2O$ production in estuaries, and modified the last section of the discussion and conclusion to address the interplay of DIN and $N_2O$ in estuaries. Furthermore, we changed the title to "Seasonal variability of nitrous oxide concentrations and emissions in a temperate estuary" to address a broader audience. We hope that these changes are sufficient to meet the reviewers' suggestions.

We changed the figures in line with suggestions from both reviewers.

**1. Review comment (RC1) – 08.03.2023**

*Line 17-18: what do you mean by "compensated the effect of decreasing dissolved inorganic nitrogen (DIN) loads"?*

Also considering the comments of reviewer 2, we revised the manuscript to highlight the relevance of our research to a broader audience. Thus, we rewrote our abstract, focusing the relevance for a broader scientific community and highlighting the connection between eutrophication and $N_2O$ emissions. We also rewrote this phrase.

| Lines | Change |
|---|---|
| L24 - 29 | Changed to: "A comparison with previous measurements in the Elbe Estuary revealed that $N_2O$ saturation did not decrease alongside with DIN concentrations after a significant improvement of water quality in the 1990s that allowed for phytoplankton growth to reestablish in the river and estuary. This effect of phytoplankton growth and the overarching control of organic matter on $N_2O$ production, highlights that eutrophication and agricultural nutrient input can increase $N_2O$ emissions in estuaries." |

*Line 25: How does $0.24\pm0.06$ Gg $N_2O$ $y^{-1}$ emission compare to global estuarine $N_2O$ emission?*

We changed the emission calculation as suggested by both reviewers. We removed the emission estimate from the abstract focusing more on the relation with DIN loads and seasonal varying drivers in the Elbe Estuary, which lead to year-round high $N_2O$ emissions. We highlighted the relevance for a broader scientific community by focusing on the connection between $N_2O$ emissions and DIN loads, as well as the linkage to eutrophication in estuaries with high agricultural loads. In the new section 4.5 of our discussion, we now compare $N_2O$ emission estimates and the resulting $N_2O$:DIN relation across estuaries.

| Lines | Change |
|---|---|
|  | Removed $0.24 \pm 0.06$ Gg $N_2O$ $yr^{-1}$ from the abstract |
| L262-265 | Comparison of $N_2O$ saturation with other estuaries |
| L421-434 | Comparison of $N_2O$ emissions and $N_2O$:DIN relation with other estuaries |

*Lines 40-42: Denitrification could also occur in anoxic water column contributing to $N_2O$ production (Ji et al., 2018; Tang et al., 2022).*

| Lines | Change |
|---|---|
| L44-45 | Added denitrification in the water column as possible production pathway. |

*Line 44: specify Port Hamburg as the third largest port in Europe.*

| Lines | Change |
|---|---|
| L55 | Changed "biggest" to "largest" |

*Line 70: how deep is the Elbe estuary? This gives an idea if sedimentary processes (e.g., N₂O production) may affect N₂O*

*concentration in the surface water column.*

| Lines | Change |
|---|---|
| L82, L84-85 | Added information about the depth of the Elbe Estuary in our study site description |

*Figure 1: There are too many city names on the map, which is distractive. It may be clearer to label only the key cities like*

*Cuxhaven or island Scharhorn where the Elbe River enters the North Sea or Oortkaten.*

| Lines | Change |
|---|---|
| L86 | Changed Map (Fig. 1) |

*Lines 85-87: Why transect sampling was performed after high tides? What's the effect of tides on N₂O concentration? Tidal*

*cycles of N₂O concentration have been observed in other estuaries (Goncalves et al., 2015; Barnes et al., 2006).*

We chose our sampling strategy (upstream against the outgoing tide) to prevent interference of tidal effects on our measurements. Our aim was to obtain comparable data for each cruise at similar tidal phase, with comparable current and mixing conditions. We started after high-tide and travelled against the outgoing tide to make sure that we did not move with the same water masses while travelling upstream.

Tidal effects will very likely affect nitrous oxide concentrations in the Elbe estuary, but this is not the focus of the present manuscript. We briefly addressed possible tidal effects in the revision.

| Lines | Change |
|---|---|
| L94-95 | Explained our chosen sampling strategy |
| L416-420 | Addressed the possible effects of tides, diel variations and currents on N₂O emissions |

*Line 116 in Equation 1: is N2Ocw the partial pressure of N2O in water? Otherwise, the saturation should be calculated as*

*the N2Ocw/N2Oeq\*100 where N2Oeq is the equilibrated N2O concentration with atmosphere. Similarly in Equation 3.*

*N2Oair should be N2Oeq.*

For our calculations, we used the average atmospheric N₂O concentrations measured on each specific day of the cruise to calculate expected atmospheric equilibrium concentrations considering the solubility function of Weiss and Price (1980) and atmospheric pressure.

| Lines | Change |
|---|---|
| L125-126 | Changed: "and in the air ($N_2O_{air}$)" to "atmospheric equilibrium concentrations ($N_2O_{eq}$)" |
| L127 | Changed Eq. 1: "$N_2O_{air}$" to "$N_2O_{eq}$" |

*Line 143: Why nitrate concentration increased at 700 km? Are there tributaries or point sources?*

We regard this a result of nitrification, rather than a point source. We now refer to potential point sources in the revised version (see below). Dähnke et al. (2008) identified the Elbe estuary along its salinity gradient as a significant source of nitrate with high nitrate production in the maximum turbidity zone (MTZ). Sanders et al. (2018) measured highest nitrification in the

Hamburg port region, which was not covered in the research done by Dähnke et al. (2008). Both studies highlighted the importance of nitrification along the Elbe estuary. Further, our results in section 4.2 showed ongoing nitrification fueled by marine organic matter along the mesohaline estuary and indicated that nitrate is produced by coupled remineralization with nitrification from both the riverine and marine site of the estuary.

We believe the offset between region of highest nitrification rates (Port of Hamburg) and nitrate peak in the estuary (stream kilometer 680-700) is a result of different spatiotemporal scales, as suggested by Sanders et al. (2018). They found that the position of the nitrate maximum and nitrate gain over the entire estuary depended on the processing rates and was thus coupled to discharge conditions. However, we decided not to address the offset in our manuscript. Hopefully, the added information regarding the point sources will help to clarify the text, but addressing the nitrate dynamics in more detail is beyond the scope of the paper.

| Lines | Change |
|-------|--------|
| L78-79 | Added information about point sources to study site description |

*Lines 148-149: Why ammonium and nitrite concentration increased near Hamburg Port? Is it due to internal organic*

*matter remineralization or point sources or sedimentary flux?*

We restructured section 4.3 addressing possible nitrogen turnover processes and benthic-pelagic coupling in more detail.

Therefore, we added a short paragraph to clarify succession of nitrogen turnover in the Port of Hamburg.

| Lines | Change |
|-------|--------|
| L344-348 | Summarized state of research regarding nitrogen turnover in the Port of Hamburg |
| L352-356 | |

*Figure 2: It is hard to tell the difference among each cruise with so many colored lines. How about presenting data from*

*the same season using the same color to illustrate the seasonality as a supplementary figure?*

We tried to implement the suggestion of the reviewer. However, we felt that the figure did not help to illustrate seasonality and therefore we decided not to include it into the supplements (see figure below).

[Figure]

**Figure 1: Salinity along the Elbe estuary (a) in spring/summer and (b) in winter. Suspended particulate matter (SPM) concentration in (mg L⁻¹) along the Elbe estuary in (c) spring/summer and (d) in winter. Particulate carbon to nitrogen ratio (C/N) along the Elbe estuary in (e) in spring/summer and (f) in winter. Particulate nitrogen (PN) content in (%) in (g) spring/summer and (h) winter. Particulate carbon (PC) content in (%) in (i) spring/summer and (j) winter. All values are potted against stream kilometers. The Hamburg port region is shown with a gray background. C/N ratios were measured with an Elemental Analyzer (Eurovector EA 3000) calibrated against a certified acetanilide standard (IVA Analysentechnik, Germany). The standard deviation was 0.05% and 0.005% for carbon and nitrogen respectively. Please note that there are no data for the suspended particulate matter composition in 2015.**

*Lines 211-218 and lines 232-234: Figure 4 a and b are both from June, summer. The linear positive relationship between*

*AOU and excess N2O suggests N2O production from nitrification (e.g., Nevison et al., 2003). The increase in the slope*

*should be interpreted as an increase in the $N_2O$ production yield or external $N_2O$ input (e.g., point source).*

We indeed identified nitrification as responsible production processes both in the mesohaline estuary (section 4.2) and the

Hamburg port region (section 4.3). In the revised manuscript, we clarified that $N_2O$ yield varied due to changes in production, rather than point sources. As stated above, we also addressed the role of point sources in the study site description.

| Lines | Change |
|---|---|
| L78-79 | Added information about point sources to study site description |
| L236 | Presented all plots of AOU vs $N_2O_{xs}$ in a revised version of Fig. 3 |
| L237-239 | Changed figure caption to match new Fig. 3 |
| L229-235 | Changed to: "Plots of excess $N_2O$ ($N_2O_{xs}$) and apparent oxygen utilization (AOU) revealed excess $N_2O$ along the entire estuary (Fig. 3). During all cruises, elevated riverine $N_2O_{xs}$ entered the estuary (stream kilometer < 620). A linear positive relationship between $N_2O_{xs}$ and AOU suggested nitrification as main production pathway in large sections of the estuary (Nevison et al., 2003; Walter et al., 2004). However, in summer, a change of slope in the Port of Hamburg as well as in the mesohaline section of the estuary suggested either increased in-situ $N_2O$ production or external $N_2O$ input. In winter, we found an increasing slope in the Hamburg Port region and in the oligohaline part of the Elbe Estuary (Fig. 3h, k)." |
| L265-267 | Rewritten: "The relation of $N_2O_{xs}$ and AOU (Fig. 3), with changing slopes in the Port of Hamburg and mesohaline estuary, was determined by either initial riverine $N_2O$ production, or in-situ production along the estuary" |
| L286-287 | Rewritten: "The $N_2O$ peak in the transition between oligohaline and mesohaline estuary was accompanied by a sudden change in the slope of the AOU vs $N_2O_{xs}$ plots, (Fig. 3), pointing towards $N_2O$ production in the oxic water column" |

*Figure 4: It would be interesting to systematically/statistically assess the relations between excess $N_2O$ and environmental*

*factors like salinity (non-conservative behavior of $N_2O$) or dissolved inorganic nitrogen (infer $N_2O$ production pathways),*

*PN, PC, and SPM. There seems to be a good relation between $N_2O$ and ammonium/nitrite concentration shown in Figure*

*2.*

We understand that a systematically and statistical assessment of the relations would help the reader to follow our discussion.

We did assess the statistical relations in sections the previous version of our discussions, but for clarity, we added a section regarding the statistical analysis in the results chapter in the revised manuscript.

During data interpretation, we tested diverse presentation and analysis methods and found that regressions were not necessarily well suited to visualize, describe and analyze our data. Correlations were distorted by the spatial offset between the ammonium, nitrite and $N_2O$ peaks, which we attribute to a succession of nitrogen bearing substances during turnover processes like nitrification. Thus, we chose another way to visualize the data in Fig. 4 (L297) and Fig. S3-S13 of the supplementary material.

Furthermore, we addressed the relations of $N_2O$ and various forms of nitrogen to identify $N_2O$ production processes and their controls in the discussion (e.g. L292-L294, L314-L315, and L356-357).

| Lines | Change |
|-------|--------|
| L160-162 | Added method section regarding statistical analysis |
| L240-259 | Added result section regarding statistical analysis |

*Lines 242-243 and Figure 5: What about the variations of the N2O%, oxygen and total nitrogen concentration? The*

*riverine N concentration is decreasing, what about the changes in other point sources of N input along the estuary (e.g.,*

*from wastewater treatment plants) or concentration in the estuary?*

We agree with the reviewer that a more detailed analysis of a long-term trend of $N_2O$ concentrations and reasons for changes would be very interesting. However, for our study we focus on seasonal variations rather than a long-term trend analysis. With section 4.1, we aimed to compare our results with a broader spatial and temporal scale by including a short comparison to other estuaries as well as with previous measurements from the Elbe estuary. This gives a hint towards temporal trends, but seasonal variability and data coverage make the long-term data difficult to interpret.

Briefly, the biogeochemical processes occurring in the Elbe estuary have drastically changed over the last 50 years: (1) the reunification of Germany and the collapse of East-German industry had led to significant improvements of water quality (e.g.

Guhr et al., 2000). (2) The decision to combat eutrophication in the North Sea in the 1980s and (3) improved waste water management resulted in a significant reduction of riverine nutrient loads (de Jong, 2007, p.2019; Van Beusekom et al., 2019;

Bergemann and Gaumert, 2010). Dähnke et al. (2008) showed that this led to a change of dominating denitrification towards significant nitrification in the Elbe estuary. Thus, a profound long-term analysis would be in need of its own paper, for which we think our data coverage is insufficient, also considering the seasonal variability. As an example, measurements from only one cruise are available for the 1990s.

| Lines | Change |
|---|---|
| L78-79 | Added: "Point sources along the estuary provide only small part of the total nitrogen input to the Elbe Estuary (Hofmann et al., 2005; IKSE, 2018)" |
| L265 | Added comparison to $N_2O$ saturation with other highly modified urban systems (Reading et al. 2020) |
| L279-280 | Included reference to Dähnke et al. (2008) and change of dominating denitrification towards significant nitrification in the Elbe estuary |
| Fig. S2 | Removed Fig. 4 from text and added it to supplementary material with more plot panels and adapted figure caption |
| L275-278 | Added: "However, since the BIOGEST study in 1997 (Barnes and Upstill-Goddard, 2011), $N_2O$ remained relatively stable at ~ 200 % saturation despite a concurrent decrease in TN concentration from ~400 µmol $L^{-1}$ to around 200 µmol $L^{-1}$ (Fig. S2, Hanke and Knauth, 1990; Barnes and Upstill-Goddard, 2011; Brase et al., 2017; FGG, 2021)." instead of the figure to the text. |

*Line 272: "this suggests"*

| Lines | Change |
|---|---|
| L305 | Changed to "suggests" |

*Line 273: how is MTZ defined? What threshold of suspended particle material is used to define the MTZ?*

Generally, the occurrence of an MTZ is unique to each estuary and is generated by the balance between river-induced flushing
and upstream transport of marine SPM, as well as a function of estuarine geomorphology, gravitational circulation and tidal
flow, trapping the particles in the MTZ (Bianchi, 2007; Sommerfield and Wong, 2011; Winterwerp and Wang, 2013). Thus,
the MTZ is usually located in the onset of the salinity gradient of an estuary (Burchard et al., 2018).
The MTZ is – also in literature – often assessed based on relative changes in SPM or turbidity, and is mostly located between
stream km 670 and 710 (e.g. Bergemann, 2004) in the Elbe estuary. During our cruises, SPM and turbidity were not always
measured consistently, depending on instrument and personnel availability – during some cruises suspended particulate matter
concentrations were measured using filtration techniques, during some cruises we obtained the data from turbidity sensors,
which are not entirely intercomparable. Therefore, we did not define a threshold of suspended particulate matter to define the
MTZ, but used relative changes of SPM or turbidity for MTZ identification.
We added color bars indicating the relative change of SPM concentrations to Fig. 4 and supplement material.

| Lines | Change |
|---|---|
| L298 | Added color bars to Fig. 4 |
| Fig. S3-S13 | Added color bars to each figure |

*Line 287 and 296-297: clarify the reference: Kappenberg and Fanger, 2007 (German?) and source of organic matter from*

*the North Sea into the Elbe estuary.*

We included a peer-reviewed reference (Schoer, 1990) that shows an upstream transport of suspended matter into the Elbe estuary due to tidal transports in the Elbe estuary.

| Lines | Change |
|-------|--------|
| L320 | Included a new reference: (Schoer, 1990) |

*Lines 311-313: How about showing the relations between ammonium, nitrite and N2O in figures?*

We decided to show the relation between ammonium, nitrite and $N_2O$ plotted against stream kilometers in Fig. 4 and

Fig. S3-S13. We found that the spatial progression of nitrogen containing substances were more illustrative than scatter plots or correlations for each substance, cruise and production areas. These relations were distorted by the spatial offset between the occurring ammonium, nitrite and $N_2O$ peaks, which we explain by a succession of nitrogen bearing substances during turnover processes like nitrification (e.g. section 4.2). Therefore, we find our choice of presentation better suited. To address this issue, though, we added a section about the statistical analysis and relations of individual parameters in the results section. We further rephrased the statement in L311-313).

| Lines | Change |
|-------|--------|
| L160-162 | Added method section regarding statistical analysis |
| L240-259 | Added result section regarding statistical analysis |
| L361-364 | Changed to: "Overall, our data showed the succession of ammonium, nitrite and $N_2O$ production (Fig. 4 and supplementary material S3-S13) confirming simultaneous denitrification and nitrification responsible pathways for $N_2O$ production in the Port of Hamburg (Brase et al. 2017)." |

*Line 315: What are R values? R is positive for nitrite concentration.*

R is the Pearson correlation coefficient. We added a short section about our statistical analysis in the *Methods* section of our manuscript. Nitrite concentrations correlated positive with $N_2O$ leading to a positive correlation coefficient.

| Lines | Change |
|-------|--------|
| L160-162 | Added method section about statistical analysis |

*Line 320-321: Is nitrification responsible for the remaining oxygen consumption?*

Yes, nitrification is responsible for the remaining oxygen consumption. We have clarified this in the text.

| Lines | Change |
|-------|--------|
| L369 | Added: "whereas the remaining 25 % stem from nitrification (Schöl et al., 2014; Sanders et al., 2018)" |

**Line 326 and Figure S1: why C/N ratio was so high in 2021 March?**

We double-checked our measurements, and the data appear correct and sound. We have no easy explanation at hand, but speculate that a calcareous algae bloom in the North Sea might be a potential cause. However, we have no further evidence for this hypothesis. We will not address this further, as the C/N ratios are not a crucial parameter for our discussion. However, we would like to keep the data in the manuscript as they might be insightful for later research and other researchers.

**Line 345-347: "Ammonium and $N_2O$ concentrations are high in the pore water of underlying sediments". Reference or example of the concentration. What about the timing of deepening and dredging works in the Hamburg Port compared to the cruise periods?**

The Elbe estuary is constantly deepened and dredged along the entire transect to grant access for big container ships. However, we did not compare operation locations with the measured $N_2O$ concentrations except for our March cruises, as we did not see big differences in the spatial variation of the $N_2O$ profiles, which were not explainable by in-situ production, nor found strong correlations between $N_2O$ and suspended particulate matter concentrations.

| Lines | Change |
|-------|--------|
| L82 | Added: "The Elbe Estuary is dredged year-round" |
| L340-342 | Added references and rephrased: "Ammonium concentrations in the sediment pore water are high (Zander et al., 2020, 2022) and $N_2O$ can be produced by nitrifier-denitrification in the sediments (Deek et al., 2013)" |
| L331-350 | Moved from section 4.4 to 4.3 |
| L389-410 | Shortened discussion about possible effects of deepening and dredging in section 4.4 |

**Lines 360-361: Has there been any $N_2O$ measurement from this wastewater treatment plant (WWTP) Köhlbrandhöft? The ammonium concentration in 2021/03 is not exceptionally high compared to previous cruises (e.g., 2020/06). What about the direct N2O output from the wastewater treatment plant?**

The operators of the WWTP measured $N_2O$ concentrations during our cruise, but did not detect elevated $N_2O$ concentrations. However, direct $N_2O$ output from the WWTP was not measured and the increased ammonium loads leaving the WWTP corresponded to the measured increase of $N_2O$ concentration. We assume that excess $N_2O$ is not produced within WWTP itself, but stems from elevated ammonium concentrations in the Elbe that are introduced with warmer waste water.

We likely did not see an extraordinary ammonium peak due to the distance of the WWTP outflow and our measurement transect, ~ 2 km. The outflow of the WWTP is located in the Southern Elbe, which joins the sample stretch at stream kilometer 626. Ammonium is probably rapidly converted to $N_2O$ as the warmer and biological active waste water enters the Elbe estuary before it reaches the Northern Elbe (and our sampling site).

We were admittedly surprised by the extraordinary $N_2O$ concentrations in March 2021, and even more so when the results were not reproduced in the following year. Consequently, we concluded that an extraordinary event must have caused the $N_2O$

peak. Since we could not detect any relation to the deepening and dredging work in the Port area and our measurements fitted
to extraordinary operation condition in the WWTP, we assumed this might be the source, especially as our hypothesis was
confirmed by the WWTP operators. Additional measurements confirmed that $N_2O$ concentration was not elevated near the
WWTP outlet under normal conditions.

| Lines | Change |
|---|---|
| L399-410 | Revised: "Another possible source of $N_2O$ is the WWTP outflow in the Southern Elbe that joins the main estuary at stream kilometer 626 (Fig. 1), matching the $N_2O$ peak at stream kilometer 627 (Fig. 2h). As explained above (section 4.3), the effect of this WWTP on $N_2O$ saturations under normal conditions should be negligible. This peak can be the result of an extraordinary event during our sampling. We indeed found that an extreme rain event occurred on March 11th 2021 (HAMBURG WASSER, pers. Comm., Laurich 2022) with a statistical recurrence probability of one to five years (https://sri.hamburgwasser.de/, last access: 04.04.2023). This rare event caused aggravated operation conditions in the WWTP at the time of sampling. While the operators could still meet the limits for the effluent levels of nitrate and ammonium, higher than usual ammonium loads exited the treatment plant at this time. We assume that these elevated ammonium WWTP loads, were rapidly converted to $N_2O$ as the warmer and biologically active waste water entered the Elbe Estuary in March 2021.An important factor for aggravated conditions was a temperature drop in the WWTP caused by cold rain water, we hypothesize that a similar rain event in warmer months would not lead to comparable $N_2O$ peaks." |

*Figure 7. Use month or season as the x axis instead of cruise number? Add description of the boxplot. Why not adding*
*error bars for emissions?*
As we restructured section 4.5 of our discussion, we also changed the figure. However, we considered the comment of the
reviewer including description of the boxplots.

| Lines | Change |
|---|---|
| L442 | Deleted Fig. 7 included new Fig. 5 |
| L443-447 | Included description of boxplots in figure caption. |

*Table 3. How is annual $N_2O$ emission calculated? Since there is a seasonal variation in the $N_2O$ flux, monthly or seasonal*
*$N_2O$ emission may be more representative. Because $N_2O$ flux was measured at a high spatial resolution, it may be useful to*
*calculate the $N_2O$ flux across the whole estuary by integrating the flux and area section by section (e.g., River section,*
*Hamburg port, Oligohaline section) instead of multiplying the average $N_2O$ flux by the whole area of Elbe estuary.*
We recalculated emissions as suggested by the reviewer: We separated the Elbe estuary into five regions: limnic (stream
kilometer 585 to 615), Port of Hamburg (stream kilometre 615 to 632), oligohaline (stream kilometre 632 to 704), mesohaline
(stream kilometre 704 – 727) and the polyhaline section (stream kilometre 727 to 750). Respective areas are found in the supplementary material S6. For seasonality, we divided our cruises: winter (March), spring (April and May), summer (June and July) and late summer/autumn (August and September). Following this, we calculated daily emissions for each section and each season. To upscale to annual emissions, we applied our calculated emissions estimates to months without measurements (winter: January to March and November to December, spring: April to May, summer: June to July and late summer/autumn: August to October).

In line with reviewer 2, we also considered different wind speeds and parameterizations to calculate the gas transfer coefficient for flux densities calculation and emission estimates. Thus, we rewrote the results section and included our emissions estimates in section 3.3 "$N_2O$ flux densities and emissions".

| Lines | Change |
|-------|--------|
| L142-151 | We included a detailed description of the calculation in the "Method" section |
| L204-220 | We included detailed results in the "Results" section (results section was rewritten to include new emission calculations) |
| Table S2 | We included flux-densities calculations using other parametrizations and wind speeds in the supplementary material |
| L262 and L414-434 | We changed flux densities and $N_2O$ emission estimates in the revised manuscript so that they fit to the new calculations |

*Line 406: Why do you think there is no seasonality in $N_2O$ emission? $N_2O$ flux is different comparing spring, summer and winter shown in Table 3.*

For the revised manuscript, we calculated emissions as suggested by the reviewer and described above. We restructured our last section of the discussion 4.5 in line with suggestions of the other reviewer focusing on the relevance of our research for a broader audience, investigating the $N_2O$:DIN relation discussing seasonal changing drivers for $N_2O$ production and emissions. Thus, we also changed our abstract and conclusion.

| Lines | Change |
|-------|--------|
| L413-447 | Restructured and rewritten section 4.5 |
| | Removed from abstract: "Surprisingly, estuarine $N_2O$ emissions where equally high in winter and summer" |
| | Removed from conclusion: "We saw no seasonality in $N_2O$ emissions, …" |

**2. Review comment (RC2) – 04.04.2023**

*1. Converting dissolved concentrations to emissions: Like many studies, here the authors measured the dissolved concentration of the gas (N2O), and then converted this into water-air emissions based on a gas transfer velocity (k). Gas transfer velocities can be highly variable, especially in estuaries where the importance (and magnitude) of factors like wind,*

*flow velocity, and water depth can all vary a lot over space and time. This complexity is reflected in the wide range of*
*empirical k value parameterisations that have been developed for estuaries (see e.g., Rosentreter et al. (2021), also Hall and*
*Ulseth (2019) for a good review of the topic, albeit for freshwater systems). However, here the authors convert measured*
*concentrations to emissions using a single parameterisation (L116-125). This creates considerable uncertainty, which is*
*not reflected in the reported estuary emissions estimates. Emissions should be recalculated using 3-5 k parameterisations,*
*and the variability of these outputs reported in the results / figures. More information should also be supplied on the wind*
*speed data used in the parameterisations. It is important to understand how the values measured during the campaigns*
*compare to 'average' conditions around the estuary when considering the upscaled seasonal emissions values (e.g., are*
*emissions estimates likely to be on the low side because cruises were only done on low-wind days?).*

As suggested, we included calculations based on three other parameterizations. We calculated and discussed the effect of wind
speeds in relation to average conditions along the estuary and added the information concerning average wind speed.

| Lines | Change |
|---|---|
| L128-151 | Calculated flux densities and emissions with four parametrizations and different wind speeds |
| L204-220 | We included detailed results in the "Results" section for the new calculations of $N_2O$ flux densities and emissions |
| Table S2 | We included flux-densities calculations using other parametrizations and wind speeds in the supplementary material |
| L262 and L414-434 | We changed flux densities and $N_2O$ emission estimates in the revised manuscript so that they fit to the new calculations |
| L416-420 | Addressed the uncertainties in the "Discussion" section |

*2. Relationship between N2O and N inputs: As discussed in the paper intro here, aquatic N2O emissions are generally*
*predicted based on N loads to the system (i.e., leaching of N, inputs from WWTPs, etc). While here N2O emissions are*
*discussed and presented, the N inputs side of the equation is not clear to me. In the site description it says that annual N*
*load were ~80 Gg y-1 (L67) – but does this mean the estuary receives this much N, or discharges this much N? And how*
*does this break down between sources (WWTPs v river discharge)? On L231 it says that N2O emissions were low relative*
*to other high N input estuaries. But how do N inputs into the Elbe stack up compare to these other estuaries? I particularly*
*wonder how the 'point source' N loads around the port might stack up with those in other urban estuaries where N2O*
*emissions have been measured, e.g., (Wells et al., 2018). Constraining the other side of the N2O emissions v N inputs*
*equations is critical for placing these findings into a more global context. Within the study, more information on N loads*
*will also be important for picking apart the seasonal emissions drivers. How much N enters the estuary at the port? Is this*
*input seasonally variable? Did it vary between the sampled years? Do these variations correspond with variations in*
*emissions (particularly the size of the winter N2O-excess excursion)?*

The N-loads of 80 Gg yr$^{-1}$ are calculated from concentrations data at the station "Seemanshoeft", which is located at the Hamburg Port (stream kilometer 628.9). In general, point sources play a subordinate role in the nitrogen input of the Elbe estuary (Hofmann et al., 2005; IKSE, 2018) with dominating agricultural sources in the upper and middle Elbe River (Hofmann et al., 2005; Johannsen et al., 2008). Further, we calculated annual varying DIN and total nitrogen (TN) loads for our observation period and listed the results in the supplements as recent TN loads were lower varying between 43.1 kt-N yr$^{-1}$ and 70.2 kt-N yr$^{-1}$ from 2015 to 2021 (FGG, 2021).

We restructured our section 4.5 of the discussion: We now address the relation of N$_2$O emissions and N inputs based on a comparison of the amount of DIN released as N$_2$O (for annual loads, seasonal loads and for each cruise separately). In the revised version, we compare the relation of flux densities and N$_2$O emission versus N input to a wider set of literature data and across estuaries. We refer to the change of drivers of N$_2$O emissions in winter (high riverine input and nitrification) versus spring and summer (organic matter) more clearly. Finally, we highlight the link of N$_2$O emissions to eutrophication to broaden the scope of our study, which we also now address in the revised abstract, introduction and conclusion.

| Lines | Change |
|---|---|
| L19-20 | Rewritten abstract: "However, in spring and summer, N$_2$O saturation and emission did not decrease alongside lower riverine nitrogen loads […]" |
| L47-54 | Adding new research aim to investigate the driving factors of N$_2$O emissions along the estuary as well as looking into N$_2$O and DIN relation. |
| L78-79 | Added: "Point sources along the estuary provide only small part of the total nitrogen input to the Elbe Estuary (Hofmann et al., 2005; IKSE, 2018)" |
| L413-447 | Rewritten and restructured section 4.5 focusing on N$_2$O:DIN relation and comparing our results with other estuaries |
| L449-469 | Rewritten conclusion |
| Table S4 and S5 | Included annual and seasonal nitrogen loads for station Seemanshoeft |

*1. Introduction: It is not entirely clear how studying N2O in the Elbe estuary will advance understanding of aquatic N2O emissions / fill a needed research gap. A stronger transition between the penultimate and last paragraphs of the discussion is needed (how does the present study relate to the broader literature). Stating a testable hypothesis, rather than just site-specific study objectives, in the last paragraph may also help make the study more clearly relevant to the broader scientific community. Is this just a case study or will the data help us understand estuary N cycling and gaseous emissions in a more fundamental way?*

In the revised manuscript, we now elaborate a research question of interest for a wider audience by studying drivers for the reported discrepancies in the N$_2$O:DIN relation (Borges et al., 2015; Marzadri et al., 2017; Wells et al., 2018). Overall, the aim of our research is to provide insight on drivers of N$_2$O productions and emissions from heavily anthropogenic impacted estuaries.

| Lines | Change |
|-------|--------|
| L47-54 | Elaborated new research question from interest for broad scientific community |
| L58-59 | Added overall goal of our research |

*2. Discussion: While I think overall the data interpretation makes sense, the discussion section currently reads as a bit*
*descriptive and could go further to place these findings in a broader context (rather than just the context of how we*
*understand the Elbe River Estuary). This could include in particular more discussion of N cycling in urban estuaries /*
*where there are point N pollution. Where else in the world would the observed seasonal patterns be expected to be found?*
*I also think there is missing some discussion of 'alternative hypotheses' – work through the logic of why denitrification is*
*not thought to be the primary driver of N2O in the estuary, and why benthic production (e.g., (Chen et al., 2022)) is also*
*ruled out. Also please carefully edit to ensure that you are not repeating results in this section.*

We now discus alternative hypotheses in more detail as described below in the replies for the individual comments. We also
discuss the effects of benthic fluxes and production in more detail (see specific comment below). We have removed results
sections from the discussion section, and focused more on comparing our finding with research from other estuaries to address
a broader audience.

*3. Conclusion: This is currently very focused on untangling what exactly is happening within the Elbe River Estuary, but*
*the implications for broader understanding of aquatic N2O production and emissions are not clear.*

We modified our research question towards general controls of $N_2O$ production and emissions from estuaries with high
nitrogen loads. Consequently, we rewrote large parts of the discussion section 4.5. Thus, we also changed our conclusion and
abstract to highlight the new findings.

| Lines | Change |
|-------|--------|
| L27-29 | Rewritten abstract: "This effect of phytoplankton growth and the overarching control of organic matter on $N_2O$ production highlights that eutrophication and agricultural nutrient input can increase $N_2O$ emissions in estuaries." |
| L47-54 | Elaborated new research question from interest for broad scientific community |
| L58-59 | Added overall goal of our research |
| L413-447 | Rewritten and restructured section 4.5 focusing on $N_2O$:DIN relation and comparing our results with other estuaries |
| L449-470 | Rewritten conclusion |

*L17-19: This sentence is not clear (how does N2O 'compensate' for decreasing N loads?), please reword.*

This statement was also unclear to reviewer 1. Please see comment above for the changes in the revised version of our manuscript.

*L22-24: "In winter, high riverine N2O concentrations led to high N2O emissions from the estuary, whereas in summer,*

*estuarine biological N2O production led to equally high N2O emissions." This is I think getting at a crucial point (that*

*although seasonal magnitude of N2O fluxes did not differ the drivers of these fluxes did), the meaning is not clear. What*

*is the difference between winter 'high N2O concentrations' and summer 'high N2O production'? Reword to be more precise*

*about these differences.*

We revised the manuscript to highlight the relevance of our research for a broader audience. Thus, we rewrote our abstract, focusing on a comparison of our results with other research in heavily managed estuaries, highlighting that we found seasonal varying drivers of $N_2O$ emissions that did not scale with DIN loads and were directly linked to eutrophication phenomena.

Thus, we also re-wrote this phrase.

| Lines | Change |
|-------|--------|
| L18-21 | Changed to: "We found that the estuary was a year-round source of $N_2O$, with highest emissions in winter when dissolved inorganic nitrogen (DIN) loads and wind speeds are high. However, in spring and summer, $N_2O$ saturations and emissions did not decrease alongside lower riverine nitrogen loads, suggesting that estuarine in-situ $N_2O$ production is an important source of $N_2O$." |

*L70: How often is 'on a regular basis'? e.g., weekly, yearly, every three years?*

| Lines | Change |
|-------|--------|
| L82-85 | Added clarification |

*L86: Suggest changing 'steaming upstream' to 'travelling upstream' (steaming sounds a bit antiquated)*

| Lines | Change |
|-------|--------|
| L95 | Changed "steaming" to "travelling" |

*L101-104: More information on number of nutrient samples collected per survey, as well as method detection limits and*

*precision, would be useful.*

We will included the detection limits and also added a short description addressing the range of samples numbers.

| Lines | Change |
|-------|--------|
| L105 | Clarified numbers of samples taken |
| L113-114 | Added detection limits |

*L109: How often was 'regularly'? e.g., before each cruise?*.

| Lines | Change |
|---|---|
| L119-120 | Clarified: "Twice a day, we analyzed two standard gas mixtures of $N_2O$ in synthetic air (500.5 ppb ± 5 % and 321.2 ppb ± 3 %) to validate our measurements." |

*L116: How often, and how, was dry air sampled during each cruise?*

| Lines | Change |
|---|---|
| L126-127 | Clarified: "Atmospheric $N_2O$ dry mole fractions were measured before and after each transect cruises using an air duct from the deck of the research vessel." |

*L122: The term 'flux densities' is not one I'm familiar with – more common to see something like 'water-air fluxes' or 'evasion'.*

In physics, fluxes per unit area are called flux densities (Nitrous oxide mass flux | environmentdata.org, 2023), whereas the term "fluxes" only describe amount of $N_2O$ moving between the sea-air interface and is unitless. The terms are indeed often used synonymously.

We would like to stick to the term "flux densities", which has been used previously by other researchers (e.g. Brase et al., 2017; Bange et al., 2019; Morgan et al., 2019; Forster et al., 2009).

*L123-125: Please provide some clarification on the upscaling approach used to calculate whole-estuary emissions. From the description it sounds like the mean flux was multiplied by the estuary surface area? Or were these calculations area-weighted, and if so at what resolution?*

We changed our way of estimate $N_2O$ emissions, which we elaborated in detail for a comment of reviewer 1 above.

| Lines | Change |
|---|---|
| L142-151 | We included a detailed description of the calculation in the "Method" section |
| L213-220 | We included detailed results in the "Results" section |
| Table S2 | We included flux-densities calculations using other parametrizations and wind speeds in the supplementary material |
| L262 and L414-434 | We changed flux densities and $N_2O$ emission estimates in the revised manuscript so that they fit to the new calculations |

*L127-128: Citation?*

| Lines | Change |
|-------|--------|
| L154 | Added a citation |

*L148: Low relative to what?*

"low" is less than 1 $\mu$mol L$^{-1}$ – we will specify this.

| Lines | Change |
|-------|--------|
| L177 | Added: "(< 1 $\mu$mol L$^{-1}$)" |

*L163-189: Separating the N2O data into different sections for the different units (molar concentrations, % saturation, water-air fluxes) is confusing as these are all inter-related. For instances, it is hard to make sense of the meaning of the molar concentrations without also considering whether these reflect changes in percent saturation (i.e., changes due to water temperature / salinity v source / production). I suggest integrating these lines of data (and thinking) to provide a clearer picture of estuary N2O patterns.*

In Figure 2, we changed N$_2$O concentrations to N$_2$O saturations. Thus, we removed our previous Fig. 3 from the text. We also changed the results sections accordingly.

| Lines | Change |
|-------|--------|
| L182-190 | Replaced Fig. 2i, j with N$_2$O saturations and changed figure caption |
| L191-203 | Changed section 3.2 to "Atmospheric N$_2$O and N$_2$O saturation" |
| L204-220 | Changed section 3.3 to "N$_2$O flux densities and N$_2$O emissions" |
| | Deleted the description of N$_2$O concentrations |
| | Removed previous Fig. 3 from the text |

*L204: High relative to what?*

| Lines | Change |
|-------|--------|
| L225 | Specified: "(> 100 %)" |

*L209-218: The AOU v N2O-excess relationship really highlights the importance, and seasonality, of the port for estuary N2O emissions, with distinct peaks in the winter and consumption in the summer. Given that this underpins the discussion around seasonal N2O source switching, I wonder if there is a way to include more than just these 'representative' plots in the main text. For instance, a table with info on AOU v N2O-excess slopes, and min-max range for the port? I think if the port data is excluded something like an ANCOVA could be used to compare shifts in slope relationships.*

In the Port region, $N_2O_{xs}$ and AOU had no linear relation during most of our cruises (e.g. June 2015 and August 2017). Therefore, a table with slopes and min-max ranges would miss crucial information. However, we will include figure S2 from the supplement in the main text so that we do not only show representative plots but all cruises.

| Lines | Change |
|---|---|
| L236-239 | Included all AOU vs $N_2O_{xs}$ plots in Fig. 3 and changed figure caption |

*L256-260: Interesting relationship between NO2- and N2O. This could be connected to previous work, e.g., (Sharma et al., 2022; Smith and Bohlke, 2019; Wertz et al., 2018)*

| Lines | Change |
|---|---|
| L291-296 | Added: "This co-occurrence of nitrite accumulation and increased $N_2O$ saturation has been interpreted as signs for $N_2O$ production via denitrification (e.g. Wertz et al., 2018; Sharma et al., 2022). However, denitrification does not seem likely in this oxic water column. Such a succession of nitrite and ammonium peaks is also typical for remineralization and nitrification, and the slight decrease of oxygen concentrations around the higher $N_2O$ saturation (Fig. 2g and i) suggests oxygen consumption, possibly caused by these two processes." |

*L314-316: This should be in the results section*

| Lines | Change |
|---|---|
| L240-259 | Added section 3.4 describing results from our statistical analysis and removed this paragraph |

*L318-324: Interesting! I wonder if the algae themselves could also be contributing to the N2O production, e.g., (Fabisik et al., 2023)*

| Lines | Change |
|---|---|
| L372-373 | Added: "Fabisik et al. (2023) showed that algae could additionally contribute to $N_2O$ production. |

*L330-332: This makes sense, but is this the only possible explanation for high emissions around the port area? What about*
*wastewater inputs, enhanced benthic production, and/or enhanced groundwater connectivity due to dredging? Some*
*discussion of these points will make this conclusion stronger.*
We restructured our discussion focusing more on elaborating potential causes for the high $N_2O$ peak. We focused on (1) point
sources – mainly the wastewater treatment plant, (2) deepening and dredging operations in the Port of Hamburg and (3) in-situ
production, discussing both production in the water column and sediment. Therefore, we moved parts of section 4.4 to 4.3

| Lines | Change |
|---|---|
| L330-356 | Restructured the paragraph |
| L330-356 and L383-410 | Moved parts from 4.4 to 4.3 and thus, adapted section 4.4 |

*L357-358: How extreme was this rain event, i.e., was it more extreme than any rainfalls over the other five years of*
*sampling? This will help verify the attribution, and also put the pulse into context. It would then be instructive to recalculate*
*the seasonal budget with and without this pulse.*
Considering the statistical recurrence probability, it is likely that similar events occurred during the last five years, but so far
not during a comparable sampling cruise. Moreover, we assume that temperature was equally important as water mass, because
cold rain water in the waste water treatment plant led to aggravated operation conditions. Thus, in warmer months the effect
might be different. We discuss this in more detail in the revised manuscript.

| Lines | Change |
|---|---|
| L213-220 | Calculated $N_2O$ emissions estimates with and without the pulse and reported the variability |
| L403-404 | Added statistical recurrence probability |
| L408-410 | Added: "An important factor for aggravated conditions was a temperature drop in the WWTP caused by cold rain water, we hypothesize that a similar rain event in warmer months would not lead to comparable $N_2O$ peaks" |

*L392: If large riverine loads were the main driver, wouldn't there be a continuous decrease in concentration over distance?*
*But instead emissions peak in the port.*
We restructured the section 4.5 and thus removed this statement from the text.

| Lines | Change |
|---|---|
| L413-447 | Restructured section 4.5 |

 *Table 2: Standard deviations for the air N2O concentrations would be helpful*

| Lines | Change |
|-------|--------|
| L210-212 | Added standard deviations to the Tab. 2 |

*Fig. 1: The most important pieces of info in this map (where sampling points are, where the port is, where the MTZ is)*
*don't really stand out. Can you adjust colours, font size, etc to better highlight these key features? A scale bar for the main*
*map would also be helpful.*

We changed the style of map to highlight key features and important locations as suggested by the reviewer.

| Lines | Change |
|-------|--------|
| L86-88 | Changed Fig. 1 and figure caption |

*Fig. 2: I'm not sure that there is much value in showing N2O concentrations (in nM) here – the % saturation information*
*in the subsequent figure is much more effective for showing fluctuations between seasons and over the salinity gradient,*
*given the relatively low concentrations and the impact of both temperature and salinity on N2O solubility. It would also be*
*helpful to have 'summer' and 'winter' headings at the top of the two columns to make the point of difference more*
*immediately obvious.*

| Lines | Change |
|-------|--------|
| L182 | Included a heading to the plot |
| Fig. S1 | Included a heading to the plot |
| L182-190 | Changed Fig. 2i,j from $N_2O$ concentrations to saturations and changed figure caption |
| | Removed previous Fig. 3 from the text |

*Fig. 3: A unified y axis scale would be helpful for picking out seasonal differences*

| Lines | Change |
|-------|--------|
| | Removed previous Fig. 3 from the text |

*Fig. 4: As above, unified axes scales would make differences between sampling dates much clearer.*

We decided against unified y-axis for all cruises. The March 2021 cruise differs so much that it is hard to see variabilities in
the other cruises if we use the same y-axis. However, we used unified x-axes and y-axes scales for all other cruises.

| Lines | Change |
|-------|--------|
| L236 | Included all cruises in Fig. 3 with unified x-axes and unified y-axes except for March 2021 |
| L237-239 | Changed figure caption |

**Fig. 5: Different y axes are needed for the different variables (N2O, O2, TN), if not different plot panels**

| Lines | Change |
|---|---|
| Fig. S2 | Moved this figure to the supplementary material |
| L275-278 | Added instead of the figure: "However, since the BIOGEST study in 1997 (Barnes and Upstill-Goddard, 2011), $N_2O$ remained relatively stable at ~ 200 % saturation despite a concurrent decrease in TN concentration from ~400 µmol $L^{-1}$ to around 200 µmol $L^{-1}$ (Fig. S2, Hanke and Knauth, 1990; Barnes and Upstill-Goddard, 2011; Brase et al., 2017; FGG, 2021)" |
| Fig. S2 | Included a plot panel for each variable and changed figure caption |

**Fig. 6: I found this to be too many variables on the same plot to make much logical sense out of. I suggest separating into two panels, one for all of the N species (y axis unit is uM N), and then another with two y axes, one for PN and one for C/N.**

We adapted the plot as suggested by the reviewer.

| Lines | Change |
|---|---|
| L298-304 | Changed figure and figure caption |
| Fig. S3-S13 | Changed figure and figure caption |

**Fig. 7: It would be helpful to use a different pattern or colour scheme to distinguish the winter v summer cruises.**

We restructured and rewrote this part of the discussion and removed Fig. 7. The new Fig. 6 shows seasonal variations in $N_2O$ saturation, nitrous oxide emissions, DIN loads and $N_2O$:DIN ratios. We considered the comment of the reviewer regarding the color scheme for the new figure.

| Lines | Change |
|---|---|
| | Removed previous Fig. 7 |
| L442-447 | Added new Figure |

**References**

**References**

Nitrous oxide mass flux | environmentdata.org: http://www.environmentdata.org/archive/vocabpref:20656, last access: 27
April 2023.

Bange, H. W., Sim, C. H., Bastian, D., Kallert, J., Kock, A., Mujahid, A., and Müller, M.: Nitrous oxide ($N_2O$) and methane
($CH_4$) in rivers and estuaries of northwestern Borneo, Biogeosciences, 16, 4321–4335, https://doi.org/10.5194/bg-16-4321-
2019, 2019.

Bergemann, M.: Die Trübungszone in der Tideelbe - Beschreibung der räumlichen und zeitlichen Entwicklung,
Wassergütestelle Elbe, 2004.

Bergemann, M. and Gaumert, T.: Elbebericht 2008, Flussgebietsgemeinschaft Elbe, Hamburg, 2010.

Bianchi, T. S.: Biogeochemistry of Estuaries, Oxford University Press, New York, 706 pp., 2007.

Borges, A. V., Darchambeau, F., Teodoru, C. R., Marwick, T. R., Tamooh, F., Geeraert, N., Omengo, F. O., Guérin, F.,
Lambert, T., Morana, C., Okuku, E., and Bouillon, S.: Globally significant greenhouse-gas emissions from African inland
waters, Nat. Geosci., 8, 637–642, https://doi.org/10.1038/ngeo2486, 2015.

Brase, L., Bange, H. W., Lendt, R., Sanders, T., and Dähnke, K.: High Resolution Measurements of Nitrous Oxide (N2O) in
the Elbe Estuary, Front. Mar. Sci., 4, 162, https://doi.org/10.3389/fmars.2017.00162, 2017.

Burchard, H., Schuttelaars, H. M., and Ralston, D. K.: Sediment Trapping in Estuaries, Annu. Rev. Mar. Sci., 10, 371–395,
https://doi.org/10.1146/annurev-marine-010816-060535, 2018.

Dähnke, K., Bahlmann, E., and Emeis, K.-C.: A nitrate sink in estuaries? An assessment by means of stable nitrate isotopes in
the Elbe estuary, Limnol. Oceanogr., 53, 1504–1511, https://doi.org/10.4319/lo.2008.53.4.1504, 2008.

Deek, A., Dähnke, K., van Beusekom, J., Meyer, S., Voss, M., and Emeis, K.-C.: N2 fluxes in sediments of the Elbe Estuary
and adjacent coastal zones, Mar. Ecol. Prog. Ser., 493, 9–21, https://doi.org/10.3354/meps10514, 2013.

Fabisik, F., Guieysse, B., Procter, J., and Plouviez, M.: Nitrous oxide (N2O) synthesis by the freshwater cyanobacterium
Microcystis aeruginosa, Biogeosciences, 20, 687–693, https://doi.org/10.5194/bg-20-687-2023, 2023.

FGG, F. E.: FIS der FGG Elbe - Physikalisch-chemische Qualitätskomponenten: Elbe - Bunthaus (Strom-km 609,8) -
Wassertemperatur - Tagesmittelwert, 2021.

Forster, G., Upstill-Goddard, R., Gist, N., Robinson, C., Uher, G., and Woodward, E.: Nitrous oxide and methane in the
Atlantic Ocean between 50°N and 52°S: Latitudinal distribution and sea-to-air flux, Deep Sea Res. Part II Top. Stud.
Oceanogr., 964–976, https://doi.org/10.1016/j.dsr2.2008.12.002, 2009.

Guhr, H., Karrasch, B., and Spott, D.: Shifts in the Processes of Oxygen and Nutrient Balances in the River Elbe since the
Transformation of the Economic Structure, Acta Hydrochim. Hydrobiol., 28, 155–161, https://doi.org/10.1002/1521-
401X(200003)28:3<155::AID-AHEH155>3.0.CO;2-R, 2000.

HAMBURG WASSER, Laurich, F.: pers. Comm.: N2O in der Elbe, 2022.

Hofmann, J., Behrendt, H., Gilbert, A., Janssen, R., Kannen, A., Kappenberg, J., Lenhart, H., Lise, W., Nunneri, C., and Windhorst, W.: Catchment–coastal zone interaction based upon scenario and model analysis: Elbe and the German Bight case study, Reg. Environ. Change, 5, 54–81, https://doi.org/10.1007/s10113-004-0082-y, 2005.

IKSE: Strategie zur Minderung der Nährstoffeinträge in Gewässer in der internationalen Flussgebietsgemeinschaft Elbe, Internationale Kommission zur Schutz der Elbe, Magdeburg, 2018.

Johannsen, A., Dähnke, K., and Emeis, K.: Isotopic composition of nitrate in five German rivers discharging into the North Sea, Org. Geochem., 39, 1678–1689, https://doi.org/10.1016/j.orggeochem.2008.03.004, 2008.

de Jong, F.: Marine Eutrophication in Perspective, Springer, Berlin, Heidelberg, 335 pp., https://doi.org/10.1007/3-540-33648-6, 2007.

Marzadri, A., Dee, M. M., Tonina, D., Bellin, A., and Tank, J. L.: Role of surface and subsurface processes in scaling N2O emissions along riverine networks, Proc. Natl. Acad. Sci., 114, 4330–4335, https://doi.org/10.1073/pnas.1617454114, 2017.

Morgan, E. J., Lavric, J. V., Arévalo-Martínez, D. L., Bange, H. W., Steinhoff, T., Seifert, T., and Heimann, M.: Air–sea fluxes of greenhouse gases and oxygen in the northern Benguela Current region during upwelling events, Biogeosciences, 16, 4065–4084, https://doi.org/10.5194/bg-16-4065-2019, 2019.

Nevison, C., Butler, J. H., and Elkins, J. W.: Global distribution of N2O and the ΔN2O-AOU yield in the subsurface ocean, Glob. Biogeochem. Cycles, 17, https://doi.org/10.1029/2003GB002068, 2003.

Reading, M. J., Tait, D. R., Maher, D. T., Jeffrey, L. C., Looman, A., Holloway, C., Shishaye, H. A., Barron, S., and Santos, I. R.: Land use drives nitrous oxide dynamics in estuaries on regional and global scales, Limnol. Oceanogr., 65, 1903–1920, https://doi.org/10.1002/lno.11426, 2020.

Sanders, T., Schöl, A., and Dähnke, K.: Hot Spots of Nitrification in the Elbe Estuary and Their Impact on Nitrate Regeneration, Estuaries Coasts, 41, 128–138, https://doi.org/10.1007/s12237-017-0264-8, 2018.

Schoer, J. H.: Determination of the origin of suspended matter and sediments in the Elbe estuary using natural tracers, Estuaries, 13, 161–172, https://doi.org/10.2307/1351585, 1990.

Schöl, A., Hein, B., Wyrwa, J., and Kirchesch, V.: Modelling Water Quality in the Elbe and its Estuary – Large Scale and Long Term Applications with Focus on the Oxygen Budget of the Estuary, Küste 81 Model., 203–232, 2014.

Schulz, K. and Umlauf, L.: Residual Transport of Suspended Material by Tidal Straining near Sloping Topography, J. Phys. Oceanogr., 46, 2083–2102, https://doi.org/10.1175/JPO-D-15-0218.1, 2016.

Sharma, N., Flynn, E. D., Catalano, J. G., and Giammar, D. E.: Copper availability governs nitrous oxide accumulation in wetland soils and stream sediments, Geochim. Cosmochim. Acta, 327, 96–115, https://doi.org/10.1016/j.gca.2022.04.019, 2022.

Sommerfield, C. K. and Wong, K.-C.: Mechanisms of sediment flux and turbidity maintenance in the Delaware Estuary, J. Geophys. Res. Oceans, 116, https://doi.org/10.1029/2010JC006462, 2011.

Van Beusekom, J. E. E., Carstensen, J., Dolch, T., Grage, A., Hofmeister, R., Lenhart, H., Kerimoglu, O., Kolbe, K., Pätsch, J., Rick, J., Rönn, L., and Ruiter, H.: Wadden Sea Eutrophication: Long-Term Trends and Regional Differences, Front. Mar. Sci., 6, https://doi.org/10.3389/fmars.2019.00370, 2019.

Walter, S., Bange, H. W., and Wallace, D. W. R.: Nitrous oxide in the surface layer of the tropical North Atlantic Ocean along
a west to east transect, Geophys. Res. Lett., 31, L23S07, https://doi.org/10.1029/2004GL019937, 2004.

Weiss, R. F. and Price, B. A.: Nitrous oxide solubility in water and seawater, Mar. Chem., 8, 347–359,
https://doi.org/10.1016/0304-4203(80)90024-9, 1980.

Wells, N. S., Maher, D. T., Erler, D. V., Hipsey, M., Rosentreter, J. A., and Eyre, B. D.: Estuaries as Sources and Sinks of
N2O Across a Land Use Gradient in Subtropical Australia, Glob. Biogeochem. Cycles, 32, 877–894,
https://doi.org/10.1029/2017GB005826, 2018.

Wertz, S., Goyer, C., Burton, D. L., Zebarth, B. J., and Chantigny, M. H.: Processes contributing to nitrite accumulation and
concomitant N2O emissions in frozen soils, Soil Biol. Biochem., 126, 31–39, https://doi.org/10.1016/j.soilbio.2018.08.001,
2018.

Winterwerp, J. C. and Wang, Z. B.: Man-induced regime shifts in small estuaries—I: theory, Ocean Dyn., 63, 1279–1292,
https://doi.org/10.1007/s10236-013-0662-9, 2013.

Zander, F., Heimovaara, T., and Gebert, J.: Spatial variability of organic matter degradability in tidal Elbe sediments, J. Soils
Sediments, 20, 2573–2587, https://doi.org/10.1007/s11368-020-02569-4, 2020.

Zander, F., Groengroeft, A., Eschenbach, A., Heimovaara, T. J., and Gebert, J.: Organic matter pools in sediments of the tidal
Elbe river, Limnologica, 96, 125997, https://doi.org/10.1016/j.limno.2022.125997, 2022.

---

## Author Response (AR2)

**Response letter**

We thank the editor and all reviewers for their constructive and helpful comments and suggestions. In the following sections, reviewer comments are written in bold italics, our answers are kept in plain font. Note that we also slightly corrected the revised manuscript for stylistic issues and minor mistakes. These changes do not affect the conclusion of the manuscript and are shown in the marked-up manuscript version.

**1. Review comment (RC1) – 26.05.2023**

*I want to thank the authors for addressing my last round of comments. For example, the $N_2O$ flux and emission have been comprehensively estimated using different parameterizations and wind speeds. However, some of authors' arguments need to be supported by evidence or the authors need to discuss the caveats in the conclusion instead of making assumptions, e.g., the effect of temperature on $N_2O$ production, and $N_2O$ discharge from wastewater treatment plants. Here are my additional comments (line number in the modified manuscript).*

We thank the reviewer for their two rounds of in-depth, very helpful and constructive comments and suggestions that really improved our manuscript. In the following, we addressed each issue separately.

*23: It is not clear how $N_2O$ production was enhanced by warmer temperature based on the data presented in this study. Is it due to positive correlation between temperature and $N_2O$ saturation shown in Table 4? But not all the datasets had such pattern.*

We removed this statement from the abstract and conclusion.

| Lines | Change |
| --- | --- |
| L23 | Removed "$N_2O$ production was enhanced by warmer temperatures…" |
| L476-477 | Removed "Biological $N_2O$ production was enhanced by warmer temperatures and…" |

*25-26: "$N_2O$ saturation did not decrease alongside the decrease in DIN concentration…"*

| Lines | Change |
| --- | --- |
| L25-26 | Changed "…$N_2O$ saturation did not decrease alongside with DIN concentrations …" to "…$N_2O$ saturation did not decrease alongside the decrease in DIN concentrations…" |

***26-27: What does it mean for phytoplankton to reestablish? Were phytoplankton limited before 1990 when the water quality***

***was bad?***

Yes, that was indeed the case. Before the German reunification in 1990, organic nitrogen mainly emerged from industries and wastewater inputs. High organic matter concentrations, high pollutants levels and low light availability inhibited the developments of algae blooms in the 1980s as shown by the lack of seasonal variability in TON concentration (see Fig. 1

below). Management measures introduced in the 1980s and 1990s have led to an improved water quality, which in turn caused a significant increase of seasonal phytoplankton dynamics (Kerner, 2000; Amann et al., 2012; Hillebrand et al., 2018).

We further elaborate the cause of limited phytoplankton growth in Section 4.5 of our revised manuscript (also see our reply to a comment below).

[Figure]

**Figure 1: Seasonal variation of TON fraction in TN concentrations in (a) 1980s, (b) 1990s, (c) 2000s and (d) 2010s (Das**
**Fachinfomrationssystem (FIS) der FGG Elbe, 2022; Schulz et al., submitted).**

| Lines | Change |
|-------|--------|
| L458-463 | Rewrote: "The significant regime change after the 1990s enabled phytoplankton growth to reestablish in the river that had previously been inhibited by high pollutant levels and low light availability (Kerner, 2000; Amann et al., 2012; Hillebrand et al., 2018; Rewrie et al., submitted). The prevailing high nitrification rates in the estuary (Dähnke et al., 2008; Sanders et al., 2018) support an overarching control of organic matter on $N_2O$ production and emissions along the Elbe Estuary." |

***61-62: The emission factor (N₂O:DIN) derived from the Elbe River could be compared to different types of riverine systems,***

***not only the rivers with high agricultural nutrient inputs. Such comparison would broaden the results.***

Thanks to the reviewer for this comment, in the revised manuscript we also addressed other riverine systems.

| Lines | Change |
|---|---|
| L61-62 | Changed from "…used the $N_2O:DIN$ ratio for a comparison with other estuaries that receive similar high agricultural nutrient inputs" to "…used the $N_2O:DIN$ ratio for a comparison with other estuaries" |
| L445-447 | Added: "In general, $N_2O:DIN$ ratios vary widely (e.g., Baulch et al., 2012; Maavara et al., 2019; Smith and Böhlke, 2019). Wells et al. (2018) even found a range from -25 % to 7 % of DIN was emitted as $N_2O$ in estuaries with low land-use intensity." |

***Figure 1: What does the blue color mean in the figure? How is the section separated? Are they the same as shown in Figure***

***3 for AOU and excess N₂O or for emission calculation?***

The light blue colour indicates Wadden Sea areas that are exposed at low tide. We incorporated this into the Figure caption, and corrected the reference. Furthermore, we realised that the black line indicating the Hamburg Port region did not exactly match the separation made in Fig. 3. Thus, we changed the figure so that both regions match.

| Lines | Change |
|---|---|
| L87-88 | Changed Fig. 1 |
| L89-92 | Adapted figure caption: "Map of the Elbe Estuary sampled during our research cruises with stream kilometers (graphic courtesy of FGG Elbe, modified after Amann et al. 2012)). The light blue color indicates Wadden Sea areas that are exposed at low tide." |

***125: "N₂O saturation was calculated…"***

| Lines | Change |
|---|---|
| L129 | Changed "$N_2O$ saturation were calculated…" to "$N_2O$ saturation was calculated" |

***155: N₂Ocw has already been defined.***

| Lines | Change |
|---|---|
| L159 | Removed "$(N_2O_w)$" as definition for $N_2O$ concentration in water |
| L161 – Eq. 4 | Changed "$N_2O_w$" to "$N_2O_{cw}$" in Equation 4 |

*158: "an indicator for N₂O production from nitrification"*

| Lines | Change |
|---|---|
| L162-163 | Changed "… an indicator for nitrification (Nevison et al., 2003; Walter et al., 2004)" "… an indicator for N₂O production from nitrification (Nevison et al., 2003; Walter et al., 2004)" |

*206-209: It may be useful to have a figure to show the spatial distribution of average N₂O flux density as shown in Figure*
*2 for N₂O saturation and nutrients.*

We added the suggested figure to the supplementary material and refer to it in the main text, see Figure 2 below.

| Lines | Change |
|---|---|
| L209-210 | Add reference to supplementary material: "…, but also include results using other parametrizations in Table S2 and Fig. S2." |
| Fig. S2 | Added Figure of nitrous oxide flux densities along the Elbe estuary |
| Fig S4-S14 | Changed figure captions and references in the main manuscript |

[Figure]

**Figure 2: Nitrous oxide flux density along the Elbe estuary calculated after Borges et al. (2004) and in-situ wind speeds (a) in spring**
**and summer, (b) in winter. Light grey shading denotes the Hamburg Port region, dark grey shading the typical position of the**
**maximum turbidity zone (MTZ, Bergemann, 2004). Note the difference in Y-axis scales for the plots.**

*Section 3.5: It's clear to have a table to show the correlations between N$_2$O and environmental factors. However, no results*

*were described in this section. How about listing a few key results here such as the positive correlation with nitrite and*

*negative relation to oxygen?*

In line with the reviewer, we added a short description of key results to Section 3.5

| Lines | Change |
|-------|--------|
| L250-253 | Added: "N$_2$O saturation showed significant negative correlation with oxygen (Table 4) as well as a consistent negative correlation with pH (Table 4 and 5). Furthermore, nitrite concentrations positively correlated with N$_2$O saturation in the freshwater section of the estuary (Table 4 and 5)." |

*278-279: If the N loading decreased, but N$_2$O concentration or flux kept constant, it may suggest the yield of N$_2$O production*

*increased or the emission factor increased.*

Thanks to the reviewer for this comment. We change the respective sentence in Section 4.1.

| Lines | Change |
|-------|--------|
| L286-288 | Changed from "Since N$_2$O saturation did not decrease in scale with riverine nitrogen input, this suggests that in-situ N$_2$O production along the estuary is important." to: "As N$_2$O saturation did not decrease in scale with riverine nitrogen input, this suggests that the yield of N$_2$O production increased along the estuary." |

*Figure 4: Colormap of the relative change of SPM concentration is not shown for (a) and (b).*

We added a color bar to the plot.

| Lines | Change |
|-------|--------|
| L307-308 | Added color bar to Fig. 3 |

*304: (d) and (f)*

| Lines | Change |
|-------|--------|
| L314 | Changed to "(d) and (f)" |

*305: What does "This" represent?*

| Lines | Change |
|-------|--------|
| L315 | Changed "This…" to "This succession of N-bearing substances (Fig. 4, Fig. S4-14) suggests…" |

*319-320: Is there any evidence to support this argument (e.g., correlation between PN and N₂O)?*

In our dataset, we see no clear correlation between PN and nitrous oxide. However, there are several reasons that lead to a distortion of the correlation: (1) High suspended particulate concentrations in the MTZ lead to high PN concentrations - but if this organic matter is highly degraded, it will not add to N₂O generation, leading to distorted correlations. (2) The succession of nitrogen bearing substances during turnover processes like nitrification leads to a spatial offset between fresh organic matter input, ammonium, nitrite and N₂O peaks. In the mesohaline estuary, we identified nitrification as main production pathway for N₂O (section 4.2), which is fueled by fresh organic material from the North Sea providing ammonium as substrate via remineralization. Thus, the amount and quality of organic matter entering the Elbe Estuary has to affect N₂O by controlling nitrification rates. This is in line with findings by Dähnke et al. (2022), who identified the reactivity of particulate matter as the key control of the occurring nitrogen turnover processes along the freshwater Elbe estuary.

*334-338: Does the 5% means the fraction of DIN loading from WWTP to the total N load into the Elbe River? Are there*

*any measurements of N₂O associated with WWTP discharge or in the WWTP? Previous studies have shown that discharge*

*of WWTPs may be important source of N₂O to the aquatic system (e.g., Beaulieu et al., 2010; Chun et al., 2020). The*

*authors may need to acknowledge the possibility of N₂O discharge from WWTP instead of rejecting such possibility.*

We calculated the wastewater discharge fraction of stream flow according to Büttner et al. (2020), which is based on water masses. For the Elbe Estuary, point sources and WWTP are considered to only play a minor role in nitrogen loadings (Hofmann et al., 2005; IKSE, 2018). During various cruises, we took detours to the outlets of the WWTP in the Southern Elbe, which joins our sampling stretch at stream kilometer 626 and we never observed a significant change in N₂O concentrations. In addition, stable isotope signatures of nitrate did not show signs of a change in source contribution or an increasing importance of wastewater in the Hamburg port region. Thus, we believe the WWTP not responsible for the elevated N₂O concentrations under normal conditions. However, water temperature, riverine nitrogen load and freshwater discharge likely affect the importance wastewater input on N₂O concentrations and emissions. Thus, we acknowledged the possibility of N₂O discharge from the WWTP in the revised manuscript by also addressing previous studies as suggested by the reviewer.

| Lines | Change |
|---|---|
| L348-350 | Added "However, discharge of WWTPs can potentially be an important sources of N₂O (Beaulieu et al., 2010; Chun et al., 2020; Brown et al., 2022), and the effect of wastewater input on N₂O concentrations and emissions may change with altered river discharge, water temperature and riverine nitrogen loads in the future." |

*357-358: Why is nitrite favorable for N₂O production by nitrification? Low oxygen does not facilitate nitrification but*

*facilitates N₂O production from nitrification.*

We noticed that the statement about nitrite concentrations favouring $N_2O$ production by nitrification and nitrifier-denitrification is misleading and repetitive. $N_2O$ production by nitrifier-denitrification is enhanced by higher nitrite concentrations, which is a sub pathway during nitrification. We revised the sentence by naming only nitrifier-denitrification.

| Lines | Change |
|---|---|
| L369 | We remove nitrification from this sentence: "High nitrite concentrations are favorable for $N_2O$ production by nitrifier-denitrification (Quick et al., 2019),…" |
| L370 | Changed from "…while low-oxygen conditions facilitate both nitrification and denitrification." to "…while low-oxygen conditions facilitate $N_2O$ production from both nitrification and denitrification." |

*361-364: Not sure how sedimentary denitrification is proved to be an important source of N₂O. For instance, do you have*

*depth profiles of N₂O to show high N₂O concentration near the bottom water?*

We have no depth profiles of $N_2O$ concentrations in the Port of Hamburg, but we performed sediment incubation experiments that showed significant $N_2O$ fluxes from the sediments into the water column. However, these preliminary results are not ready for publication yet.

Brase et al. (2017) elaborated the production process leading to elevated $N_2O$ concentrations in the Hamburg port in detail, which is why we decided against a repetitive elaboration as we made the same observations: Available nitrate in the Port of

Hamburg possibly trigger $N_2O$ production via denitrification. Further, at minimum oxygen concentrations, the linear relation between AOU and $N_2O_{xs}$ breaks indicating other processes affecting either oxygen consumption and/or $N_2O$ production. We only measure surface oxygen concentrations and thus, we speculate that lower oxygen levels in deeper water layers and sediments may enable denitrification. Previous studies showed ongoing denitrification and nitrifier-denitrification in the sediments of this region (Deek et al., 2013) and Dähnke et al. (2022) even found signs for possible water column denitrification in the low oxygen zones. Thus, it only seems plausible that denitrification contributes to elevated $N_2O$ concentrations in the

Port of Hamburg. However, we currently cannot assess the contribution of sediments and the water column to overall nitrous oxide production and their respective controls.

| Lines | Change |
|---|---|
| L374-379 | Added: "Overall, our data showed the succession of ammonium, nitrite and $N_2O$ production (Fig. 4b and supplementary material S3-S13) as well as a breakup of the linear relation between AOU and $N_2O_{xs}$ in the Port region (Fig. 3). In combination with previous nitrogen process studies performed in the Elbe Estuary (Deek et al., 2013; Sanders et al., 2018; Dähnke et al., 2022), this supports simultaneous sedimentary denitrification and nitrification in the water column as responsible pathways for $N_2O$ production in the Port of Hamburg (Brase et al. 2017)." |

*407: This is a hypothesis but not an assumption: elevated ammonium concentration led to higher N₂O production.*

| Lines | Change |
|-------|--------|
| L422 | Changed to "We hypothesize that elevated ammonium WWTP loads were rapidly converted to $N_2O$…" |

*408-411: Why is the case (your hypothesis) if temperature is an important driver of N₂O production (warmer water likely*
*facilitates N₂O production)?*

High (cold) rainwater inflows led to a temperature drop in the WWTP that led to the aggravated operation conditions. In
summer, a comparable rain event would not result in a temperature drop in the WWTP and thus, no or at least not the same
aggravated operation conditions as in winter would occur. Lower ammonium concentration would exit the treatment plant,
which would not trigger intense $N_2O$ production in the estuary. We clarified that the inflow of cold rainwater was the cause
for aggravated operation conditions in the revised manuscript.

However, our results indicate that weather events can have drastic effects of on WWTPs and their respective rivers/estuaries.
The IPCC (2022) expects more frequent and extensive weather extremes, including both droughts and heavy rainfall.
Furthermore, we assume water temperature, riverine nitrogen load and freshwater discharge may influence the importance of
input from the wastewater treatment plant on $N_2O$ concentrations and emissions. Therefore, a possible assessment of inputs
from WWTPs in light of climate change with the overarching aim of evaluating management measures is an interesting topic
for future research.

| Lines | Change |
|-------|--------|
| L419-421 | Added: "This rare event caused a temperature drop in the WWTP due to high inflows of cold rainwater leading to aggravated operation conditions at the time of sampling." |
| L424-427 | Changed from "An important factor for aggravated conditions was a temperature drop in the WWTP caused by cold rain water, we hypothesize that a similar rain event in warmer months would lead to comparable $N_2O$ peaks"

 to

 "An important factor for aggravated conditions was a temperature drop in the WWTP caused by cold rain water (HAMBURG WASSER, pers. Comm., Laurich 2022), we therefore hypothesize that a similar rain event in warmer months would not have the same effect." |

*434: Any ideas about the drivers? temperature and oxygen?*

We consider several drivers responsible for seasonal varying $N_2O$:DIN relation. (1) Temperature: Sanders et al. (2018)
measured higher nitrification rates with warmer water temperatures along the Elbe Estuary. Further, warmer temperatures tend
to increase microbial processes (Murray et al., 2015; Quick et al., 2019) that could either lead to reduced or enhanced $N_2O$
production depending on the prevailing biogeochemical conditions. (2) Oxygen: Several researchers found oxygen availability
a key driver for $N_2O$ production in estuaries (e.g., de Bie et al., 2002; Rosamond et al., 2012; Murray et al., 2015; Yevenes et al., 2017), which also reflects in our data with significant negative correlations between oxygen and $N_2O$ saturation (Table 4
and Table 5). (3) Phytoplankton blooms in the upstream river and North Sea proving substrate for $N_2O$ production in the
estuary as well as leading to enhance oxygen depletion in the Hamburg Port further fuelling $N_2O$ production. We will briefly
address these possible drivers for the seasonal variation in the revised manuscript.

| Lines | Change |
|-------|--------|
| L452-455 | Added "…showing that this relationship even varies seasonally on site due to changing drivers for $N_2O$ production and emissions, e.g., temperature (Murray et al., 2015; Quick et al., 2019) and oxygen levels (de Bie et al., 2002; Rosamond et al., 2012; Yevenes et al., 2017)." The effect of organic matter availability is discussed in the following paragraph (L456-463). |

*456-454: I am trying to understand the decoupling of historical trend of $N_2O$ and DIN. Is there any historical change in*
*the concentration of organic matter in the river?*
Before the German reunification in 1990, organic nitrogen mainly emerged from industries and wastewater inputs. High
organic matter concentrations, high pollutants levels and low light availability inhibited the developments of algae blooms in
the 1980s as shown by the lack of seasonal variability in TON concentration (see Figure 1 to comment above). Management
measures introduced in the 1980s and 1990s have led to an improved water quality, which in turn caused a significant increase
of phytoplankton dynamics (Kerner, 2000; Amann et al., 2012; Hillebrand et al., 2018; Rewrie et al., submitted). Furthermore,
the improved oxygen conditions led to a shift from dominating denitrification to nitrification (Dähnke et al., 2008). The re-
establishment of primary production provides substrate for coupled remineralization and nitrification in the estuary (Sanders
et al., 2018; Dähnke et al., 2022) enhancing $N_2O$ production. We elaborated this connection in more detail in the revised
manuscript.

| Lines | Change |
|-------|--------|
| L458-463 | Rewrote: "The significant regime change after the 1990s enabled phytoplankton growth to reestablish in the river that had previously been inhibited by high pollutant levels and low light availability (Kerner, 2000; Amann et al., 2012; Hillebrand et al., 2018; Rewrie et al., submitted). The prevailing high nitrification rates in the estuary (Dähnke et al., 2008; Sanders et al., 2018) support an overarching control of organic matter on $N_2O$ production and emissions along the Elbe Estuary." |

**2. Review comment (RC3) – 13.06.2023**

*The manuscript reports on $N_2O$ concentrations, emission rates and the factors controlling this in the Elbe estuary. Overall,*

*I found this to be a very thorough data set that was well presented and interpreted. The figures were of a high quality and*

*the text was well written and organised. The factors giving rise to $N_2O$ emissions were well considered and discussed. This*

*manuscript will be of interest in the context of global $N_2O$ budgets as well as understanding the factors controlling $N_2O$*

*emissions. I only have a few minor comments. I congratulate the authors on producing such a well-rounded manuscript.*

We thank the reviewer for their nice and helpful comments about our paper and include their suggested changes.

*Table 1, final column. DIN concentration (as opposed to load).*

| Lines | Change |
|---|---|
| L103 – Table 1 | Changed "Average DIN load ($\mu$mol L$^{-1}$)" to "Average DIN concentrations ($\mu$mol L$^{-1}$)" |

*Line 360 – a larger fraction of N moving through the estuary is denitrified. This does not mean denitrification is more*

*intense, but rather the products accumulate (or are depleted) within the estuary due to longer residence times (in this case*

*$N_2O$).*

| Lines | Change |
|---|---|
| L372-374 | Changed from: "…, because denitrification and nitrification are more intense during longer  residence times (e.g. Nixon et al. 1996; Pind et al. 1997; Silvennoinen et al. 2007; Gonçalves et al. 2010)." To "…, because longer  residence times lead to the possible accumulation of $N_2O$ produced from either nitrification or denitrification (e.g. Nixon et al. 1996; Pind et al. 1997; Silvennoinen et al. 2007; Gonçalves et al. 2010)" |

*It is concluded that reducing nitrogen inputs alone would not reduce $N_2O$ emissions. It seems to me that N loads have been*

*reduced, but not enough to stop phytoplankton growth. Wouldn't reducing N loads further reduce phytoplankton growth*

*and eventually reduce $N_2O$ emissions?*

This is true. A reduced nitrogen input would reduce phytoplankton growth and thus organic matter availability and $N_2O$

emissions.  However, our data adds to the growing number of studies that clearly show a decoupling of the DIN:$N_2O$ ratio and the development of phytoplankton blooms is not solely controlled by nutrient inputs, but also by e.g., temperature, residence time, water depth and grazing. Thus, complex biological and chemical processes control phytoplankton dynamics (Scharfe et al., 2009; Dijkstra et al., 2019; Kamjunke et al., 2021), which will possibly change significantly in the future due to climate change (IPCC, 2022). A holistic approach to water quality mitigation and climate change adaptation is needed to prevent high

$N_2O$ emissions. We rewrote this section to highlight the other factors controlling phytoplankton dynamics in the revised manuscript.

[revised manuscript text omitted]

Schulz, G., van Beusekom, J. E. E., Jacob, J., Bold, S., Schöl, A., Ankele, M., Sanders, T., and Dähnke, K.: Low discharge
intensifies nitrogen retention in rivers – a case study in the Elbe River, Sci. Total Environ., submitted.

Silvennoinen, H., Hietanen, S., Liikanen, A., Stange, C. F., Russow, R., Kuparinen, J., and Martikainen, P. J.: Denitrification
in the River Estuaries of the Northern Baltic Sea, AMBIO J. Hum. Environ., 36, 134–140, https://doi.org/10.1579/0044-
7447(2007)36[134:DITREO]2.0.CO;2, 2007.

Smith, R. L. and Böhlke, J. K.: Methane and nitrous oxide temporal and spatial variability in two midwestern USA streams
containing high nitrate concentrations, Sci. Total Environ., 685, 574–588, https://doi.org/10.1016/j.scitotenv.2019.05.374,
2019.

Walter, S., Bange, H. W., and Wallace, D. W. R.: Nitrous oxide in the surface layer of the tropical North Atlantic Ocean along
a west to east transect, Geophys. Res. Lett., 31, L23S07, https://doi.org/10.1029/2004GL019937, 2004.

Wells, N. S., Maher, D. T., Erler, D. V., Hipsey, M., Rosentreter, J. A., and Eyre, B. D.: Estuaries as Sources and Sinks of
N2O Across a Land Use Gradient in Subtropical Australia, Glob. Biogeochem. Cycles, 32, 877–894,
https://doi.org/10.1029/2017GB005826, 2018.

Yevenes, M. A., Bello, E., Sanhueza-Guevara, S., and Farías, L.: Spatial Distribution of Nitrous Oxide (N2O) in the Reloncaví
Estuary–Sound and Adjacent Sea (41°–43° S), Chilean Patagonia, Estuaries Coasts, 40, 807–821,
https://doi.org/10.1007/s12237-016-0184-z, 2017.

---

## Author Response (AR3)

**Response letter**

We thank the editor for handling and carefully inspecting our manuscript. In the following sections, editor comments are written in bold italics, our answers are kept in plain font.

*Thank you for submitting this revised version to Biogeosciences. I have read it with pleasure and I am happy to inform you that your paper is now accepted for publication. However, while reading, I identified a few technical corrections.*

*Line 110-111: Clarify that you report particulate nitrogen (PN), particulate organic carbon (PC). I guess your PC data and C/N ratio only refer to the organic carbon fraction.*

Unfortunately, this is not the case. For multiple cruises, we also have organic carbon data. However, for some cruises only total particulate carbon measurements are available. Therefore, we decided to present and use only total particulate carbon fractions for C/N ratios. We clarified this in the manuscript.

| Lines | Change |
|---|---|
| L110-111 | Added:
 "[…] and subsequently analyzed for suspended particulate matter (SPM), particulate nitrogen (PN), total particulate carbon (PC) and C/N ratios (Fig. S1)." |

*Line 443: I guess the range is 25 to 7 % rather than -25 %.*

We checked the reference to validate -25%.

In Wells et al. (2018), p. 888, L:49-50: *"Although 1% of DIN entering the estuaries is expected to be released as $N_2O$ (Kroeze et al., 2005), here $N_2O_{total}$ accounted for anywhere from -25% (Moolooah dry season) to +7% (Nerang dry season) of $N_{in}$ […]"*

They found $N_2O$ under saturation in estuaries with low land-use intensity leading to a negative relation between $N_2O_{total}$ and nitrogen input.

21 *Line 474: Rewrie et al. reference as submitted. If you can update it to accepted/in press then include it, otherwise I suggest*

22 *deleting because it is not essential for the point you make (it is a reference in a list of 4).*

23 The research article is now published and accessible as online version before being included in an issue

24 (https://doi.org/10.1002/lno.12395). We updated the reference in the main text and in the reference list.

| Lines | Change |
|---|---|
| L455-456 | Changed citation to: "(Kerner, 2000; Amann et al., 2012; Hillebrand et al., 2018; Rewrie et al., 2023)" |
| L474 | Changed citation to: "(Kerner, 2000; Amann et al., 2012; Hillebrand et al., 2018; Rewrie et al., 2023)" |
| L734-736 | Changed reference:

"Rewrie, L. C. V., Voynova, Y. G., van Beusekom, J. E. E., Sanders, T., Körtzinger, A., Brix, H., Ollesch, G., and Baschek, B.: Significant shifts in inorganic carbon and ecosystem state in a temperate estuary (1985–2018), Limnol. Oceanogr., in press, https://doi.org/10.1002/lno.12395, 2023." |

25 **References**

26 Amann, T., Weiss, A., and Hartmann, J.: Carbon dynamics in the freshwater part of the Elbe estuary, Germany: Implications
27 of improving water quality, Estuar. Coast. Shelf Sci., 107, 112–121, https://doi.org/10.1016/j.ecss.2012.05.012, 2012.

28 Hillebrand, G., Hardenbicker, P., Fischer, H., Otto, W., and Vollmer, S.: Dynamics of total suspended matter and
29 phytoplankton loads in the river Elbe, J. Soils Sediments, 18, 3104–3113, https://doi.org/10.1007/s11368-018-1943-1, 2018.

30 Kerner, M.: Interactions between local oxygen deficiencies and heterotrophic microbial processes in the elbe estuary,
31 Limnologica, 30, 137–143, https://doi.org/10.1016/S0075-9511(00)80008-0, 2000.

32 Rewrie, L. C. V., Voynova, Y. G., van Beusekom, J. E. E., Sanders, T., Körtzinger, A., Brix, H., Ollesch, G., and Baschek, B.:
33 Significant shifts in inorganic carbon and ecosystem state in a temperate estuary (1985–2018), Limnol. Oceanogr., in press,
34 https://doi.org/10.1002/lno.12395, 2023.

35 Wells, N. S., Maher, D. T., Erler, D. V., Hipsey, M., Rosentreter, J. A., and Eyre, B. D.: Estuaries as Sources and Sinks of
36 $N_2O$ Across a Land Use Gradient in Subtropical Australia, Glob. Biogeochem. Cycles, 32, 877–894,
37 https://doi.org/10.1029/2017GB005826, 2018.